# Spike-phase coupling of subthalamic neurons to posterior perisylvian cortex predicts speech sound accuracy

Matteo Vissani [1,2] ✉, Alan Bush[1,2], Witold J. Lipski[3], Latané Bullock[1,2,4], Petra Fischer [5], Clemens Neudorfer [1,2], Lori L. Holt [6], Julie A. Fiez [7], Robert S. Turner [3] & R. Mark Richardson [1,2] ✉

Speech provides a rich context for understanding how cortical interactions with the basal ganglia contribute to unique human behaviors, but opportunities for direct human intracranial recordings across cortical-basal ganglia networks are rare. Here we have recorded electrocorticographic signals in the cortex synchronously with single units in the basal ganglia during awake neurosurgeries where participants spoke syllable repetitions. We have discovered that individual subthalamic nucleus (STN) neurons have transient (200 ms) spike-phase coupling (SPC) events with multiple cortical regions. The spike timing of STN neurons is locked to the phase of theta-alpha oscillations in the supramarginal and posterior superior temporal gyrus during speech planning and production. Speech sound errors occur when this STN-cortical interaction is delayed. Our results suggest that timely interactions between the STN and the posterior perisylvian cortex support auditory-motor coordinate transformation or phonological working memory during speech planning. These findings establish a framework for understanding cortical-basal ganglia interaction in other human behaviors, and additionally indicate that firing-rate based models are insufficient for explaining basal ganglia circuit behavior.

In everyday conversation, humans produce speech with remarkable accuracy and speed. Fluent speech requires the coordination and sequential movement of oral articulators on the order of milliseconds[1,2]. Brain networks with both cortical and subcortical nodes subserve the coordination of speech[3]. Cognitive neuroscience has made significant progress in refining the cortical speech-motor control network delineated by non-invasive imaging[4,5] using invasive recordings of the lateral perisylvian cortex[2,6]. However, less is known about the subcortical contributions to speech, especially how different nodes in the network transmit and share information.

The cortico-basal ganglia network is a structural foundation for supporting motor control[7,8], including human orofacial motor control for speech. Studies of speech and neurological speech impairments strongly support the idea that basal ganglia play a role in speech production. Positron emission tomography and functional magnetic resonance imaging have shown basal ganglia nuclei activation during speech production tasks[9–12] and have suggested a role of basal ganglia in timing, rhythm control, and prosody[13–16]. Clinical observations in patients with basal ganglia lesions or diseases affecting the basal ganglia bolster the findings from basic neuroscience. Lesions to adult basal ganglia can induce

[1]Department of Neurosurgery, Massachusetts General Hospital, Boston, MA, USA. [2]Harvard Medical School, Boston, MA, USA. [3]Department of Neurobiology, Systems Neuroscience Center and Center for Neuroscience, University of Pittsburgh School of Medicine, Pittsburgh, PA, USA. [4]Program in Speech and Hearing Bioscience and Technology, Division of Medical Sciences, Harvard Medical School, Boston, MA, USA. [5]School of Physiology, Pharmacology & Neuroscience, University of Bristol, Bristol, UK. [6]Department of Psychology, The University of Texas at Austin, Austin, TX, USA. [7]Department of Psychology, University of Pittsburgh, Pittsburgh, PA, USA. ✉e-mail: mvissani@mgh.harvard.edu; mark.richardson@mgh.harvard.edu

stuttering[17,18], articulatory impairments[19], and dysprosody[20]. Individuals with a mutated FOXP2 gene—which is thought to primarily affect neurons in the basal ganglia[21]—experience apraxia of speech along with linguistic and grammatical impairments, despite normal intelligence and hearing[22]. Approximately 90% of patients with Parkinson's disease (PD), whose most severe cardinal motor symptoms stem from basal ganglia pathology, suffer from a speech disorder known as hypokinetic dysarthria[23–25].

Deep Brain Stimulation (DBS) of the subthalamic nucleus (STN), a key basal ganglia node, reliably improves gross motor symptoms in PD, but its effects on speech are poorly understood. There is currently no consensus on why STN-DBS leaves speech unaffected or mildly improved in some patients[26–32] but contributes to speech decline in others[30,33–36]. Recordings from awake DBS surgeries offer a rare window to study the interactions between the STN and cortex during speech. The discovery of single unit[37–42] and population level[43,44] activity in the STN that tracks multiple aspects of speech production[41,45–47] and emerging evidence for anatomical[48] and functional connectivity[49] between the STN and sensorimotor and auditory cortical areas, raises the question of how STN and cortex interact to mediate speech-related behavior.

We established an intraoperative DBS protocol to simultaneously record local field potentials (LFPs) from high-density electrocorticography (ECoG) strips over speech cortex and single-unit activity from microelectrodes in the STN while PD patients completed a syllable repetition task[41,42,45,46,48,50]. This paradigm allowed us to study cortico-subcortical spike-phase coupling (SPC), which measures the degree to which spikes occur more often at certain phases of cortical oscillations. Importantly, SPC reveals inter-region coupling beyond changes in single-neuron firing rate or LFP oscillations power[51–54]. We tested the hypothesis that inter-areal SPC between STN neurons and cortical regions is modulated during the planning and execution of speech. We found that STN neurons phase-locked to cortical oscillations during short time intervals, which we call transient spike-phase coupling (t-SPC) events. Individual STN neurons had a preferred frequency at which they phase-locked to cortical LFPs: either in theta-alpha frequency band or in the beta frequency band. Furthermore, the cortical sites these STN units coupled to were spatially segregated; theta-alpha t-SPC events clustered over posterior perisylvian cortex (supramarginal gyrus (SMG) and superior temporal gyrus (STG)), while beta t-SPC events concentrated over sensorimotor cortex (precentral gyrus (PreCG) and postcentral gyrus (PostCG)). Participants were more likely to make substitution and omission speech errors on trials with lower, delayed theta-alpha t-SPC events. Thus, we discovered a temporally resolved, mechanistic characterization of cortical-basal ganglia interaction during speech production that furthers our understanding of information coding in the cortico-basal ganglia loop.

## Results

We studied intracranial recordings in 24 English-speaking participants (see Table S1 for clinical details) undergoing STN-DBS surgery for the treatment of Parkinson's disease. High-density electrocorticography (ECoG) across the left ventral sensorimotor cortex, STG, and inferior frontal regions were recorded simultaneously with single-neuron activity from the STN (Fig. 1A). Following the presentation of an auditory cue of a syllable triplet comprised of three unique phonotactically legal consonant-vowel (CV) syllables, participants were instructed to repeat the syllable triplet (speech production) at their own pace into an omnidirectional microphone (64 recording sessions, $2.67 \pm 0.62$ sessions, $379.88 \pm 99.49$ trials on average across participants) (Fig. 1B). Participants produced the CV-CV-CV sequences in $1.35 \pm 0.41$ s with a phonetic accuracy of $56.4 \pm 26.9\%$ (a triplet was considered inaccurate if any phoneme was off-target). Phonetic errors included consonant substitutions, such as the transformation of plosives into fricatives (e.g., /g/->/v/) and vice-versa ($66.3 \pm 20.7\%$), vowel substitution ($8.1 \pm 11.6\%$) and omissions ($25.7 \pm 19.9\%$) (Fig. S1 and Source Data). We did not observe an effect of the syllable position in the triplet on phonetic error frequency. Trained

speech-language pathologists annotated articulatory and voice features of each phoneme production (see Supplementary Text). All participants displayed some extent of articulatory imprecision ($6.8 \pm 5.3\%$ across phonemes) and creaky voice ($3.24 \pm 5.7\%$ across phonemes) (Table S2).

### Cortical potentials and subthalamic firing rates during speech production

Before addressing the complex interactions between cortical LFPs and STN single-neuron firing during speech, we analyzed each of the signals independently. We decomposed cortical LFPs from the lateral temporal and frontal cortex into time-frequency representations using Wavelet basis functions. We inspected LFP spectral components from 4 to 140 Hz. The expected cortical evoked activity was observed during both listening (locked to auditory cue onset) and speech production (locked to speech onset) (Fig. 1)[55–57]. Figure S2 illustrates different patterns of evoked activity in five representative electrodes from four participants. Spectrograms demonstrated consistent neural suppression in lower $\alpha$–$\beta$ frequencies (8–30 Hz) as well as elevation in the $\gamma$ range (50–150 Hz) during auditory cue presentation and speech production. $\beta$ power suppression was a ubiquitous phenomenon occurring across time and not temporally specific to processes related solely to speech. A large fraction of STG electrodes displayed either transient or sustained increased $\gamma$–activity in response to auditory cues[58]. In line with previous work[45,57], PreCG and PostCG showed $\gamma$–elevation preceding speech onset and during speech production, which likely reflects speech-related processing. The same channels demonstrated above-baseline $\beta$ increase (i.e., rebound) after the speech offset. Our data also revealed more complex response profiles, such as $\gamma$–activation of STG electrodes during speech (Fig. S2), consistent with the role of STG during auditory feedback[56,59].

From STN microelectrode recordings, we identified 245 neurons. 211 were stable and isolated (Fig. 1D shows recording density, on average $3.28 \pm 1.25$ neurons per recording session). Spike sorting and quality metrics were conducted as previously described[41,52]. We found neurons' instantaneous firing rates during the speech task were, as expected, heterogeneous both within and across recording sessions and patients (Fig. 1D). $N = 84/211$ neurons (40%) exhibited a significant increase in their firing rate in a window around the speech onset. Other neurons ($N = 37/211$, 18%) displayed a decrease in their firing rate. Interestingly, $N = 23/211$ neurons (11%) showed mixed behavior with both increased and decreased firing rates. The remaining neurons ($N = 67/211$, 32%) did not exhibit a significant modulation of firing rate during the speech production task. STN neurons with distinct firing rate modulations were not significantly spatially segregated within the STN, as assessed by comparing the average distance between these neuron categories against a null distribution of distances based on our sampling of recording sites (all $p_{\text{perm}} > 0.05$). For a comprehensive description of firing rate modulation in this dataset, see Lipski et al.[42].

### STN neurons lock transiently in a specific frequency band with cortical LFPs

Our preliminary analyses above showed that cortical LFP spectral power and STN firing rates were task-modulated. However, neuronal networks encode information with complex multivariate interactions, beyond what is found in power and firing rate changes[52,54]. Our simultaneous recordings at two different nodes in the cortical-basal ganglia loop allowed us to probe these network interactions. Do STN neurons consistently fire at a certain phase of cortical LFPs during speech planning and production? And if so, what is the duration of this spike-phase interaction? To address these questions, we used a variable-window width SPC estimation that provides an unbiased, time-resolved estimate of the strength of SPC across multiple frequencies (4–140 Hz) over the entire duration of the task, overcoming the limitations of traditional event-locked analyses that maximize the temporal precision only around a single event of interest (Fig. 1E). We set a target SPC temporal resolution of 50 ms, resulting in a series of

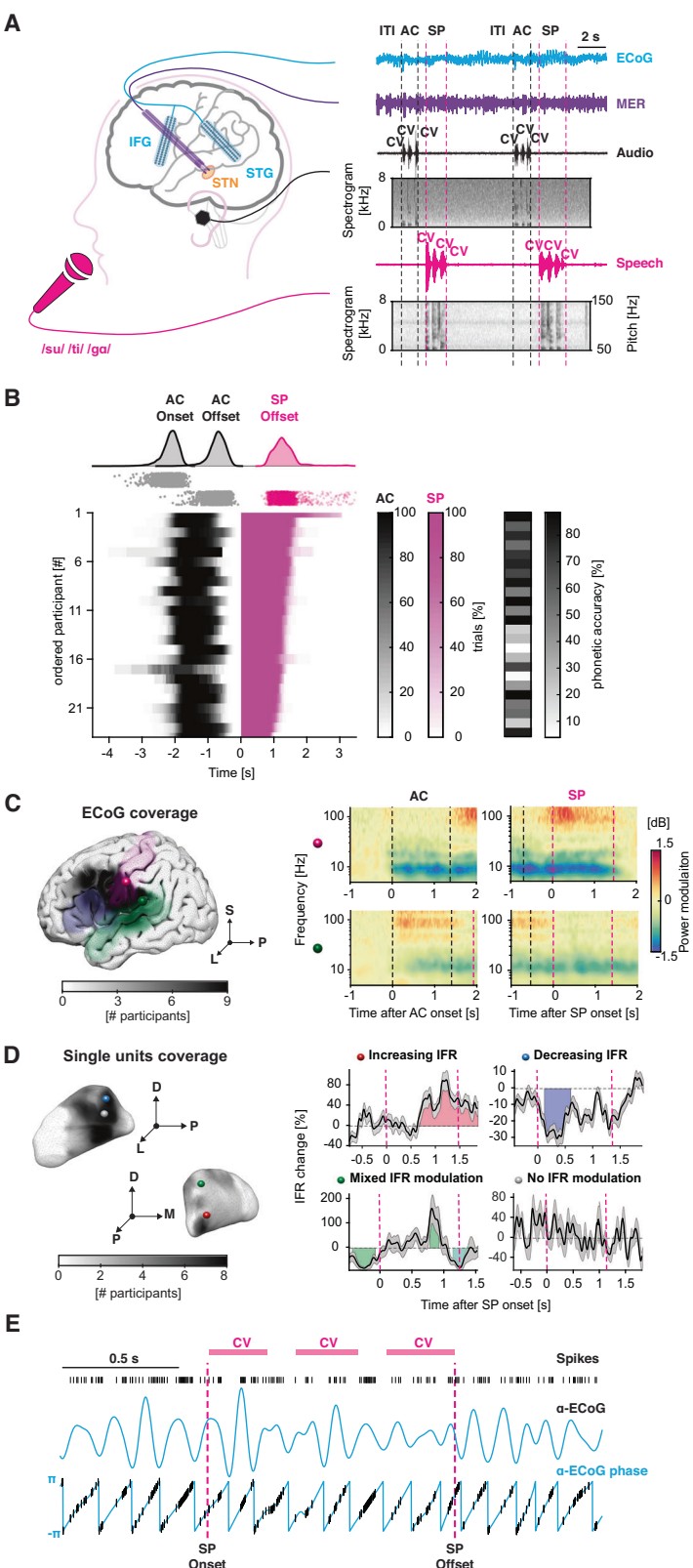

anchor points between contiguous behavioral events. We then adjusted window widths (target number of 0.15 s) around those anchor points to account for variability in the number of spikes during low and high firing rate periods, resulting in more accurate and less biased SPC estimates (please refer to "Methods" and Supplementary Text for details). The average window width was 0.15 ± 0.01 s (across pairs) with an average number of 350 ± 169 spikes across trials per window.

We obtained 19755 time-frequency SPC maps, with each map representing an STN neuron-cortical LFP pair. Maps specified SPC in frequency, from 4 to 140 Hz, and in time: from 0.75 s before the auditory cue onset to 0.75 s after speech production termination. Neurons had on average ~93 ± 33 cortical LFP pairs. Table S3 contains details about the number of pairs, participants, STN neurons, and ECoG contacts included in the main analysis. Cluster-based permutation tests revealed that ~11%

**Fig. 1 | Quantification of STN-cortex spike-phase coupling during an intraoperative syllable triplet repetition task. A** Illustration of the syllable triplet repetition task. Participants were instructed to repeat unique consonant-vowel ("CV") syllable triplets (magenta). The auditory stimuli were presented through earphones (black). High-density electrocorticography (ECoG) strips were placed in auditory and sensorimotor areas through the burr hole (cyan). Microelectrode recordings were acquired in the subthalamic nucleus during functional mapping (purple). Spectrograms of the audio signals are shown. **B** Timing of behavioral events, relative to speech-onset. Heatmap of the duration of the auditory cue (AC) and speech production (SP) windows expressed as a percentage across trials for each participant. The average phonetic accuracy of the produced syllables for each participant is shown on the right. **C** ECoG strips localizations. The coverage of the ECoG strips across participants is superimposed on three different target areas from the Destrieux atlas[26]: postcentral gyrus (purple), inferior frontal gyrus (blue), and superior temporal gyrus (green). Exemplary auditory-locked and speech-locked spectrograms of activity in the postcentral gyrus (purple sphere) and

superior temporal gyrus (green sphere) after normalization with respect to the baseline are displayed. **D** MER localization. Coverage of single units across participants is depicted in grayscale on the STN surface. Spheres denote the location of four exemplary neurons with different categories of instantaneous firing rate (IFR) modulation: Increasing (red), Decreasing (blue), Mixed (green), and No (gray) firing rate modulation. The plots show the percentage change in instantaneous firing rate (IFR) relative to baseline during the speech production window (indicated by the magenta vertical dashed line). The horizontal black line represents the mean IFR across trials, while the gray shaded area denotes the standard error of the mean (SEM). Colored patches highlight time bins with significant firing rate modulation (refer to "Methods" for details). **E** Exemplary transient spike-phase coupling in the $\alpha$ range during the speech production window (magenta dashed line). Spike timestamps, $\alpha$ oscillations, and instantaneous phase are illustrated. Magenta bars delineate the duration of the syllable triplet. ITI inter-trial interval used as baseline, AC auditory cue, SP speech production, IFR instantaneous firing rate.

(2148/19755) of the time-frequency maps displayed significant SPC. The distribution of the percentage of significant SPC pairs across participants is illustrated in Fig. S3.

When averaging only SPC maps ($N = 2148$) that were significant at the single-pair level, we observed SPC primarily in two distinct frequency ranges. STN neuron spiking uncoupled with cortical $\beta$ (13–21 Hz) with respect to baseline starting at auditory cue presentation and persisted throughout speech production (Fig. 2A). After speech offset, STN spikes increased in $\beta$ SPC beyond baseline, consistent with movement-offset $\beta$ LFP power rebound[60]. During the speech production interval, tonic ($\theta$–$\alpha$)-SPC below 10 Hz became a prominent feature (Fig. 2A). Notably, these results were consistent whether we averaged all SPC maps ($N = 19755$) or the most significant SPC map for each unit ($N = 211$) (Fig. S4 and Source Data).

We next sought to investigate the extent to which single-pair SPC maps accurately reflect the group-level SPC patterns observed during prolonged SPC changes. Strikingly, single-pair SPC maps showed that STN neurons locked to cortical LFP phases during transient episodes (Fig. 2B), which we termed t-SPC event. Although most significant SPC maps exhibit only one t-SPC (78%, 1682/2148), we also found examples with multiple (up to seven) t-SPC events, which can occur during different key events of the task and in different frequency bands (Fig. 2B, C). Thus, periods of increased or suppressed group-level SPC reflect the type of t-SPC event most likely to occur.

We then characterized the task-related timing and frequency centroid of t-SPC events to test whether STN neurons display speech-related frequency-specific SPC. Our analysis revealed that t-SPC events had a median duration of 0.268 s and occurred most frequently in the $\beta$ range (~16 Hz) (Fig. 2D). We observed a mild negative correlation between t-SPC duration and t-SPC frequency centroid ($R^2 = 0.12$ ($\rho = -0.35$), $p < 0.001$), suggesting that STN spiking is more likely to lock to low-frequency cortical oscillations for longer periods of time. Figure 2E lists all the t-SPC events ordered by frequency band. $\theta$–$\alpha$ t-SPC events significantly aggregate during speech production ($p_{perm} < 0.05$, permutation test) (Fig. 2F). Notably, $\alpha$ t-SPC events occurred throughout the entire speech production duration, whereas $\theta$ t-SPC events signaled preferentially the final part of the utterance. Moreover, $\alpha$ t-SPC events decreased during the auditory cue presentation ($p_{perm} < 0.05$, permutation test). $\beta$ t-SPC events were more prominent during the baseline, dipped during both auditory cue presentation and speech production, and rebounded above baseline levels after the termination of the utterance ($p_{perm} < 0.05$, permutation test). Neither $\gamma_L$ nor $\gamma_H$ t-SPC event occurrence showed prominent deviation from uniform distribution during the task.

We observed that SPC strength was frequency-band specific at the individual pair level (Fig. 2G). We tested whether this specificity was also evident when we aggregated t-SPC events at the single-neuron level. In other words, we examined the relationship between frequency

bands for neurons that had multiple t-SPC events: if a neuron coupled in one frequency band, was it more or less likely to couple in another band? To do this, we calculated a t-SPC index for each neuron, defined as the ratio of t-SPC events to pairs, and used an entropy-based metric to quantify the frequency specificity (please see Fig. S5). In general, we employed the t-SPC index for all analyses throughout the manuscript in which the neuron was the statistical unit of observation. A striking proportion of units ($N = 203/211$ 96%, frequency specificity ~0.56 higher than chance level: 0.08) was significantly specific to a frequency band. Finally, neurons that exhibited increased $\theta$–$\alpha$ t-SPC index did not show any modulation of $\beta$ SPC index, and vice-versa (Fig. S6).

We confirmed that t-SPC events observed in our data were not a by-product of the natural cluster tendency arising in small samples of random distribution. We evaluated the number of cycles of oscillations spanned by t-SPC events and the count of potential t-SPC events observed in the permuted SPC maps, previously used to convert the SPC maps into z-scores (see "Methods"). We used two cycles as the lower bound for a well-defined SPC event, as is commonly chosen in LFP oscillatory base analyses for $\beta$ bursts[61,62]. Ninety-nine percent of t-SPC events had more than two cycles (on average eight cycles). There were ~10 times more t-SPC events in the actual SPC maps compared to the shuffled SPC maps ($p_{perm} < 0.001$, permutation test across all frequency bands) (Fig. S7 and Source Data). These control analyses suggest that t-SPC events reflect genuine, physiological SPC mechanisms.

In summary, neurons tended to spike-phase couple to cortical LFPs transiently (for 0.25 s) and in a single frequency band. Neurons that coupled in multiple bands coupled in the $\theta$ and $\alpha$ range; these neurons seemed to couple to $\theta$ and $\alpha$ indiscriminately. Consequently, $\theta$ and $\alpha$ coupling were treated as a single entity in some analyses throughout the manuscript.

## STN-cortical spike-phase coupling is spatially organized and changes across task epochs

Previous studies have shown that neural activity in the cortex and STN feature spectral topographies during resting state[63,64] and movement execution[65,66]. Consistent with these findings, when we grouped SPC maps by cortical regions of interest (ROIs), we observed qualitatively distinct SPC patterns (Fig. S8). We extracted two t-SPC metrics to better delineate the spectral topography of the cortico-subcortical SPC during speech production. First, we quantified the spatial density of t-SPC events as the percentage of t-SPC events to total pairs, calculated separately at both the cortical and STN levels (see Fig. S5). We then compared this spatial density to a null distribution to identify ROIs with a significantly high or low prevalence of t-SPC events (Fig. 3A). Second, we characterized the temporal occurrence of t-SPC events, defined as the likelihood of observing at least one t-SPC event in each STN neuron-ECoG contact pair, across time and frequency band. Similarly, we tested t-SPC temporal occurrence against a null distribution to identify

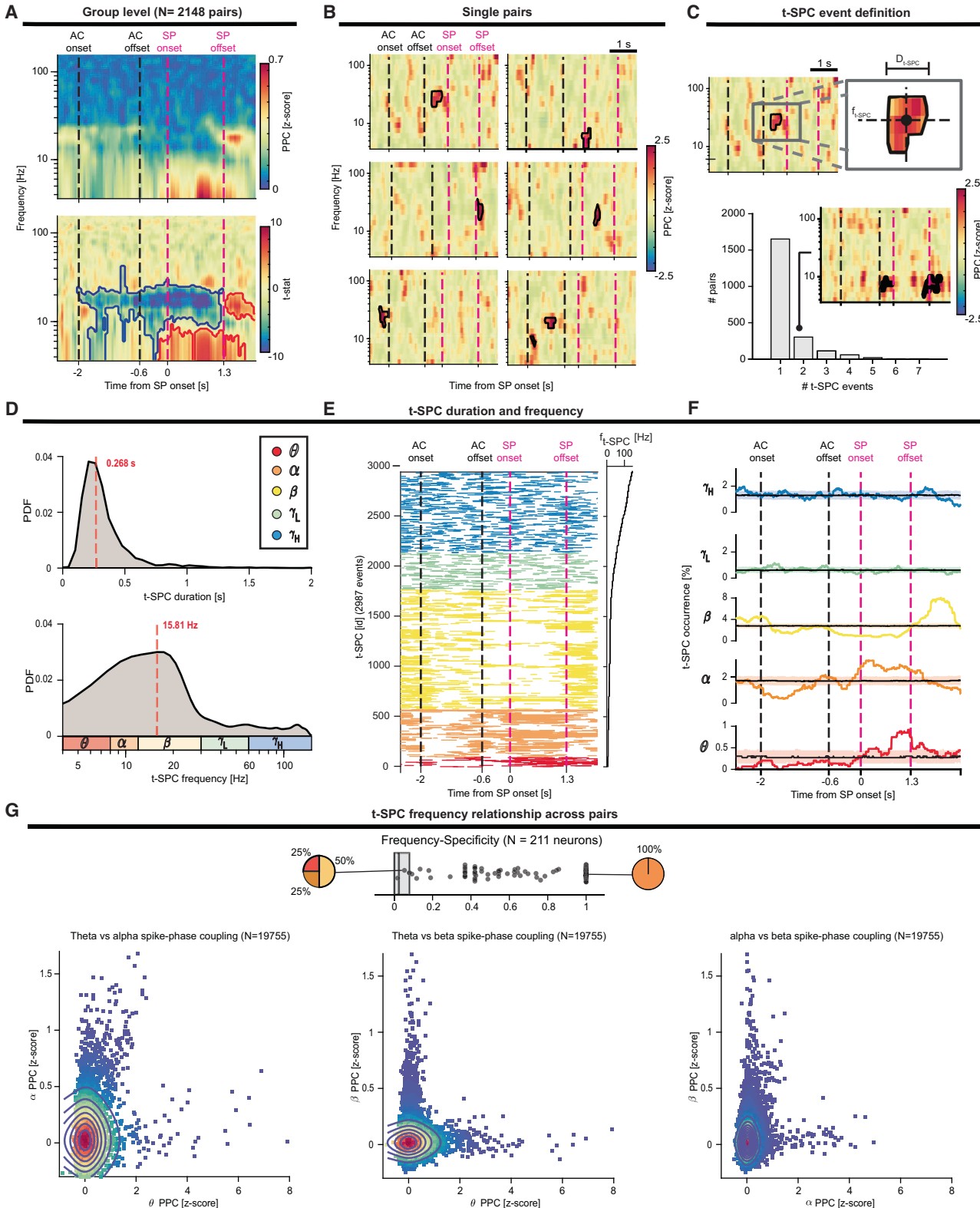

significant windows of aggregation (high overlap) or dispersion (low overlap) of t-SPC events (Fig. 3B, see "Methods" for details).

At the cortical level across all frequency bands, most t-SPC events were detected in the PostCG (17% significant pairs, t-SPC spatial density = 26%, $p_{perm} < 0.05$, permutation test) and SMG (13% significant pairs, t-SPC spatial density = 18.45%, $p_{perm} < 0.05$, permutation test). Significant numbers of t-SPC events were found also in the STG (14% significant pairs, t-SPC spatial density = 15%), PreCG (11% significant

pairs, t-SPC spatial density = 14%) and subcentral gyrus (SCG, 11% significant pairs, t-SPC spatial density = 16%), but t-SPC spatial density was lower in the middle and inferior frontal areas (Fig. 3A).

We observed aggregation (i.e., high occurrence) of $\theta$ t-SPC events during speech production in SMG and STG ($p_{perm} < 0.05$). Similarly, $\alpha$ t-SPC events dispersed (i.e., low occurrence) during auditory cue presentation and aggregated during speech production in SMG and STG regions, in addition to other areas such as PreCG and PostCG

**Fig. 2 | STN neurons show transient task-related coupling to the cortex in either θ−α or β bands. A** Average of the spike-phase coupling (SPC) maps with significant spike-phase coupling (N = 2148 pairs). The pairwise-phase consistency (PPC) index is compared to the permutation distribution and expressed as z-score. Group-level statistical test (t-stat) of the significance of the z-score PPC with respect to the baseline across all the significant pairs. Red and blue lines contour regions of significant SPC increase or decrease, respectively. Black and magenta vertical dashed lines denote auditory cue (AC) and speech production (SP) windows. **B** Examples of single-pair SPC maps show that STN neurons preferentially locked to cortical phases only during brief and transitory episodes. **C** Definition of the transient-SPC event (t-SPC event) in a single-pair SPC map. We calculated the onset and offset times, temporal duration, and the frequency centroid for each t-SPC event. Most pairs exhibit only one t-SPC event, as shown by the barplot. The inset plot depicts an exemplary SPC map with two t-SPC events in the same frequency band.

**D** Distribution of the t-SPC duration and t-SPC frequency centroid. To augment the readability of the t-SPC frequency distribution, we adopted a logarithmic scale. The red dashed line depicts the median of the distribution. **E** List of the t-SPC events (N = 2987) ordered by frequency centroid. **F** t-SPC events occurrence grouped by frequency band. Shaded areas illustrate the 5th and 95th percentiles of the permutation distribution for the aggregation test. θ (red), α (dark orange), β (yellow), γL (green) and γH (blue). **G** (Top) t-SPC frequency-band specificity (defined as one minus the entropy of the distribution of t-SPC events across frequency bands; see "Methods") is depicted for each neuron (N = 211). The pie charts depict the proportion of the t-SPC frequency band for two exemplary neurons. Dark gray boxes indicate the 5th and 95th percentile of the permutation distribution. (Bottom) 2D distribution of the SPC strength expressed as PPC (z-score) across all pairs (N = 19755) between different frequency bands (left: θ vs α, center: θ vs β, and right: α vs β). Colormap and contours indicate the 2D density of the scatter plot.

($p_{perm} < 0.05$). Baseline α t-SPC events were observed mostly in the MFG. Interestingly, α t-SPC events were not significantly spatially clustered around their centroid ($x = -62.23$ mm, $y = -13.04$ mm, $z = 30.4$ mm, $p_{perm} = 1$, Fig. S9 and Source Data). β t-SPC events were present at baseline and later dispersed temporally from auditory cue presentation through speech production in the PostCG and SCG ($p_{perm} < 0.05$). Interestingly, the β t-SPC rebound was a more widespread phenomenon, observed in PostCG and SCG, as well as in cortical regions like PreCG, SMG, and STG which did not exhibit t-SPC during the baseline ($p_{perm} < 0.05$). β t-SPC events were spatially clustered around their centroid ($x = -65.27$ mm, $y = -10.14$ mm, $z = 28.41$ mm, $p_{perm} < 0.05$, Fig. S9 and Source Data). $\gamma_L$ and $\gamma_H$ t-SPC events showed no preferential spatio-temporal distribution. We also compared the t-SPC event duration and centroid frequency across different ROIs. Longer t-SPC events with a lower frequency centroid occurred in the PostCG (~320 ms, ~20 Hz) and SMG (~310 ms, ~18 Hz) ($p_{perm} < 0.05$, permutation test) (Fig. S10 and Source Data). These results indicate that different epochs of speech perception and production are accompanied by frequency-specific STN-cortical SPC signatures.

We next investigated the location of STN units involved in SPC. Since the STN is not fully aligned with the MNI coordinates, we rotated the MNI reference frame to align with the STN's principal axes or components (PC): posterior-anterior axis (PC1), dorsal-ventral axis (PC2) and medial-lateral axis (PC3) (see Fig. S11A, B and "Methods" for details). θ t-SPC events were significantly aggregated in the posterior-medial region of the STN ($p_{perm} < 0.05$, $\theta_1$ in Figs. 4, S11C and S12, and Source Data). Moreover, θ t-SPC events were localized more dorsally (higher MNI z-coordinate) compared to t-SPC events in other frequency bands (Fig. S13 and Source Data). Two α t-SPC hotspots ($p_{perm} < 0.05$, Figs. 4, S11C and S12) were identified in the posterior-dorsal ($\alpha_1$) and posterior-ventral ($\alpha_2$) region of the STN. Of note, MFG SPC exclusively contributed to the posterior-ventral cluster. Overall, α t-SPC events were localized significantly inferior/ventrally (lower MNI-coordinate and PC2 coordinate) compared to t-SPC events in other frequency bands (Fig. S13 and Source Data). Spatial density analysis in the β range demonstrated more β SPC in the dorsolateral part of the STN during the baseline and rebound phases ($p_{perm} < 0.05$, $\beta_1$ in Figs. 4, S11C and S12). β SPC density appeared more focal during rebound than during the baseline (Fig. S11C and Source Data). A transient increase in β SPC events during auditory cue presentation was observed in the centro-medial region of the STN. Thus, STN spikes during t-SPC events exhibited a degree of frequency-dependent spatial specificity. Table S4 summarizes centroids of t-SPC event location and peak of t-SPC spatial density for each frequency band on the cortex and STN.

## Speech sound errors occur when θ−α spike-phase coupling is delayed
If SPC is an indicator of information transfer between cortex and STN, we hypothesized that SPC would correlate with speech performance.

Performance was defined by phonetic accuracy, i.e., percentage of speech sound errors. Accordingly, we split trials into correct and error trials based on whether the participant substituted or omitted any phonemes during speech production (see Fig. S1 for the patterning of errors and Source). We then computed SPC for correct and error trials (Fig. 5). As high-frequency SPC did not show any significant task-related modulation, we restricted this analysis only in pairs with significant SPC in the 4−40 Hz range and at least 20 trials in each condition (827 pairs in 46 neurons, please refer to Table S5 for details). This analysis revealed that error trials exhibited lower θ−α SPC preceding speech production, followed by an increase in θ−α SPC after the speech termination (Fig. 5A, B). Notably, no difference was observed during speech production. This correlation held whether we considered the SPC map (Fig. 5A) or the t-SPC occurrence (Fig. 5B) as a measure of SPC strength ($p_{perm} < 0.05$, permutation test). We also observed a significant increase of SPC strength in the high-β range (20−25 Hz) in error trials (Fig. 5A), but the result did not hold true when we looked at the SPC occurrence (Fig. 5B). We further hypothesized that θ−α t-SPC events occurred earlier in accurate trials than in error trials. We found that the median onset of θ−α t-SPC events is earlier in accurate trials than in error trials, in a within-neuron analysis ($p_{perm} < 0.05$, Fig. 5C, D). t-SPC duration was not affected by phonetic accuracy before or after speech production ($p_{perm} > 0.05$, Fig. 5E).

## Firing rate modulation predicts the preferred spike-phase coupling frequency
We used a variable-window width and pairwise-phase consistency (PPC) correction (Eq. (4)) to ensure that changes in coupling strength were not merely a result of firing rate modulation. However, these two neural phenomena may represent distinct, yet overlapping, mechanisms of modulation[54,67]. For most of these analyses, we used the t-SPC index−the ratio of the number of t-SPC events to all possible pairs at the single-neuron level−as an SPC measure to correlate with other single-neuron properties such as firing rate (see Fig. S5).

To investigate this question, we asked whether STN neurons with higher baseline firing rates had more cortical coupling (Fig. S14A and Source Data). Neurons showed no significant correlation between average firing rate and t-SPC index (ratio of t-SPC events to all possible pairs at the single-neuron level) ($R^2 = 0.006$, $p_{perm} = 0.23$), or average firing rate and t-SPC centroid frequency ($R^2 = 0.006$, $p_{perm} = 0.27$).

We then asked if change in firing rate for a given neuron was associated with increased coupling, and in which frequency bands. We plotted t-SPC index changes between behavioral epochs against z-scored firing rate modulation (with respect to baseline) during speech production (see Fig. S14B, "Methods" and Source Data). Again, t-SPC index changes were not correlated with firing rate modulation in any frequency band. We observed (θ−α) t-SPC changes (increase: 31/211 units, 15% and decrease: 2/211 units, 1%) only in neurons exhibiting low or negative firing rate changes (<5 z-score). Among the 27 neurons that displayed β t-SPC events during the

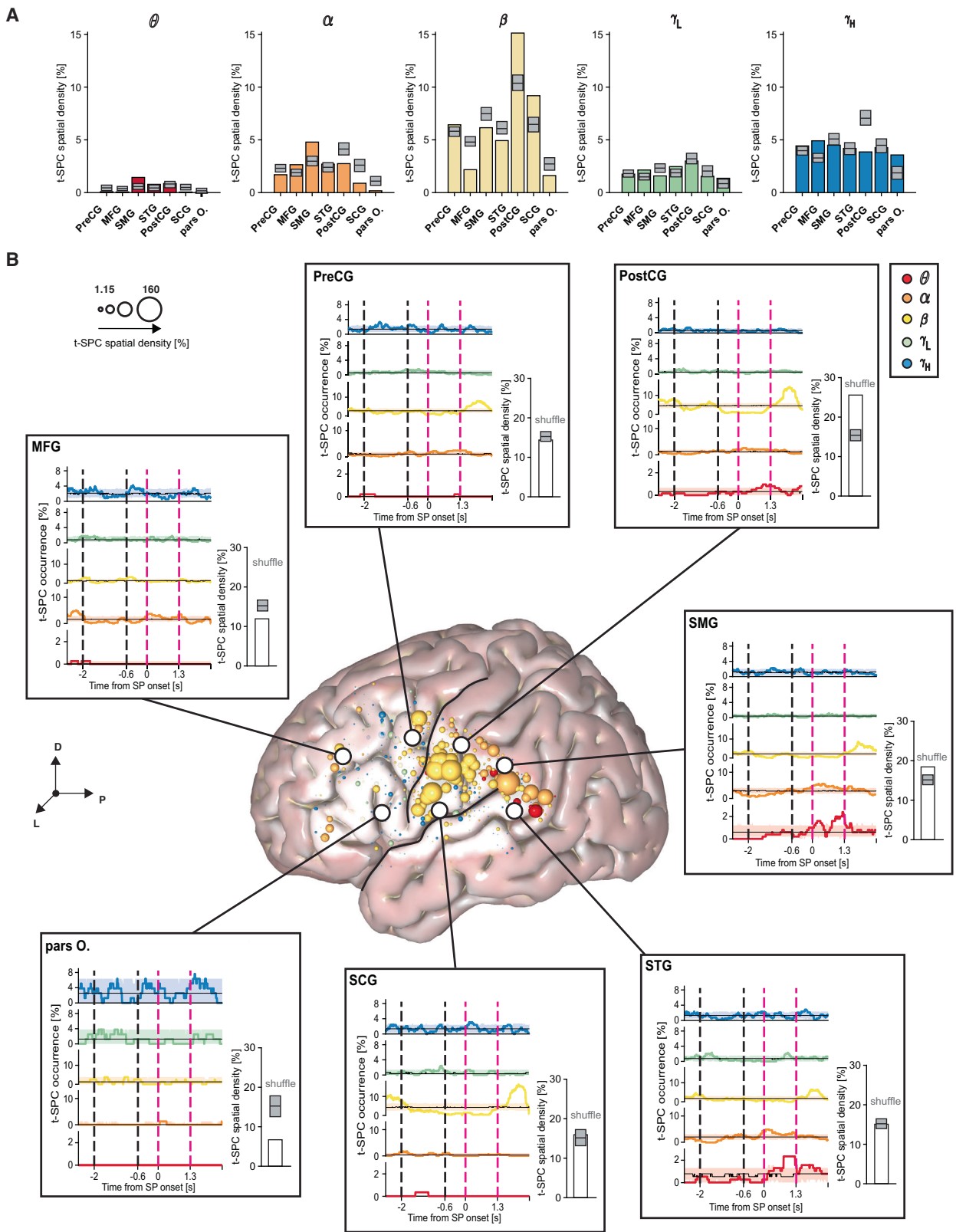

baseline, 25 (93%) neurons significantly reduced their $\beta$ t-SPC during speech production, either completely (22/25) or partially (3/25). Interestingly, a fraction of neurons (16/211, 8%) exhibited a slight increase in $\beta$ t-SPC during speech production, suggesting a partial maintenance of the $\beta$ SPC at the single-unit level. We found that these neurons ($N = 16/211$, 7.6%) are not specifically clustered in a specific region in the STN ($X = -13.50$ mm, $Y = -14.74$ mm, $Z = -8.25$ mm,

$p_{\text{perm}} = 0.065$). Interestingly, at the cortical level, these neurons mainly couple to the subcentral gyrus ($X = -62.82$ mm, $Y = -4.19$ mm, $Z = 29.14$ mm, $p_{\text{perm}} < 0.01$). Importantly, we observed no significant differences in firing rate changes between neurons that decreased $\beta$ t-SPC density during speech production or increased $\beta$ t-SPC density after speech termination and neurons with no changes in $\beta$ t-SPC density (Fig. S14 and Source Data).

**Fig. 3 | STN neurons couple to the SMG-pSTG in θ–α during speech. A** Spatial density of the t-SPC events across frequency bands in seven regions of interest, as derived from the Destrieux atlas[26]. We applied the t-max correction across ROIs in each panel to control for multiple comparisons. **B** Cortical spatial density map (of 2 mm) across frequency bands. The size of the spheres represents the degree to which t-SPC events are localized in a 2 mm radius around the center of the spheres. Inset plots illustrate the overall t-SPC spatial density (white bar plots) and t-SPC event occurrence in each region of interest. Shaded areas illustrate the 5th and 95th percentiles of the permutation distribution for the aggregation test. Dark gray boxes indicate the 5th and 95th percentile of the permutation distribution for the spatial preference test. Regions with spatial density higher or lower than the permutation distribution are labeled as high or low spatial preference. Black and

magenta vertical dashed lines denote auditory cue (AC) and speech production (SP) windows. θ (red), α (dark orange), β (yellow), $\gamma_L$ (green) and $\gamma_H$ (blue). Black lines on the cortical surface delineate two anatomical landmarks: the Sylvain fissure (SF), which divides the temporal from the frontal and parietal lobes, and the Central sulcus (CS), which separates the Precentral gyrus (PreCG) anteriorly from the Postcentral gyrus (PostCG) posteriorly. List of cortical regions of interest: Precentral gyrus (PreCG), Postcentral gyrus (PostCG), Supramarginal gyrus (SMG), Subcentral gyrus (SCG), Superior temporal gyrus (STG), posterior Superior temporal gyrus (pSTG), Middle frontal gyrus (MFG), and the orbital part of the inferior frontal gyrus (pars O.) The anatomical reference of the frame shows the dorsal (D), lateral (L), and posterior (P) directions.

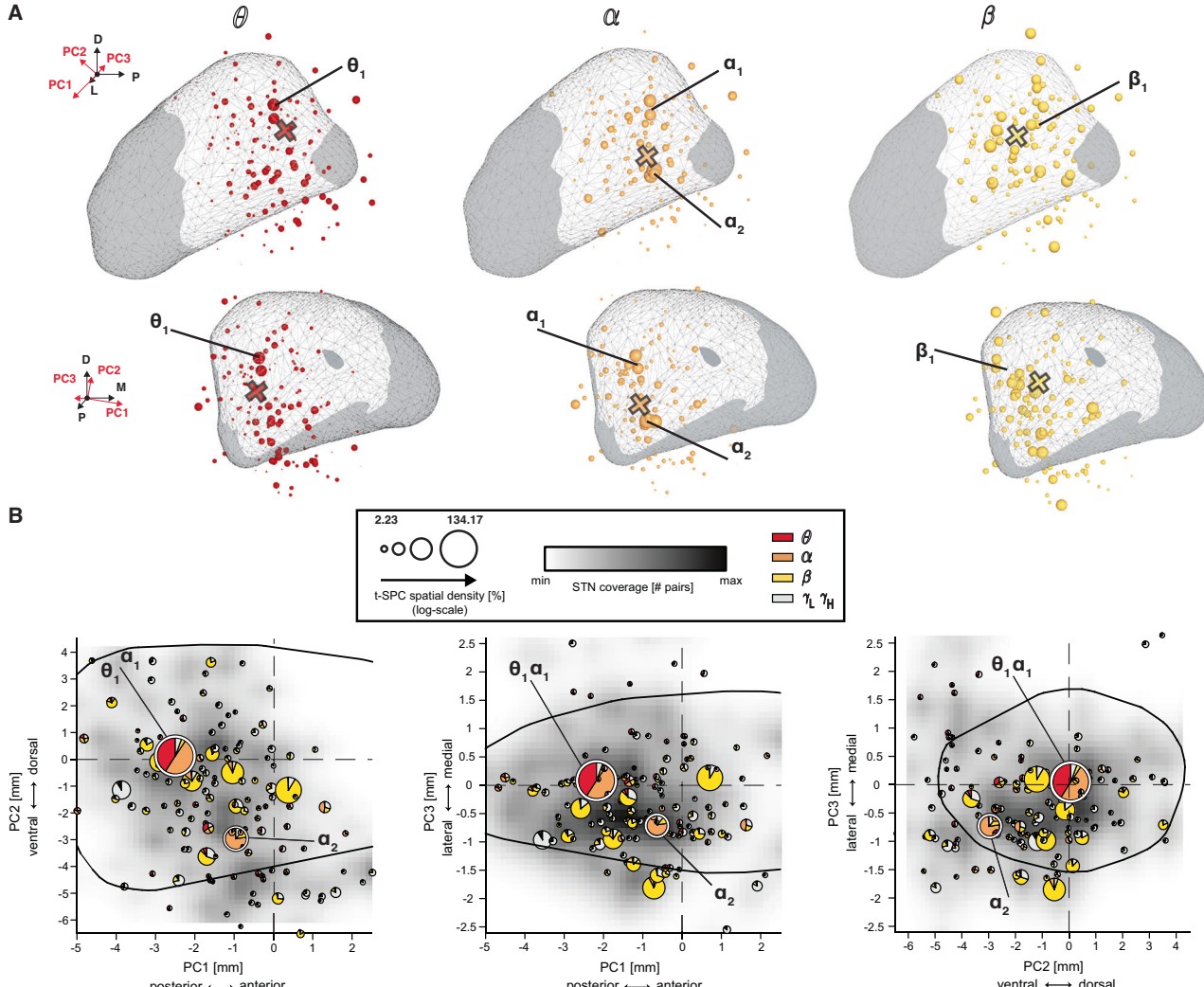

**Fig. 4 | θ–α SPC neurons are localized in the posterior part of STN.**
**A** Subthalamic spatial density maps (radius of 1 mm) across frequency bands. STN regions sampled by microelectrode recordings are depicted as white overlay. The size of the spheres represents the degree to which t-SPC events are localized in a 1 mm radius around the center of each sphere. To augment the readability of the visualization, we adopted the logarithmic scale for the spatial density. The anatomical reference of the frame shows the relative orientation between the dorsal (D), lateral (L), and posterior (P) directions and the first three principal components directions (PC1: anterior-posterior axis, PC2: dorso-ventral axis and PC3: medio-

lateral). θ (red), α (dark orange), and β (yellow). Black and magenta vertical dashed lines denote auditory cue (AC) and speech production (SP) windows. Cross indicates the spatial centroid of the t-SPC event locations. $\theta_1$, $\alpha_1$, $\alpha_2$, and $\beta_1$ depict the location of peaks of the t-SPC spatial density. **B** Spatial density of the t-SPC events mapped along the three principal component axes. The intersection of the two dashed black lines represents the STN center of mass. The radius of the pie-chart represents the t-SPC spatial density across bands. The black contour delineates the STN border as depicted by the DISTAL atlas. Note that principal component scores represent actual physical distances in mm.

Next, we compared the speech-related SPC across different firing rate categories. Neurons whose firing rates were modulated by speech (8.3%) exhibited similar t-SPC indices as neurons without speech-related firing rate modulation (8.7%). Among the speech-modulated

neurons, those with a decreased firing rate had the highest t-SPC index (13.4%), surpassing both neurons with an increased firing rate (6.5%) and those with mixed modulation (10%) ($p_{perm} < 0.01$, Fig. S15A and Source Data). Next, we compared the centroid duration and frequency

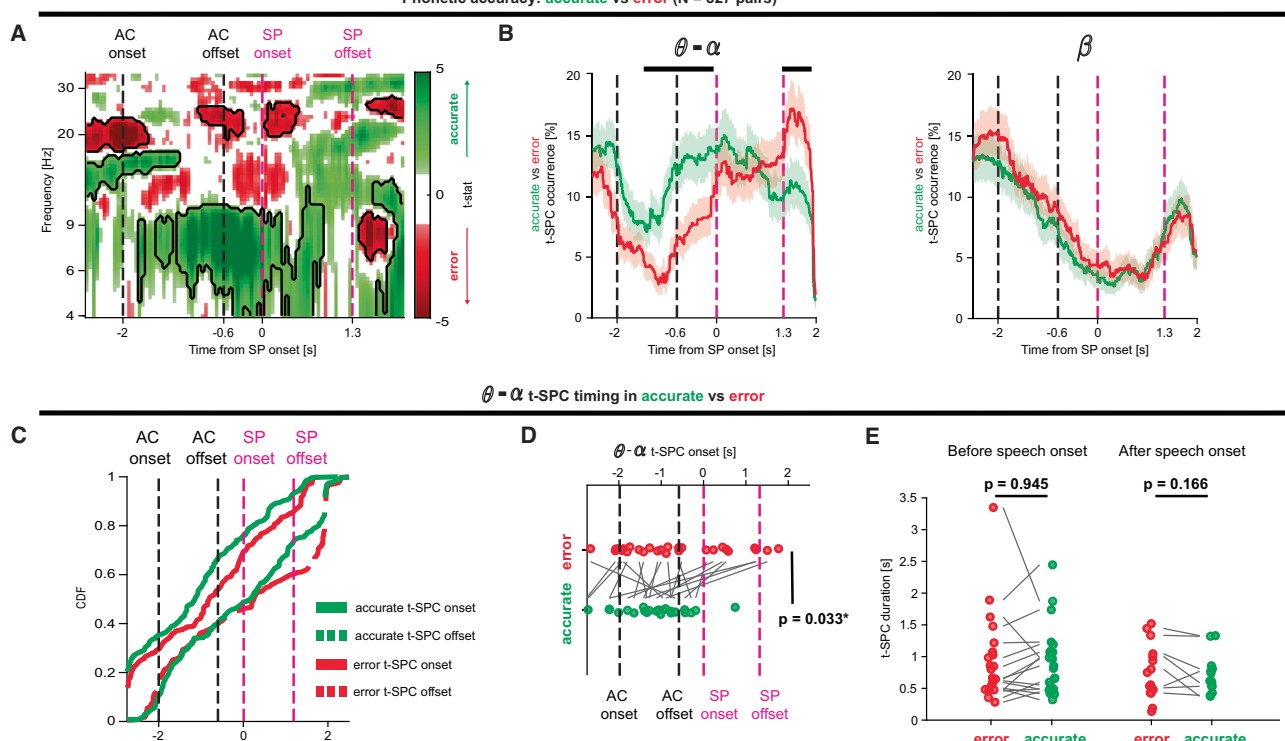

**Fig. 5 | Phonetic errors are correlated with delayed, reduced θ–α STN-cortex coupling. A** Comparison of the spike-phase coupling (SPC) maps ($N = 827$, 74 neurons in 18 participants) between trials with and without phonetic errors (see "Methods"). In error trials (red), θ spike-phase coupling before speech production onset was significantly lower than in correct trials (green) (cluster-based permutation test). **B** t-SPC events occurrence in trials with and without phonetic errors grouped by frequency band (θ–α: left, β: right). Shaded areas illustrate the 5th and 95th percentiles of the bootstrapped distribution (1000 bootstraps) of t-SPC events occurrence. The thick line denotes the mean. Black bars denote time bins in which error t-SPC occurrence is different between correct and error trials (cluster-based permutation test). **C** Cumulative distribution of the θ–α t-SPC onset and offset (see Fig. 2C and "Methods") in error and correct trials. **D** Comparison of the median θ–α t-SPC onset at the single-neuron level ($N = 20/46$ neurons in 13 participants). Black and magenta vertical dashed lines denote auditory cue (AC) and speech production (SP) windows. **E** Comparison of the median θ–α t-SPC duration at the single-neuron level before ($N = 17/46$ neurons in 13 participants) and after ($N = 8/46$ neurons) speech duration. Note that in (**D**, **E**) we only included neurons with significant θ–α t-SPC events in both accurate and error conditions. Two-sided permutation t-tests were used to compare t-SPC onset and duration within neurons.

of t-SPC events across these firing rate categories. Neurons with mixed firing rates (~0.31 s) and decreased firing rates (~0.31 s) had longer t-SPC events, with median centroid frequencies of 18 Hz and 20 Hz, respectively (Fig. S15B, C and Source Data). Both the group-level SPC maps and t-SPC event analyses revealed that only neurons with decreased firing rates significantly contributed to θ t-SPC events during speech production (Fig. S16A–C). In contrast, neurons with either decreased or increased firing rates exhibited similar profiles for α t-SPC events. θ–α t-SPC events occurred less frequently during auditory cue presentation and more frequently during speech production. Notably, neurons with increased or mixed firing rate modulation predominantly contributed to the aggregation of β t-SPC events during the rebound phase. Only neurons with mixed firing rate modulation showed a significant aggregation of β t-SPC events during the baseline period. In summary, neurons with decreasing firing rates exhibited t-SPC events dominated by θ and α rhythms, while neurons with increasing firing rates showed t-SPC events in the α–β range. This "band-pass" profile was particularly narrowband in β for neurons with mixed firing rate modulation. No distinct pattern of t-SPC coupling was observed in neurons without firing rate modulation at any level of analysis (Fig. S16A–C). These results suggest that the pattern of firing rate modulation in STN neurons—whether increasing, decreasing, or mixed—affects the frequency specificity of speech-related phase-of-firing coding.

We also examined the cortical distribution of SPC for each category of speech-related firing modulation. All categories exhibited a preference for coupling with PostCG, while neurons with decreased and mixed firing rates also showed significant coupling with the SMG. Additionally, neurons with decreased firing rates showed the highest coupling to the STG among all firing rate categories (Fig. S16D and Source Data). In summary, while firing rate modulation alone does not fully explain the dynamics of SPC, our results indicate that the pattern of firing rate change is strongly correlated with distinctive patterns of SPC, characterized by specific spectral, temporal, and anatomical features.

## Frequency dependence of preferred phase of coupling reflects cortico-subthalamic time delays

When an STN neuron locks to a cortical oscillation, we can extract the phase at which the locking occurs. The specific phase of the locking—such as the rising edge, peak, or trough—has been shown to encode key information, such as object identity in working memory[68,69] and contralateral versus ipsilateral movement in motor control[51,52]. After standardizing polarity across all t-SPC events (see "Methods"), we tested whether STN neurons consistently locked to a specific phase. We analyzed each frequency band independently and estimated the time-resolved population-level preferred phase (Fig. S17A). We found that t-SPC events in the α range, but not other frequency bands, are significantly coherent around the same phase of firing across pairs (108°, during the decay after the peak of oscillation) during speech production ($p < 0.05$, Hodges–Ajne test). In contrast, β t-SPC events are uniformly locked around the trough (−90°) of the oscillation

before and after the $\beta$ t-SPC rebound ($p < 0.05$, Hodges–Ajne test), consistent with previous studies that employed electroencephalography or low-density EEG strips[54].

Extracting phase across all t-SPC events helped us answer another important question in our investigation of cortical-subcortical coordination during speech: does cortex lead STN, or does STN lead cortex? We leveraged the population-level broadband t-SPC frequency distribution to estimate the directionality and magnitude of latency of information transfer, using t-SPC events in the 5–40 Hz range (see "Methods"). We plotted the relationship between the t-SPC phase and the frequency of the locking (Fig. S17B and Source Data), finding a clear linear relationship between phase and frequency. The linear relation suggests that STN spikes occur at a consistent time lag relative to the peak of the rhythmic ECoG activity[54]. The slope of the fit translated to a positive time lag with cortical activity leading STN of 40.92 ms ($R^2 = 0.76$, $p_{perm} < 0.001$) (Fig. S17B bottom). When we performed the same analysis over time, we found that this relationship was particularly consistent during the $\beta$ t-SPC rebound (39.81 ms, black dots in Fig. S17B bottom). Interestingly, STN spiking led to ECoG activity during speech production (−32 ms), while $\theta$–$\alpha$ t-SPC events were more frequent than $\beta$ t-SPC events (green curve in Fig. S17B bottom). These results suggest that $\theta$–$\alpha$ t-SPC and $\beta$ t-SPC observed here may reflect separate information flows between STN and cortex.

### Control analyses

To ensure the robustness and validity of our SPC estimates, we conducted a comprehensive set of control analyses. First, as our work heavily relies on the concept of t-SPC events, we performed simulations to assess the ability of our pipeline to reliably identify genuine SPC events under a range of conditions relevant to our dataset (Fig. S18 and Supplementary Text for details). The simulations systematically varied parameters related to task design (e.g., number of trials, variability in intra-participant behavioral events), neural activity (e.g., true SPC strength, SPC duration, baseline firing rate, firing rate modulation), and controllable signal conditioning settings (e.g., sampling rate, number of anchor points, target window width). Our results demonstrate that the variable-width procedure combined with a cluster-based permutation test effectively identifies non-spurious t-SPC events. Additionally, this approach achieves high accuracy in estimating the timing and duration of these events across a wide range of conditions. However, we found that the pipeline performance is particularly sensitive to the sampling rate, requiring sampling rates of at least 1 kHz, underscoring the importance of careful parameter selection during data acquisition and preprocessing. Second, we examined the impact of removing the event-related component of the ECoG signal before the computation of the SPC metric. After eliminating this component, the profile of the SPC maps remained largely unchanged, even at lower frequencies (Fig. S19A, B). Indeed, we found no evidence of phase reset of oscillations at the onset of auditory cue and speech production (Fig. S19C). These findings suggest that the event-locked components (trial-averaged speech-locked signals) did not significantly influence the observed SPC pattern. Third, we assessed the influence of periods with high (top 10th percentile) or low (low 10th percentile) oscillation amplitude. When excluding these specific periods, the results remained comparable (Fig. S20). We further explored the influence of power magnitude on SPC by examining changes in power during speech production in ECoG contacts, with a specific focus on the $\theta$–$\alpha$ bands in relation to SPC (Fig. S21). We found that, unlike the $\beta$ band, where we observed both $\beta$ power suppression and $\beta$-SPC suppression during speech production, the $\theta$–$\alpha$ power and $\theta$–$\alpha$ SPC exhibited distinct patterns. Specifically, $\theta$–$\alpha$ power was suppressed during speech production, while $\theta$–$\alpha$ SPC increased. This indicates that the increase in $\theta$–$\alpha$ SPC during speech cannot be attributed solely to an increase in the overall amplitude of $\theta$–$\alpha$ oscillations. Next, we explored whether ECoG contacts with significant SPC differed from those without SPC in terms of task-related power modulation. We found that ECoG contacts in the SMG, PreCG, and PostCG regions with significant SPC were more responsive to the task, showing greater suppression of low-frequency oscillations and enhanced high $\gamma$ activity during speech production (Fig. S21). These control analyses reinforce the significance of the observed SPC patterns.

## Discussion

Using simultaneous recordings from the perisylvian cortex and STN while Parkinsonian participants performed a syllable repetition task in the operating room, we discovered novel aspects of the neural coding of speech production that inform general principles of cortical-basal ganglia network information transfer. We found that STN neurons phase-locked to cortical oscillations in transient (~268 ms long) events. Any given neuron tended to lock to only one cortical oscillation frequency band and the type of firing rate modulation was predictive of the frequency of phase-of-firing. We identified one STN population that locked in the $\theta$–$\alpha$ range, and another that locked in the $\beta$ range. These events showed differential patterns across cortical regions and across auditory perception and speech production epochs of the task. $\beta$ t-SPC events clustered over ventral SMC and were prominent just after speech offset. Meanwhile, $\theta$–$\alpha$ t-SPC events clustered over SMG and STG and were prominent during speech. In exploring the relationship between t-SPC events and phonological speech production errors, we found that participants produced more errors in trials with delayed $\theta$–$\alpha$ t-SPC events.

### Principles of cortico-basal ganglia network interactions

Our results align with the notion of the cortico-basal ganglia thalamic loop subserving temporal integration for modulating motor control[70–72], consistent with previous evidence of SPC between STN neurons and cortical field potentials during limb movement[52–54,67,73]. Similarly, cortical oscillations manifest as transient bursts, whose onset is preceded by an increase of SPC with the STN[74]. These transient periods might represent "open windows" for effective communication between STN and cortex. The duration of these windows may be constrained by a slower subcortical neural timescale and requirements of a given motor instantiation[50,74].

Consistent with findings from limb movement studies[52,54,67], our results demonstrate that STN neurons preferentially couple with $\beta$ oscillations across broad cortical regions. This SPC was generally suppressed during task execution, with a rebound observed following speech termination. In contrast, we did not detect an increase in $\gamma$ coupling during speech preparation, a phenomenon previously reported between the STN and PreCG during movement preparation[51,75,76], where it has been interpreted as a modulatory signal associated with reaction time and movement facilitation. Cortical oscillations lead to phase overfiring activity in the STN, especially in the $\beta$ range post-speech. We found a delay of ~41 ms aligning closely with other reports[52,54]. Notably, the STN led the cortex exclusively when SPC in $\theta$ and $\alpha$ was more pronounced during speech production. While this delay does not necessarily signify synaptic transmission delay, it is consistent with the transmission of information through the cortico-basal ganglia loop. In light of our recent findings suggesting the presence of monosynaptic connections between non-motor regions of the cortex (sensorimotor and auditory areas) and the STN[48], it is not out of the question that the SPC we report here is anatomically rooted in the hyperdirect pathway.

The STN SPC found in this study complements previous studies describing $\gamma$ amplitude changes in the STG and SMG[56,77] and STN single-unit correlates in speech[41,45,47,49], reinforcing the importance of the STN as a hub that processes multimodal cortical information[78]. Prior studies have reported that STN neurons encode phonetic characteristics during speech production[47] and that STG lexical-encoding $\gamma$ signals are projected into the STN prior to speech production[49]. A

recent study showed that changes in functional connectivity between STN and language regions predicted the downstream effect of dopaminergic medication on speech-related cognitive performance[79]. Coming from a different recording modality and different measures of connectivity, our results bolster the finding that STN is involved in speech circuitry. Future work may investigate how much articulatory and acoustic information is encoded in SPC specifically.

At the neuron level, changes in SPC during speech production were not correlated with speech-related changes in the instantaneous firing rate or the baseline instantaneous firing rate. Other studies have reported similar decoupling between SPC strength and firing rate in STN neurons during movement[52,53]. STN neurons were preferentially coupled to a single frequency band which was significantly explained by the pattern of firing rate modulation in STN neurons. STN neurons with decreasing firing rates exclusively drove $\theta$ SPC. In a previous study[41], we found these neurons to be temporally locked to the onset of the auditory cue. Conversely, neurons with increasing firing rates displayed SPC in $\beta$, which was notably narrowband around 17 Hz in neurons with mixed firing rate dynamics. The phenomenon of SPC in response to behavioral state transitions and activity shifts remains relatively underexplored. Broicher and colleagues[80] utilized dynamic-clamp experiments to replicate in vivo-like conditions in hippocampal pyramidal neurons, showing that SPC frequency profiles are modulated by conductance states and input firing rates. Specifically, neurons in low-conductance states with reduced firing rates exhibited low-pass coupling, whereas neurons in high-conductance states with elevated firing rates displayed band-pass coupling. These differences can be attributed to mechanisms such as spike rate adaptation, which modulates the input-output gain (current-voltage relationship) and functions as a high-pass filter[81], as well as frequency resonance intrinsic to the spike-generation process. Our findings suggest that similar mechanisms underlie SPC dynamics in the STN. These results underscore the dynamic nature of phase-of-firing coding within the STN, driven by a complex interplay of neural network states and the intrinsic adaptive properties of individual neurons during speech-related tasks. We also found that the pronounced overall reduction of $\beta$ SPC observed at the population level during speech production did not reflect a uniform reduction of SPC at the single-unit level. A small subset of neurons (8%) increased their $\beta$ SPC during speech production, suggesting a partially maintained $\beta$ SPC and the presence of a distinct functional $\beta$ SPC network mainly rooted in the SCG. This aligns with two other studies that reported similar subpopulations of STN neurons, which increased their $\beta$ SPC during motor activity[52,54]. The functional relevance of this partially maintained $\beta$ SPC during speech production remains uncertain. Our findings overall underscore the notion that information can traverse the cortico-basal ganglia loop either through changes in the firing rate activity or spike timing. The presence of SPC between STN neurons and narrowband cortical oscillations does not imply that STN neurons generate and resonate coherent rhythms with the cortex[82]. For example, neurons displaying significant $\theta$ SPC with the STG do not necessarily oscillate in the $\theta$-rhythm at the population level. Therefore, our STN SPC topographies during speech production would not necessarily align with STN power-based topographies based on LFPs recorded at rest[64,83].

## $\theta$–$\alpha$ spike-phase coupling with SMG-pSTG

We uncovered a neural correlate of speech errors in our syllable repetition task: delayed, lower $\theta$–$\alpha$ t-SPC between STN and SMG-pSTG (posterior STG) (Fig. 5). Here we discuss two possible interpretations of how the $\theta$–$\alpha$ SPC differences relate to the speech errors. One possibility is that the errors were related to phonological working memory (PWM)[84]. We defined error trials as those in which at least one off-target phoneme was produced. Participants frequently made substitution errors in which the off-target phoneme was perceptually dissimilar to the target phoneme (Fig. S1); production errors were thus

unlikely the result of perceptual errors. Instead, the errors may be rooted in the failure of PWM to maintain the proper sequence in memory until speech production. Additionally, $\theta$–$\alpha$ SPC significant differences appear in the second half of the auditory window (Fig. 5B) when we would expect a reliance on PWM to maintain the syllable sequence. $\theta$–$\alpha$ SPC was observed predominantly in cortical regions that have long been implicated in PWM: the inferior parietal cortex and adjacent regions in pSTG (Fig. 3)[85,86]. $\theta$ SPC has been documented as a mechanism subserving working memory[87], lending further credibility to the PWM account of the speech errors.

Another possibility is that lower $\theta$–$\alpha$ SPC in error trials is related to auditory-motor integration, or the interface between auditory input and motor programs in the speech production system[1]. Because our auditory stimuli were phonotactically legal but meaningless, participants could not rely on lexical or semantic anchors to remember the verbal sequences. Participants would instead have to rely heavily on the "dorsal stream" of auditory processing in speech in the dual-stream model of speech processing[88]. The dorsal stream translates from sensory information to a motor encoding. The neurobiological cornerstone of the dorsal stream is situated just adjacent to the SMG-pSTG complex we identified in this study, in an area referred to as "Spt" (Sylvian-parieto-temporal). Spt, at the parieto-temporal boundary in and around the posterior Sylvian fissure, has been extensively studied for its sensorimotor properties[89–92]. Spt is critical for auditory repetition as it is hypothesized to compute a "coordinate transform" from auditory to motor space[1]. Lesions to this area can cause conduction aphasia—the selective deficit of verbatim repetition, despite fluent spontaneous speech and intact language comprehension[93,94]. Here, we find evidence that Spt and adjacent regions in the posterior perisylvian cortex might achieve this well-established auditory-motor interfacing by recruiting the BG, and specifically by leveraging $\theta$–$\alpha$ coupling with STN.

## Models of speech production

The cortical task-activated speech regions in this study—SCG, PostCG, PreCG, SMG, and pSTG—are key parts of the DIVA[95,96], state-feedback control[97], and hierarchical state-feedback[1] accounts of speech production. However, it is challenging to map our results directly onto these models because they (1) are largely activation-based and thus agnostic to electrophysiological mechanisms like LFP-spike inter-areal interaction and (2) focus on single-word production rather than speech sequencing, (3) do not detail different basal ganglia nodes like STN.

We briefly address the gradient order DIVA (GODIVA) model here because it concerns *speech planning* mechanisms[96,98], which is informative for the $\theta$–$\alpha$ SPC differences we observed >1 s before speech onset in accurate versus error trials. GODIVA posits a phonological content buffer for upcoming speech sounds. The buffer maintains multiple speech sounds in parallel, releasing them serially at the appropriate time. Our results highlight the role of the pSTG-SMG in the buffering process, while GODIVA posits that the buffer is subserved by areas in and around the posterior inferior frontal gyrus. Two possible explanations for this apparent discrepancy are as follows. First, GODIVA is largely grounded in evidence from activation-based studies, while our t-SPC metric is a measure of connectivity. Although they are often closely linked, activation and connectivity are separate mechanisms that can reveal different patterns of neural coding. Electrodes in the inferior frontal gyrus were active in the high-gamma range during both the speech planning and production window—but did not communicate with STN via t-SPC events. Second, our verbatim repetition syllable task may require greater reliance on auditory-to-motor coordinate transform than many of the orthographically cued tasks which informed GODIVA. The nature of the phonological processing required in this auditorily-cued task design may shift the phonological processing load from inferior frontal regions to the posterior perisylvian regions (pSTG-SMG) highlighted in this study.

Future computational models of speech may be able to work across levels of abstraction to maintain tractability but also consider mechanistic descriptions of brain interactions. At a minimum, our data and results inform future computational models of speech production that integrate the basal ganglia.

## Clinical implications: STN-DBS and transcranial magnetic stimulation

Beyond expanding theoretical frameworks, our results may have important implications for clinical therapies. Although many Parkinsonian motor symptoms can often be satisfactorily controlled by STN-DBS, stimulation-induced effects on the speech-motor system can be heterogeneous[99]. How can stimulating the same target nucleus consistently ameliorate some Parkinsonian symptoms yet have mixed and variable effects on the speech-motor system? Our results align with the notion that variability in DBS lead placement can explain most of the reported variance of outcomes in the literature[63]. Relative to the optimal therapeutic target defined by Caire et al.[100] ($x = -12.6$ mm, $y = -13.4$ mm, $z = -5.9$ mm), the spatial centroid of our speech-related STN SPC is at least 2.5 mm distant and overall located more posterior and ventral (Table S4). This aligns with studies that found detrimental effects on speech outcomes when stimulating more posteriorly[101–104] and ventrally[105]. However, all these studies simply compare the speech outcomes between DBS ON and DBS OFF conditions without considering the stimulation amplitude and the spread of the stimulation volume toward neighboring regions. Hypotheses for future investigation include stimulating the STN in areas of SPC density peaks to test for altered integration of sensorimotor and auditory signals. Non-invasive neuromodulation techniques, like transcranial magnetic stimulation (TMS), have been evaluated as therapies to alleviate symptoms in Parkinson's disease[106–108]. Our results could inform TMS studies targeting speech symptoms. Studies have demonstrated an improvement in hypokinetic dysarthric symptoms by stimulating around the pSTG-SMG complex implicated in this study[109,110]. Further research is warranted to what degree TMS may alleviate more motoric versus more cognitive aspects of PD speech symptoms[111].

## Limitations

Our findings should be interpreted in the light of several limitations. First, our intracranial recordings are from patients with PD. Caution must be exercised when interpretations of human neurophysiology are drawn from observations collected in a pathological state. Specifically, differences in the STN baseline firing rate[112], abnormal subcortical beta oscillations[62,113–115], and loss of movement specificity[116] that characterize the Parkinsonian state may confound the distinction of whether our observations generalize to speech in individuals without PD. There are no opportunities to record from human basal ganglia nuclei that are not in a pathological state; however, future research can clarify which aspects of our results generalize to non-pathological basal ganglia. Second, because recording locations were clinically determined, we had uneven coverage of the STN and of the lateral speech-motor cortex. Most microelectrode trajectories traversed the dorsolateral part of the STN, the clinical target for PD DBS[63]. Hence, sampling of the ventromedial region of the STN is limited. ECoG coverage also varied across participants and spanned a limited region of the cortical surface. We cannot rule out any other interaction of the STN with other cortical regions. Lastly, we are unable to draw any conclusions based on our data regarding speech specificity because patients completed only the speech task in the operating room. We therefore can't weigh in on the differences between cortico-basal ganglia interaction in speech versus limb motor control. Given the differential patterning of speech and non-speech-motor control in PD and treatments for PD[117–119], future research may explore and compare different movement modalities.

In summary, we discovered evidence that STN neurons are linked to the phase of the cortical oscillations during speech. These insights

provide a deeper understanding of how different types of information are processed in basal ganglia-cortical loops and have significant implications for understanding the role of the human basal ganglia in sensorimotor integration for speech and other behaviors[120].

## Methods

### Participants

Electrophysiological signals were recorded intraoperatively from 24 participants (20 males and 4 females, age: 65.4 ± 7.1 years; mean ± SD) with Parkinson's Disease undergoing awake stereotactic neurosurgery for implantation of DBS electrodes in the STN (Table S1 for clinical details). Participants performed up to 4 sessions of the task, leading to a total of 64 sessions, after overnight dopaminergic medication withdrawal. All procedures were approved by the University of Pittsburgh Institutional Review Board (IRB Protocol #PRO13110420) and all participants provided informed consent to participate in the study.

### Method details

**Speech production task**. Participants were tasked to intraoperatively repeat aloud CV syllable triplets. The stimuli were presented auditorily via earphones (Etymotic ER-4 with ER38-14F Foam Eartips) and were delivered at either low (-50 dB SPL) or high (-70 dB SPL) volume using BCI2000 as stimulus presentation software. The absolute intensity was tailored to each participant's comfort level, keeping fixed the difference between high and low conditions at 25 dB SPL. The experiment utilized a set of phonemes consisting of four consonants (/v/, /t/, /s/, /g/) with different manners of articulation and three cardinal vowels (/i/, /a/, /u/) with distinctive acoustic properties. We created a unique set of 120 triplets of CV syllables, forbidding CV repetition within the triplet and balancing syllables and phoneme occurrence, and CV position within the triplet across a run of the task. The audio produced by the participant was recorded with a PRM1 Microphone (PreSonus Audio Electronics Inc., Baton Rouge, LA, USA) at 96 kHz using the Zoom-H6 portable audio recorder (Zoom Corp., Hauppauge, NY, USA).

**Neural recordings**. As part of the standard DBS clinical procedure, functional mapping of the STN was performed using microelectrode recordings (MER) acquired with the Neuro-Omega recording system (Alpha-Omega Engineering, Nof HaGalil, Israel) using parylene insulated tungsten microelectrodes (25 μm in diameter, 100 μm in length). The microelectrodes were oriented using three trajectories (Central, Posterior, and Medial) of a standard cross-shaped Ben-Gun array with a 2 mm center-to-center shaping. MER signals were referenced to the metal screw holding one of the guide cannulas used to carry the microelectrodes and recorded at 44 KHz. Prior to STN mapping, participants were temporarily implanted with two high-density subdural electrocorticography (ECoG) strips consisting of 54 or 63 contacts, respectively (PMT Contact). These strips were placed through the standard burr hole, targeting the left ventral sensorimotor cortex, and left inferior frontal gyrus. Signals from ECoG contacts were referenced to a sterile stainless-steel subdermal needle electrode placed on the scalp and acquired at 30 kHz with a Grapevine Neural Interface Processor equipped with Micro2 Front Ends (Ripple LLC, Salt Lake City, UT, USA).

**Electrode localization**. We localized the ECoG strips and DBS leads using well-established pipelines in the literature. For ECoG strips, contact locations were determined using the Randazzo localization method[121] that utilizes a preoperative T1 weighted MRI scan, an intraoperative fluoroscopy, and a postoperative CT scan (github.com/Brain-Modulation-Lab/ECoG_localization). CT and MRI were coregistered using SPM and then rendered into a 3D skull and brain using Osirix (www.osirix-viewer.com) and Freesurfer (https://surfer.nmr.mgh.harvard.edu) software. The position of the frame's tips on the skull and the implanted DBS leads were used as fiducial markers, which were coregistered and aligned with the projection observed in the

fluoroscopy. The position of the contacts in the ECoG strip was manually marked on the fluoroscopy image and then projected to the convex hull of the cortical surface. To extract the native coordinates of individual contacts, we leveraged the known layout of the ECoG strip. All coordinates were then transformed into the ICBM MNI152 Non-Linear Asymmetric 2009b space, employing the Symmetric Diffeomorphism algorithm implemented in Advanced Normalization Tools (ATNs). For DBS lead reconstruction, we used the Lead-DBS localization pipeline[122]. Briefly, the process involved coregistering the MRI and CT scans, and manually identifying the position of individual contacts based on the CT artifact, constrained by the geometry of the DBS lead used. The coordinates for the leads in each participant's native space were rendered after this process. Custom Matlab scripts (github.com/Brain-Modulation-Lab/Lead_MER) were then used to calculate the position of the micro- and macro-recordings from the functional mapping based on the position of the lead, the known depth, and tract along which the lead was implanted in each hemisphere. Anatomical labels were assigned to each contact based on the Destrieux atlas[123] for cortical contacts, and the DISTAL atlas[124] for subcortical contacts.

### Quantification and statistical analysis

**Phonetic coding.** To extract phoneme characteristics from the produced speech signals such as onset and offset times, IPA code, and accuracy, we employed a custom Matlab GUI (github.com/Brain-Modulation-Lab/SpeechCodingApp). Phonetic coding of each produced phoneme was performed by a trained team of speech pathology students using Praat (https://www.fon.hum.uva.nl/praat/). Discrepancies between the produced phoneme and the target phoneme were labeled as phonetic errors. We identified three types of errors: consonant substitution (e.g., /g/ produced as /v/), vowel substitution (e.g., /u/ produced as /i/), and phonemic omission (e.g., /su/ /ti/ /ga/ produced as /su/ /i/ /ga/). The same trained team of speech pathology students also evaluated articulation disorders and voice quality at the single phoneme level. Please refer to the Supplementary Text for details.

**Behavioral events.** For each trial, we defined four different behavioral epochs: baseline epoch as a 500 ms time window between −550 ms and −50 ms prior to the auditory cue onset, auditory cue presentation as the window during which syllable triplets were presented auditorily (~1.5 s duration), speech production as the variable time window during which participants repeated aloud the syllable triplet (~1.6 s duration on average) and post-speech as the 500 ms time window after the speech offset.

**Electrophysiological data alignment.** To temporally align the continuous recordings from the Ripple, Neuro-Omega, and Zoom-H6 systems, we employed a linear time-warping algorithm based on the stimulus and produced audio channels. We defined the Ripple files as the "leader" time and independently aligned the Neuro-Omega and Zoom-H6 recordings to it. To this end, we first coarsely align the files from different sources manually (no warping) by marking easily identifiable landmarks on each file (i.e., the beginning of the first trial). We then split the files into chunks of around 100 s and performed a staged optimization procedure, independently in each chunk, to find the precise alignment and warping factor. In the first stage, the envelopes of the corresponding audio signals from the two files were calculated at 100 Hz, by calculating the maximal absolute value in 10 ms bins. We next found the delay ($j$) between the envelopes (Eq. (1)), which maximized their cross-correlation ($r_j$) and adjusted the "follower" channel accordingly:

$$r_j\left(\vec{x}, \vec{y}\right) = \frac{1}{n}\sum_i \frac{\left(x_i - \bar{x}\right)}{\sigma_x} \frac{\left(y_{i+j} - \bar{y}\right)}{\sigma_y} \qquad (1)$$

Next, we applied the following time-warping algorithm: we calculated a smooth interpolation function $f(t)$, such that $f(t_i) = y_i$ for all time points $t_i$, where $y_i$ is the corresponding follower signal value. We defined the time-warping function $\omega(t) = t_0 + t_p + (t - t_p)\gamma$, where $t_p$ is the "pivot time" defined as the midpoint of the leader chunk to synchronize, $\gamma$ is the time-warping factor and $t_0$ a small "time translation" correction. Using this function, we calculated the time-warped follower signal $\vec{s}$ such that $s_i = f(\omega(t_i))$. We then optimized the time-warping parameters to maximize the correlation between $\vec{s}$ and the leader signal pattern $\vec{p}$, that is, $\omega = \text{argmax} r_0(\vec{p}, \vec{s})$. We did the optimization using *fminsearch* in Matlab, by minimizing the cost function (Eq. (2)), as follows:

$$C(t_0, \gamma) = -\frac{\vec{p} \cdot \vec{s}}{\sqrt{\vec{s} \cdot \vec{s} \times \vec{p} \cdot \vec{p}}} + k_0 t_0^2 + k_1(\gamma - 1)^4 \qquad (2)$$

where the regularization parameter $k_0$ was set to 0.0003 and $k_1$ to 0.001. To achieve sub-millisecond precision, a second stage was done using the same synchronization algorithm on the raw audio signal, low-pass filtered to 5 kHz and resampled to 10 kHz for computational efficiency. Note that the fitted warp factor $\gamma$ typically differed from unity in one part in $10^5$, meaning that the correction amounted to 1 ms every 100 s, and was very consistent within-subject and file type. The tolerance of the synchronization was defined as the maximal mismatch in synchronization between adjacent 100 s chunks calculated for each participant. Sub-millisecond synchronization precision was achieved. Note that a 1 ms mismatch only represents a 3% change in phase in the high $\beta$ range and a 10% change for high $\gamma$.

**Electrophysiological data preprocessing.** ECoG preprocessing was performed using custom code based on the Fieldtrip toolbox[125] implemented in Matlab, available at (github.com/Brain-Modulation-Lab/bml). Data was low-pass filtered at 250 Hz using a 4th-order Butterworth filter, downsampled to 1 KHz, and stored as a Fieldtrip object. Metadata such as descriptions of each session, phonetic coding, event times, and electrode locations were stored in annotation tables. We applied a 5th-order high-pass Butterworth filter at 1 Hz to remove drifts and low-frequency components. Segments with conspicuous high-power artifacts were identified using an automatic data cleaning procedure[126], based on a power-based threshold. Specifically, we extracted power at frequencies in different canonical bands (3 Hz for $\delta$, 6 Hz for $\theta$, 10 Hz for $\alpha$, 21 Hz for $\beta$, 45 Hz for $\gamma_L$, and 160 Hz for $\gamma_H$) by convolving ECoG signals with a 9-cycles Morlet wavelet. A time bin was classified as artifactual if its log-transformed power in any band exceeded a threshold defined as the mean ± 2.5 std (~10-fold higher than the mean). Trials with time segments flagged as artifactual were discarded and channels with more than 30% of artifactual time bins were not included in the analysis.

**Spike sorting.** Spike sorting was performed using Plexon (https://plexon.com/products/offline-sorter/)[41]. We used a 4th-order Butterworth high-pass filter with a cut-off frequency at 200 Hz and set a manual threshold to extract putative waveforms. Single units were discriminated and graded based on factors such as cluster isolation in the principal component, the spike sorting's stability over time, a refractory period of at least 3 ms in the inter-spike interval distribution, and the shape of the waveform.

**Instantaneous firing rate.** To analyze changes in spike rate activity, we followed the procedure described in ref. 41. We sought elevated and reduced firing activity by computing the instantaneous firing rate (gaussian kernel, $\sigma = 25$ ms) and the inter-spike interval (smoothing window 25 ms), which scales with the reciprocal of the instantaneous firing rate, respectively. We aligned these quantities with speech

production onset and analyzed time bins from auditory cue onset through speech production offset. A neuron was considered as a Decreasing firing rate neuron if the inter-spike interval exceeded for at least 100 ms the upper 5% of a normal distribution with mean and standard deviation calculated during the baseline period. Similarly, a neuron was considered as an Increasing firing rate neuron if the instantaneous firing rate exceeded for at least 100 ms the upper 5% of a normal distribution with mean and standard deviation calculated during the baseline period. Neurons that exhibit both modulations were named as Mixed firing rate modulation neurons, while neurons that did not exhibit any significant speech-related firing changes were labeled as No firing rate modulation neurons. For a comprehensive description of the firing rate modulation, please refer to Lipski et al.[42].

**Time-frequency decomposition.** Time-varying power and phase were obtained by applying the Hilbert Transform to the band-pass filtered ECoG signal. The signal was band-pass filtered using a 4th-order Butterworth filter, with the following frequency ranges: 5–8 Hz for $\theta$, 8–12 Hz for $\alpha$, 12–20 Hz for low $\beta$, 20–30 Hz for high $\beta$, and center frequencies ranging from 40 to 150 Hz with a bin width of 10 Hz, incrementing by 10 Hz for $\gamma$.

**Spike-phase coupling implementation.** To calculate SPC, we considered each possible pair of neurons and ECoG signals that were synchronously recorded. We enforced the following criterion for determining the eligibility of pairs ($N = 19{,}755$) for subsequent analysis: a minimum of 10 trials with a stable firing rate and clean ECoG signal. The strength of the SPC was quantified by the phase-locking value (PLV, Eq. (3)), which represents the magnitude of the circular average of unit complex vectors corresponding to the ECoG phase at the time of each spike $\varphi_t$, as follows:

$$\text{PLV} = \left| \frac{\sum_{t=1}^{N} e^{j*\varphi_t}}{N} \right| \qquad (3)$$

where $N$ is the number of spikes included in the window. PLV is bounded between 0 and 1, indicating lack or perfect SPC, respectively. Importantly, PLV is inflated toward 1 when $N$ is low. When $N$ is sufficiently large ($N > 50$), the pairwise-phase consistency (PPC, Eq. (4)) yields an unbiased estimator of SPC[127], as follows:

$$\text{PPC} = \frac{N}{N-1}\left(\text{PLV}^2 - \frac{1}{N}\right) \qquad (4)$$

In the absence of SPC, PPC is expected to be centered around zero, including negative value, when $N$ is finite. As N increases towards infinity, the PPC tends to $PLV^2$. Although different methods have been proposed to estimate SPC, we opted to use PLV (and its extension PPC) because it is one of the most established methods and its limitations have been extensively studied in literature[127–129]. To ensure that changes in SPC depicted genuine and comparable neural correlates, methodological considerations must be discussed. First, the presence of speech-related fluctuations in the instantaneous firing rate poses a challenge in selecting a fixed window size for calculating the phase-locking value or PPC over time. This is because variable $N$ can result in uncontrollable and variables biases. Furthermore, low $N$ can lead to noisy estimates of PPC. Second, intra- and inter-participant variability in speech production onset and duration makes the event-locked analysis less accurate for the alignment of data around all key task events and not just for one single event (the one used for locking). To overcome all these limitations, we employed a variable-window width SPC estimation procedure developed by Fischer and colleagues[51]. First, we defined five intervals: from 0.75 before the auditory cue to auditory cue onset, from auditory cue onset to offset, from auditory cue offset to speech production onset, from speech production onset to offset,

and from speech production offset to 0.75 s after. Second, we subdivided these intervals into 21 equidistant anchor points, resulting in 101 anchor points for each trial to ensure 10 anchor points in a 0.5 s window on average (50 ms time resolution). Third, we scaled the width of the window centered at each anchor point such that the sum of spikes ($N$) across all trials would match a target number as closely as possible. The target number was defined as the average number of spikes in a window of 0.15 s, but always greater than 25 to avoid fewer representative samples. This process allowed to enlarge/shrink the computational window during reduced/increased firing rate periods, ensuring that $N$ remained constant over time. Note that we allowed variable number of spikes across participants to reduce variability in the window width. Finally, each window was placed symmetrically around each anchor point, and we subsequently calculated the PLV metric and applied the PPC correction. The resulting 19755 SPC maps (PPC values of size 16 frequency bins x 101 time points) were smoothed using a time-frequency window ([2, 2] size) and rescaled to the average duration of the event intervals. These maps were then event-locked and averaged across pairs and participants.

**Phase polarity standardization.** When computing the PLV or PPC, information about the preferred phase is not retained. To identify the preferred phase at which spikes are bundled, we calculated the circular mean using the CircStat toolbox[130]. However, it is important to exercise caution when comparing preferred phases across recordings due to the relative orientation between neural sources and electrodes (i.e., source mixing) and the use of different re-referencing schemas, as these factors can obscure the interpretation of the instantaneous absolute phase[131]. For instance, by applying the bipolar schema, the order of subtraction between two electrodes can flip throughs to peaks and peaks to throughs. To ensure that phases were meaningfully computed across recordings, we applied an automatized polarity-standardization procedure[51]. Specifically, we flipped phases (+$\pi$) such that $\gamma$ peaks in the 60–80 Hz range consistently coincided with increases in the local high-frequency activity, which served as a polarity-invariant proxy of background unit activity[132,133]. We computed this proxy by high-pass filtering the ECoG signal at 300 Hz, full-wave rectifying it, and low-pass filtering it with a cut-off of 100 Hz. Flipping procedure was required in 7111/19755 pairs (36%).

**Spike-phase coupling events.** To further correct the SPC maps for any residual bias and identify genuine increases in SPC, we converted PPC values into $z$-scores relative to a permutation distribution and performed a cluster-based permutation test[134]. We built the permutation distribution by shuffling the trial association between STN spikes and EcoG phases 500 times. We paired spike timings from the $i$th trial with EcoG phases from the $j$th trial (where $i \neq j$). Importantly, to be conservative and preserve the natural appearance of clusters, we applied the same randomization across time-frequency bins. Suprathreshold clusters ($p < 0.05$) were identified in both the original SPC map and in each permutation SPC map by computing the $z$-score relative to the permutation distribution. If the absolute sum of the $z$-scores within the original suprathreshold clusters exceeded the 95th percentile of the 500 largest absolute sums of $z$-scores from the permutation distribution, it was considered statistically significant. These significant clusters in the SPC map were referred to as transient-SPC events (t-SPC events). SPC maps that contained at least one t-SPC event were considered significant. Please refer to Supplementary Text and Fig. S18 for an in-silico validation of the t-SPC event identification.

**Spike-phase coupling event characteristics.** To fully characterize each t-SPC event, we defined a set of characteristics in the time, frequency, and phase domain. In the temporal domain, we calculated the onset and offset times, temporal duration, and the center of the event (i.e., the mean of the onset and offset). For the frequency domain, we

calculated the frequency centroid (Eq. (5)), as follows:

$$f_{t-SPC} = \frac{\sum_i^T \sum_j^F PPC(i,j) * j}{\sum_i^T \sum_j^F PPC(i,j)} \qquad (5)$$

where $I$ and $j$ are the $i$th time bin and $j$th frequency bin enclosed within the boundaries of the t-SPC event. In the phase domain, we calculated the circular mean of the t-SPC event phases.

**Time occurrence.** We aimed to quantify the temporal occurrence of t-SPC events, which we defined as the likelihood of observing at least one t-SPC event in a given STN neuron-ECoG contact pair during each time bin and in each frequency band (refer to Fig. S5). To achieve this, we transformed each t-SPC event into a binarized vector, where each time-point (at intervals of 5 ms) was labeled either as part of a t-SPC event and assigned a value of 1, or as part of a non-t-SPC period and assigned a value of 0. We then calculated the mean of these binarized vectors at each time bin, expressed as a percentage. Higher values of this quantity indicated a greater temporal aggregation (i.e., overlap) of t-SPC over time, whereas lower values indicated dispersion. To identify significant time windows of aggregation or dispersion, we employed a permutation test, adapting the approach used to calculate significant beta bursts overlap, as described in ref. 61. We generated a permutation distribution of the time occurrence due to chance by setting a variable break point in the 0 s of the binarized vectors (no slicing of t-SPC events), reversing the two segments, and joining them together. We repeated this process 500 times and extracted the permutation distribution over time. We considered t-SPC events to be significantly dispersed when the time occurrence fell below the 5th percentile of the permutation distribution and significantly aggregated when the time occurrence rose above the 95th percentile of the permutation distribution. By applying this method, we were able to rigorously determine changes in time occurrence even when the number of t-SPC events was low and to eliminate spurious trends of aggregation when the number of t-SPC events was high.

**Spatial density.** We quantified the spatial density of t-SPC events by calculating the ratio between the number of t-SPC events and pairs and expressing it as a percentage, both at the cortical and STN levels (refer to Fig. S5). To calculate the spatial density on the cortical surface, we used two region-of-interest-based methods. In the first method, we identified seven ROIs in the Destrieux atlas[123] that satisfied a minimum coverage criterion (>7 participants and >100 pairs, see Table S3 for details): PreCG, PostCG, SMG, STG, Middle frontal gyrus (MFG), and the orbital part of the inferior frontal gyrus (pars O.). We calculated the spatial density in each region of interest and determined whether t-SPC events were preferentially located in any region of the brain or whether they exhibited no spatial preference, both overall and within frequency band. To test for spatial preference, we created a null permutation distribution (spatial uniform distribution) by shuffling the spatial label of each t-SPC event 500 times. We then compared the spatial density of the original data to the 5th–95th percentiles of the spatial density permutation distribution. Regions with spatial density below the 5th percentile and above the 95th percentile were classified as having "low" or "high" spatial preference, respectively. In the second method, we created a cortical spatial density map by calculating the spatial density in a spherical region of interest with a radius of 2 mm centered around the ECoG recording locations in the MNI space. These maps were converted and displayed in SurfIce as nodes. For the STN domain, we built an STN spatial density map by locating spheres (1 mm radius) around the STN neuron locations. Again, we used SurfIce for visualization. For each spatial density map, we identified the peak in each frequency band. Additionally, we projected the STN neuron locations onto the three principal directions of the STN extracted from

the DISTAL atlas image[124], following the procedure as described in ref. 83. To preserve the physical meaning, i.e., distance in mm, of the principal component decomposition, we multiplied the principal component scores by the standard deviation of the MNI coordinates. The principal component coordinates (PC1: antero-posterior direction, PC2: dorso-ventral direction and PC3: medio-lateral direction) represents a more suitable reference of frame, as the STN is not fully spatially aligned with the MNI coordinates (Fig. S11A, B). Spatial density computation was repeated for each of the three principal axes using a size of 0.8 mm.

**Spike-phase coupling at the single-neuron level (t-SPC index).** To compare the SPC at different epochs of the task or frequency bands across neurons and to control for the effect of the firing rate (see "Control analysis"), we computed the ratio between the number of t-SPC events and pairs for each neuron and expressed it as a percentage across different task epochs and frequency bands (refer to Fig. S5). We termed this quantity as t-SPC index, and it was used for correlation analyses in which the statistical unit of observation was the single neuron. We also assessed the extent to which neurons preferentially couple to the same frequency band. We normalized the t-SPC index across frequency bands (total sum = 1) and defined the frequency specificity as one minus the entropy of the normalized distribution. With this definition, high (e.g., peaked distribution) and low specificity (e.g., uniform distribution) are mapped onto 1 and 0 values, respectively (see Fig. 2G).

**Spatial aggregation.** To extend and further corroborate our findings in the spatial domain, we also conducted a region-of-interest-free analysis, both at the cortical and STN levels. MNI and PC coordinates (and their centroid) of t-SPC events were compared across frequency bands using a permutation test. We then investigated whether t-SPC event locations (within each frequency band) were more spatially aggregated around their centroid than expected by chance (uniform distribution). To this end, we computed the average Euclidean distance between t-SPC events locations and their centroid[83], and compared against a null distribution of surrogate average Euclidean distances obtained by randomly sampling recording locations 500 times.

**Relationship between STN spike-phase-coupling topography and DBS anatomical STN targets.** To investigate the relationship between the frequency-specific STN topographies and optimal DBS target for motor symptom control in PD, we calculated the Euclidean distance between frequency-wise spatial centroids and the location of DBS contacts commonly used for therapeutic stimulation[63,100].

**Time delay analysis.** As STN neurons often lock to cortical signals within a narrow frequency range, power-based estimates of time delay between STN and cortex might be suboptimal[1]. We calculated time delays using the phase-based analysis, as described in ref. 54. First, we computed the mean preferred phase of units that were significantly locked in each frequency bin (5–30 Hz range). We then averaged these phases to obtain a grand average phase for each frequency bin. By analyzing the gradient of these phases, we determined whether the ECoG channel led (positive sign) or lagged (negative sign) relative to the STN neuron, and at what latency this occurred. To test the significance of the time delay, we repeated 500 times the computation using randomly selecting mean angles from each frequency bin. To obtain a p-value, we compared the correlation coefficient in the original data and the 5th–95th percentiles of the correlation coefficient permutation distribution. In Fig. S17B we calculated the relative time occurrence of t-SPC events in the $\theta-\alpha$ and $\beta$ range and it is quantified as a contrast $(A-B)/(A+B)$, where A represents the time occurrence of SPC in the theta/alpha frequency band (red and orange curves in

Fig. 2F) and B represents the time occurrence of SPC in the beta frequency band (yellow curve in Fig. 2F).

**Relationship between spike-phase coupling and speech behavior.** To examine the link between SPC and speech behavior, two sets of analyses were conducted on phonetic accuracy. We restricted this analysis only to the low-frequency range (4–40 Hz). For phonetic accuracy, trials were categorized into correct (100%) and error (<100%) groups. Only significant (from the main analysis) pairs with ≥20 trials in each condition were included (46 neurons, 827 pairs across 18 participants, see Table S5 for details). To balance the number of trials, we subsampled 20 trials in each condition and ran the SPC pipeline 50 times. SPC maps were subtracted and averaged across subsamples. We converted the PPC values into $z$-scores relative to a permutation distribution defined as the difference between the permuted values in the two conditions. Significance was evaluated using the same procedure as above (see "Spike-phase coupling events"). The significant clusters in the difference SPC map were referred to as t-SPC events signaling time-frequency bins in which the first condition was either higher or lower than the second one according to the sign of the $z$ value. For statistical comparison at the group level between the two conditions, we converted the $z$-scores to $t$ values and generated 500 permuted samples by randomly permuting the order of subtraction of the two SPC maps. $P$ values were estimated using the null distribution and corrected using again a cluster-based procedure.

**Control analysis.** To further ensure the reliability of the t-SPC events identified by our cluster-based permutation analysis, we conducted two control analyses. Firstly, we required that a t-SPC event contain at least two cycles of oscillation at the centroid frequency to be classified as reliable, thus ruling out brief and transitory noise-driven clusters. Secondly, we recognized that surrogate SPC maps generated during the permutation procedure may contain surrogate t-SPC events due to natural cluster tendency arising in small samples of random distribution, which can be mistakenly identified as non-random. To this end, we $z$-scored the surrogate PPC maps and conducted the same cluster-based permutation, defining surrogate t-SPC events as those that met the same criteria as the original t-SPC events. We then compared the number of observed t-SPC events to that of the surrogate t-SPC events. We also carried out several control analyses to rule out confounding factors that might have influenced the SPC changes we observed: differences in firing rates, differences in ECoG power, and a phase reset around speech production onset. Although the SPC pipeline is designed to remove any firing rate bias in the SPC estimation, we sought to investigate genuine firing rate effects by plotting firing rate changes against SPC changes across STN neurons. To ensure that phase estimates were not based on unreliable low amplitude oscillation (during $\beta$ suppression), we repeated the analysis and discarded instantaneous phase samples in which the instantaneous power fell below the 10th percentile. We also checked whether bouts of oscillatory power (during $\theta$ and $\gamma$ increase) biased the SPC estimation by discarding instantaneous phase samples in which the instantaneous power rose above the 90th percentile. We further explored the influence of power magnitude on SPC by examining changes in power during speech production in ECoG contacts, with a specific focus on the $\theta$–$\alpha$ bands in relation to SPC. We categorized ECoG contacts into six groups (No-SPC, SPC in any band, $\theta$–$\alpha$ SPC, $\beta$ SPC, $\gamma_L$ SPC, and $\gamma_H$ SPC) and compared the speech-locked power modulation with respect to the baseline across frequency bands ($\theta$, $\alpha$, $\beta$, $\gamma_L$, $\gamma_H$) and cortical ROIs. To ensure balanced comparisons, we included only conditions with at least 20 ECoG contacts and estimated the distribution of the mean by resampling 20 ECoG contacts 500 times. We examined the impact of phase resetting on brain oscillations, which can generate event-related activity. To this end, we run two complementary analyses. First, we aligned all the trials to the speech production onset, averaged the ECoG signals across trials to obtain evoked activity, and subtracted this component from individual trials before conducting the SPC analysis. Second, we quantified whether auditory cue or speech production onset reset the phase of the ECoG oscillations. We estimated the event-locked SPC the same way as the SPC, except that ECoG segments were aligned at the auditory cue and speech production onset. Each trial thus contributed one spike to the SPC computation.

**Statistical analysis.** We used the RainCloud library for the visualization of data distributions[135]. Kolmogorov-Smirnov test revealed that the normality assumption of the distribution was rarely satisfied. For this reason, we decided to apply a series of permutation tests (1000 permutations unless stated otherwise) throughout the manuscript whenever the definition of a null distribution was methodologically justified. An exception is represented by circular data (e.g., phases) that required the usage of the CircStat toolbox[130]. When multiple pairwise permutation tests were applied over different ROIs (e.g., Figs. 3A and S10) or frequency bands (e.g., Figs. S7 and S9A, B and Source Data), we controlled the family-wise error rate by applying the t-max correction[136], also referred to as joint correction. This correction works as follows[137]: on each permutation of the data, the test statistic is computed for each comparison and the most extreme value (either positive or negative) across comparisons is taken. Repeating this procedure multiple times produces a single, more-conservative permutation distribution, against which the actual test statistic is compared. All results were assessed at a statistical significance of $\alpha = 0.05$.

### Reporting summary
Further information on research design is available in the Nature Portfolio Reporting Summary linked to this article.

## Data availability
The data of this study is hosted in the Data Archive BRAIN Initiative (DABI, https://dabi.loni.usc.edu/dsi/1U01NS098969) and is available upon request. No participant-identifiable information will be disclosed. The datasets generated and/or analyzed and the statistical tests used during the current study are attached as Source Data files. Source data are provided with this paper.

## Code availability
Example code to reproduce the main results is published at Github (https://github.com/Brain-Modulation-Lab/code_SPC_ECoG_STN_Speech) and Zenodo[138] (https://doi.org/10.5281/zenodo.12610957).

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

## Acknowledgements

We would like to thank the research participants for their generous contribution of time and effort in the operating room and the additional experimenters who acquired and organized the data. This work was funded by the National Institute of Health (BRAIN Initiative), through grants U01NS098969, U01NS117836, and R01NS110424 to R.M.R. We extend our gratitude to Frank H. Guenther for the fruitful discussion on the modeling implications of this work.

## Author contributions

A.B. and W.J.L. wrote experimental code and performed experiments and recorded data. A.B., W.J.L., L.L.H., J.A.F., R.S.T., and R.M.R. designed experiment. R.M.R performed the surgery and supervised the project. P.F. helped to implement the SPC pipeline and contributed to the interpretation of the SPC results. C.N. wrote parts of the discussion and helped to create the 3D visualization of the SPC topography. L.B. wrote parts of the introduction and the discussion. M.V. analyzed data, prepared figures, and wrote the first draft of the manuscript. All authors discussed results at all stages of the project and revised the manuscript.

## Competing interests

The authors declare no competing interests.
