## [Transparent Peer Review file · Nature Communications]

Spike-phase coupling of subthalamic neurons to posterior perisylvian cortex predicts speech sound accuracy

Corresponding Author: Dr Matteo Vissani

Version 0:

Reviewer comments:

Reviewer #1

(Remarks to the Author)

In the reported study, the authors used high-density electrocorticography (ECoG) to record local field potentials across cortical areas known to participate in speech perception, planning, and production, with simultaneous recording of STN single-unit activity during DBS surgeries in 24 participants when they spoke syllable repetitions. The authors reported that individual STN neurons have transient (200ms) spike-phase coupling (SPC) events with multiple cortical regions. The spike timing of STN neurons was coordinated with the phase of theta-alpha oscillations in the posterior supramarginal and superior temporal gyrus during speech planning and production. Speech sound errors occurred when this STN-cortical interaction was delayed.

The results are very interesting and novel, the methodological procedures are robust. We (I myself and an early career research scientist in my lab) had few comments and questions, mainly in the results and methods sections. Here below are the comments and questions, in order of where they appear in the manuscript, now in order of importance:

1- Results (line 81-82): Figure S2 is not complete.

2- The authors mentioned in Line 86 that, 'premotor and postcentral gyrus electrodes showed –suppression and –elevation preceding the speech onset and during speech production'. In the task, speech production is followed by the auditory cue, which was associated with beta reduction. Whether the changes shown preceding the speech production is actually the response to the auditory cue?

3- The authors showed that neurons' instantaneous firing rates during the speech task was heterogeneous both within and across recording session (also shown in Figure 1-D). – We are wondering if there is any spatial distribution of each neuron category: Are the increasing/decreasing IFR neurons clustered in sweet spots?

4- Line 119 – Only 11% (2148/19755) of the time-frequency maps displayed significant SPC. What are the neuron types for those which displayed significant SPC: is there any consistent trend in their firing rate in response to speech production: increase or reduce, or are they very heterogeneous? Similar to the LFP channels showing SPC: are they showing most speech related modulation, such as more beta reduction and theta/alpha increase during speech?

5- Line 132, 'single-pair SPC maps showed that STN neurons entrain cortical LFP phases during transient episodes (Figure 2B), which we termed t-SPC' – The word 'entrain' implies a strong causal relationship, I'm wondering if this is what the authors really want to mean, but I don't think there is any proof for this.

6- Results (line 140): The variable-window width procedure to estimate SPC maps, as described in the methods, alters the notion of accurate timing, given that each time bin can have a different window length. Did the author consider this when calculating the median duration of the events? In addition, as permutation is used for statistics, are the 'duration' and the number of significant SPC events sensitive to the total number of points in the 2D time-frequency map, the resolution in time and frequency, as well as any smoothing along time and along frequency in the 2D mat? How did the authors decide on those parameters?

7- Results (line 150-151):

a- The t-SPC modulations seems to have similar patterns to power modulation in theta, alpha and beta but not in gamma. Is the lack of t-SPC modulation in gamma is related to the varying time-window procedure? since gamma is usually time-

locked to speech onset...

b- Figure 1 shows that gamma power modulation is prominent around speech onset mainly in postCG cortical contacts. Investigating the STN-ROIs SPC maps (ex STN-PostCG, STN-STG,) may reflect region-specific modulations that are shadowed when doing the analysis on all contact pairs.

8- Results (line 216-222): is the density calculated with respect to all contact pairs across all ECoG contacts or with respect to the contacts within the region of interest? this would impact the results if certain ROIs have relatively lower ECoG coverage than others.

9- Results (line 254, reference to Figure S10): Figure S10 do not show the information described. Maybe they should refer to Figure S11.

10- Results (line 261, reference to Figure S11-C): There is no Figure S11C, should be corrected to Figure 12-C

11- Results (line 295-297): Do these neurons cluster in a specific location in the STN? and do they couple with a specific region in the cortex?

12- Results (line 311-313): it seems that neurons with mixed firing rates modulate only beta t-SPC and no other frequency bands. Is there any reference in the literature relating these types of neurons with beta modulation (not necessarily in speech)?

13- Results (line 320-322): Are these results illustrated in Fig S14-D? Fig S14-D suggest that SMG and postCG have high spatial preference for neurons with decreased firing rate. STG seems to be within the 5th-95th percentile of the permutation distribution (grey box)

14- Methods (line 550-552): How does this linear time wrapping algorithm works considering three different recording systems with different sampling rates? the authors should explain more on the temporal alignment or provide reference.

(Remarks on code availability)

I've browsed the link to the shared code, but haven't tried them myself.

Reviewer #2

(Remarks to the Author)

The manuscript provides an insightful examination of spike-phase coupling (SPC) between subthalamic nucleus (STN) neurons and cortical regions during speech production by collecting and analyzing the dataset from patients with Parkinson's disease (PD) undergoing deep brain stimulation (DBS). The results showed that, during speech production, STN neurons exhibited transient coupling with cortical oscillations, especially in the theta-alpha frequency range, and that delayed or reduced SPC is linked to speech sound errors, suggesting a mechanistic role of the STN in coordinating speech via basal ganglia-cortical pathways. These findings suggest that the STN plays a critical role in coordinating speech by modulating the timing and accuracy of motor commands through its interactions with cortical regions, providing new insights into the understanding and refinement of the current models of speech production. My primary concerns pertain to the research background, motivation, and interpretation of the results. Below are the detailed issues that need to be addressed.

The manuscript examined the role of the STN in speech production, focusing on its interactions with cortical regions. However, the Introduction did not provide a comprehensive presentation of the extensive documentation available on the basal ganglia's involvement in speech production. Established theoretical models such as DIVA and a range of empirical studies (e.g., Narayana et al., *NeuroImage Clin.*, 2020; Parkinson et al., *NeuroImage*, 2012; Perez et al., *Laryngoscope*, 2019; Zarate and Zatorre, *NeuroImage*, 2008) that detail this role were insufficiently introduced. Significant contributions of the basal ganglia to speech production, as evidenced in studies on PD patients (e.g., Liotti et al., *Neurology*, 2003; Pinto et al., *Brain*, 2004; Narayana et al., *Hum. Brain. Mapp.*, 2010; Tang et al., *Hum. Brain Mapp.*, 2018), were also not adequately discussed. This omission misses an opportunity to fully contextualize the STN's critical function in speech mechanisms within the broader neurological framework. Moreover, the differential effects of STN-DBS in ON and OFF states on speech production (e.g., Pinto et al., *Brain*, 2004; Behroozmand et al., *Parkinsonism Relat. Disord.*, 2019) provide compelling evidence of the STN's significant impact, which underscored the relevance of examining STN-cortical interactions. These findings should be comprehensively discussed in the Introduction to better contextualize the research purpose.

In addition, the Introduction did not adequately justify the necessity and efficacy of using SPC to investigate STN and cortical interactions in speech production. The manuscript would benefit from an expanded description on how SPC offers advantages over alternative methods for revealing subcortical-cortical interactions to substantiate the selection of this approach. Furthermore, the rationale for employing a syllable triplet repetition task remains unclear. The manuscript did not explain why this particular task was chosen, nor does it address whether the task's specificity is crucial for probing STN-cortical interactions effectively, which is important to clarify whether the findings are task-dependent.

On the other hand, although this study presents compelling findings on the interactions between STN and cortical networks in the context of SPC, several aspects of their interpretations require further clarification. Firstly, in the second paragraph of the Discussion, the authors highlighted the consistency of SPC between STN neurons and cortical LFPs in motor control and speech production. Given the shared but distinct neural networks underlying these two functions, it would be valuable to further discuss their similarities and differences in STN-cortical SPC patterns. This discussion would be helpful for

understanding why DBS is effective for movement disorders yet can be ineffective or even detrimental to speech disorders. Secondly, the authors proposed that alpha-band SPC is involved in the internal monitoring of speech output and provides feedback for prediction accuracy. This hypothesis warrants deeper exploration by offering a more comprehensive explanation according to existing literature to support this idea. Thirdly, in lines 429-433, the authors suggest that their findings have significant implications for existing models of speech production (e.g. DIVA and SFC). The manuscript would benefit from a more detailed discussion of how these findings can refine these models by elucidating the role of the basal ganglia in the feedback and feedforward control of speech production, through interactions with cortical regions such as the STG, SMG, and IFG. Additionally, the implications of these findings for optimizing brain stimulation techniques, both invasive (e.g., DBS) and non-invasive (TMS), should be addressed more thoroughly. Specifically, elucidating the connectivity between the basal ganglia or STN and speech-related cortical areas could facilitate the localization of TMS hotspots tailored for treating speech disorders in PD, drawing parallels with successful applications in mental health treatments. Finally, the Discussion section addresses an excessive number of limitations without adequate clarity or conciseness. The first three points, concerning the appropriateness and effectiveness of the analysis methods, should be presented and addressed at the outset of the limitations to affirm the methods' validity. The latter points pertain to limitations specific to the study, particularly the generalizability of findings from PD patients undergoing DBS to populations without neurological disorders, which require clearer delineation and discussion.

There are several minor concerns requiring attention to ensure clarity and consistency:

- 1) There is inconsistent use of key terms and their abbreviations throughout the text, such as SPC (spike-phase coupling), PD (Parkinson's disease), STG (superior temporal gyrus), and LFP (local field potentials). It is imperative that each term be defined upon its first occurrence, followed by consistent use of the abbreviation thereafter to avoid confusion and enhance readability.
- 2) In the last paragraph of the Introduction, the authors referred to a protocol developed by their group but did not provide supporting references. It is essential to clarify whether this protocol was developed specifically for the current study or if it has been previously established.
- 3) It appears that only partial data were presented through sub-scores of UPDRS-III related to speech functions. For a more comprehensive understanding, it would be beneficial to include detailed assessments of speech functions, including additional quantitative measures that offer deeper insights into the speech capabilities and limitations of the participants.

(Remarks on code availability)

Reviewer #3

(Remarks to the Author)

This is a review of "Spike-phase coupling of subthalamic neurons to posterior opercular cortex predicts speech sound accuracy" (Nature Communications 5188438) by Vissani et al. The manuscript describes an experiment in which patients undergoing DBS electrode placement for Parkinson's underwent simultaneous recording of units in the subthalamic nucleus (STN) of the basal ganglia along with cortical surface (ECoG) electrodes while performing a speech repetition task. The authors performed an exhaustive set of analysis designed to look at spike-phase coupling (SPC) between STN units and cortical oscillations, primarily focusing on low-frequency oscillations (Beta and theta-alpha), examining both group-average SPCs and transient events. Results were compared across different regions, timing of hearing/speaking, and based upon speech production errors. They found the presence of speech-associated low frequency SPCs, with anatomic and temporal differences, as well as some correlation with speech errors.

Overall the work is interesting and well done. These patients provide a unique data set with the potential for insight into the role of basal ganglia in speech, which is something not well understood owing to the difficulty in neuroimaging. These have the potential to be an important advance, as much of what we know about the basal ganglia in vocal communication is from songbirds, and relatively little is known in mammals (although STN doesn't quite match the analogous fields of primary focus in birds). The analyses performed are robust and well done.

My biggest concern, however, is that the manuscript is incredibly difficult to read and follow along. This is a product of the large number of analyses done and, unfortunately, it became very easy to lose sight of the 'forest for the trees.' It was difficult to try and fit each new section/analysis into the big picture. By the time I got to a summary in the discussion, I was lost and had forgotten what had been done earlier. I also found myself having to look back at the methods almost every other sentence to understand what was being done. I strongly encourage the authors to revise the manuscript to make this work more accessible by 1) strengthening of the up-front summary/outline of the results, 2) when introducing each new analysis give a little more detail in the body of the manuscript so that the reader has a better sense of the general goals/approach of that analysis, and 3) end each section with text that better ties the results of that analysis back to the overall conclusions.

I detail some specific comments below:

1. Was there any evidence of auditory responses in the STN neurons? Most of the examples shown focus on the motor/speech side, and not much is shown about the perceptual responses. I ask because striatal auditory responses have been described in animals, but I do not think they recorded from STN.
2. Figure S4, would be useful to see corresponding t-stat maps like shown in Figure 2.
3. For the figures looking at the individual frequency bands, i.e. Fig 2E, 2F, it might be useful to flip the y-axes so that low frequency bands are at the bottom and high at the top. This is both more intuitive and aligns better with the time frequency maps.
4. Line 141-143 discusses that low frequencies have longer t-SPCs than higher. This is, perhaps, not unexpected. Low

- frequencies inherently last longer per cycle than high. So a spike-entrained event on a given cycle should show longer effects. This is why wavelet-type time-frequency analyses are typically done rather than just short-time fourier transforms.
5. Lines 153-159/ Fig S5. I am a bit confused about the comparisons across different frequencies. Previous analyses were done at the unit-ECoG pairing level, this seems to be at the unit level. So is this comparison looking at t-SPC similarity across all pairs associated with a given unit, or is this across all repetitions for a given pair (as suggested by line 153). The number of data points in Fig 2G suggests pairs. Also for these comparisons, were analyses done only when both pairs had significant responses, or was an analysis done to see if a unit/pair had one significant t-SPC, what were the chances of its other SPC analyses being significant, or similar patterns without being significant (i.e. how robust are these some of the unit responses, did they only have a few signif SPCs, or if a unit had one, they were likely to have others)?
 6. Fig S6, would be useful to see these analyses in a frequency band-specific manner. Also what is the difference between the top and bottom plots?
 7. Fig S7B. I am a bit confused on what is being plotted on the bottom. What is meant by relative occurrence of theta-alpha and Beta events?
 8. Line 216,-217. I think this is a great place to tease some of the analysis methods, as I had discussed in my comments about readability. I assume when you say 'density' you actually mean 'spatial density'. It would be very useful to mention here that you calculated the relative spatial density by doing blah blah (in general terms), otherwise 'density' is mentioned from left field, and then reviewer is not scratching their heads and then scanning through the long methods to find out what was being analyzed. Also, on lines 221-222 it says density was 'less frequent'. I am not sure I know what that means, since I thought this was a measure of spatial position.
 9. Figure S9 (and a few other places). Permutation tests were done to compare the different distributions. Was this done as a series of pairwise testing, or one large permutation, shuffling all the data together? If the former, were multi-comparisons corrections done?
 10. Line 257 – The principle component results are mentioned in passing, without having discussed that a PC spatial analysis was done and for what reason. This ties back to my comments about readability, and needing to introduce things a little better before giving their results.
 11. Figure S13 / lines 285-287. How was the mismatch between the small number of units and the larger number of t-SPC pairs handled? It would seem there would be redundancy of the unit side (i.e. each unit had several SPCs pairings).
 12. Also on Fig S13/S14. I am not sure I understand or can intuit what a change in spatial density means or how it was calculated. I can see a change in the fraction/number of SPCs, but since nothing is moving anatomically, are we really talking about a change in 'density' or just frequency of responsive events. For example, how can a neuron have a higher density, I thought that was a spatial measure (line 302-303)?
 13. It would be interesting to discuss the different t-SPC responses across areas by unit firing type (Fig S14). Is there any intuition to be gained from this analysis about specific modes of communication for a neuron that has increased vs decrease firing? They do seem to have somewhat different t-SPC bands.
 14. Figure 5, it seems like the accuracy effects extend very early into the auditory bins, which seems strange to me. Was any analysis done to associate responses with specific words/phonemes being tested? How do we interpret the fact that the primary differences between error and non-error trials are really in the auditory testing (or even earlier) and really go away once speech production actually begins. That seems strange to me, almost suggests more of a sensory/perceptual error than a motor one. It is difficult to tell since there is not a longer delay period between stimulus presentation and repetition.
 15. Also in Figure 5, why are there so few neurons for this analysis? Only about a third of the whole paper's units are included in 5A, and even less in 5D/E.
 16. I am not sure I completely buy the conclusion that these results suggest some sort of error monitoring/generation, since there is little difference in SPCs during the speaking period, when the speech error is actually happening. There is a little difference after speech offset in the theta-alpha band, but this is rather late given what we know about error-related signals for speaking happening during actual production, at least for sensory cortex. I guess there could be some sort of very late linguistic error detection that is delayed, but it still doesn't fit with most linguistic error models which posit a more on-line monitoring function (perhaps delayed for segmental speech parameters, but shouldn't only be occurring well after speech stops).
 17. Were any other parameters of speech/voice examined beyond phonemic errors? Parkinson's has known effects on pitch and loudness, which are also known to improve with DBS as discussed.
 18. Methods – I need more details on the time warping procedure used. All the plots seem to show a fixed onset/offset of sound presentation and speech. The former is obviously easier to design, but variable speech production makes the latter more variable. How did you do a time warp without affecting the frequency contents of the ECoG oscillations? Time warping has been done on spike trains in the past, which should still preserve relative firing rates (mostly – they can cause distortions, but usually works out on average), but continuous time signals are far more susceptible to distortion and frequency shifts with compression or expansion.

Minor Points

1. Figure S2 is cut off, I only see 1.5 panels
2. Figure 1D, I need more details of what is being plotted. What does the shading correspond to? I assume the error bars are SEM (text sort of hints at that, but doesn't explicitly say that)
3. I don't see Figure 1E mentioned in the text (unless there is typo on line 108 where it cites Figure 1D).
4. Nit picky, but at times the manuscript uses too many digits of precision that are likely not meaningful (i.e. neurons having average 92.78 LFP pairs, I don't think that additional precision adds much). Not a big deal, though.
5. Fig 2G, could you please title/label the three different plots
6. I can't find Figure S12 referenced anywhere in the manuscript.
7. Line 552 – do you really mean 'sub-millimetric' precision or sub-millisecond?

(Remarks on code availability)

Version 1:

Reviewer comments:

Reviewer #1

(Remarks to the Author)

I think the authors have invested substantial effort in addressing our comments. The first version of the manuscript was already very technical and sometimes hard to follow, but they managed to reorganize the results section and provide methodological clarifications as well as new figures and tables, which I think have improved the manuscript's clarity and rigor.

(Remarks on code availability)

Reviewer #3

(Remarks to the Author)

The authors have done a good job exhaustingly addressing the comments made by the other reviewers and myself, I have no other concerns.

(Remarks on code availability)

Author Rebuttals to Initial Comments

Reply to reviewers for Vissani et al. submission NCOMMS-24-37840

Color code: Original(black), Our reply (blue), Edits included in revised manuscript (red)

We thank the reviewers for thoroughly assessing our manuscript and providing critical and constructive feedbacks for the manuscript. We are delighted to see that all reviewers recognized the relevance and rigor of our findings. We greatly appreciate the many suggestions offered by reviewers, which we took to heart and implemented. We believe that the resulting changes have substantially improved the manuscript.

We have provided detailed responses to the reviewers' comments below. Here is a summary of the key changes made to the manuscript:

1. Reorganization of Results:

- We have prioritized findings related to the Main Figures and moved all analyses concerning Supplementary Figures (e.g., the relationship between spike-phase coupling (SPC) and instantaneous firing rate (IFR), time delay analysis, and control analyses) to the final paragraphs.

2. Title Revision:

- The title now uses the term *perisylvian* instead of *opercular* to more accurately refer to the SMG-posterior STG (pSTG) complex.

3. Introduction and Discussion Revisions:

- Both sections were extensively revised based on the reviewers' suggestions. Specifically, we now discuss three alternative hypotheses to explain the observed differences in theta-alpha SPC before speech production in error versus accurate trials.

4. Methods:

- The number of permutations used to compare distributions was increased from 500 to 1000, and we applied the t-max correction to control for multiple comparisons. Note: this adjustment does not apply to the cluster-based permutation test used to define SPC significance (500 permutations). We added Supplementary Text to provide further methodological details.

5. New Figures and Tables:

- **Figure S5:** Provides a detailed illustration of the computation of the main metrics used in the manuscript.
- **Figure S18:** Includes an *in-silico* test to validate the robustness of the SPC pipeline.
- **Figure S21:** Displays power modulation in ECoG contacts with and without SPC.
- **Table S3 and S5:** Report the cortical and subcortical coverage for each analysis.
- **Table S2:** Offers additional details on speech impairment in the cohort of PD participants.

In addition to what was requested, we would also like to highlight that our curated dataset is hosted in the Data Archive BRAIN Initiative (DABI, <https://dabi.loni.usc.edu/dsi/1U01NS098969>) and is available upon request. We also converted all the data needed to reproduce the main results of this paper into a preprocessed dataset (exemplary recordings with spike-phase coupling metrics for all available pairs) available on Zenodo (Vissani, 2024) (<https://doi.org/10.5281/zenodo.12610957>). We are also making available example code that shows how to utilize these files on Github (https://github.com/Brain-Modulation-Lab/code_SPC_ECoG_STN_Speech). We modified the Code and Data availability section accordingly.

Point-to-point reply

Reviewer #1:

In the reported study, the authors used high-density electrocorticography (ECoG) to record local field potentials across cortical areas known to participate in speech perception, planning, and production, with simultaneous recording of STN single-unit activity during DBS surgeries in 24 participants when they spoke syllable repetitions. The authors reported that individual STN neurons have transient (200ms) spike-phase coupling (SPC) events with multiple cortical regions. The spike timing of STN neurons was coordinated with the phase of theta-alpha oscillations in the posterior supramarginal and superior temporal gyrus during speech planning and production. Speech sound errors occurred when this STN-cortical interaction was delayed.

The results are very interesting and novel, the methodological procedures are robust. We (I myself and an early career research scientist in my lab) had few comments and questions, mainly in the results and methods sections. Here below are the comments and questions, in order of where they appear in the manuscript, now in order of importance:

We thank reviewer 1 for their thorough consideration of our work. The reviewer raised important methodological considerations which we fully addressed as outlined below.

R1.1: Results (line 81-82): Figure S2 is not complete.

We apologize for this oversight. We fixed the visualization of Figure S2, which we report here for reviewer's convenience.

Figure S2: Electroecorticographic channels from four exemplary participants. Electrode locations and time-frequency spectrograms locked to auditory cue (AC) (black dashed line) onset and speech production (SP) (magenta dashed line) onset are depicted for each participant. Two representative sensorimotor electrodes (e1, e2) and three auditory electrodes (e3, e4, e5) are shown. Time-frequency spectrograms are normalized with respect to the baseline (during the inter-trial interval (ITI)) and expressed as dB.

R1.2: The authors mentioned in Line 86 that, ‘premotor and postcentral gyrus electrodes showed –suppression and –elevation preceding the speech onset and during speech production’. In the task, speech production is followed by the auditory cue, which was associated with beta reduction. Whether the changes shown preceding the speech production is actually the response to the auditory cue?

We agree with reviewer’s comment. We believe that the desynchronization of beta oscillations in the precentral (PreCG) and postcentral (PostCG) gyri does not encode speech-specific information. Instead, it likely reflects the release of cortical inhibition, facilitating movement execution, cognitive flow, and attentional focus (Engel & Fries, 2010; Peter et al., 2022). Beta power decreases during auditory cue presentation, continues to decrease until speech termination, and rebounds afterward. For details, please refer to Figure S2 and the new Figure S21 (included here for convenience), which illustrates beta power modulation across multiple regions of interest during the speech production task. In contrast, we interpret gamma synchronization in the PreCG and PostCG as reflecting speech-related processes, consistent with extensive literature on this topic (Bouchard et al., 2013; Crone et al., 2006). Gamma synchronization occurs preceding and during speech production, aligning with its role in speech processing.

However, we acknowledge that late auditory processing cannot be entirely ruled out in our event-locked power modulation analysis. This limitation arises because the task design does not explicitly isolate auditory and speech-related processes. Moreover, the median time interval between the auditory cue offset and speech production onset (0.6 s) may be insufficient to fully separate these processes.

We amended the Results section as follows:

*Figure S2 illustrates different patterns of evoked activity in five representative electrodes from four participants. Spectrograms demonstrated consistent neural suppression in lower α - β frequencies (8-30 Hz) as well as elevation in the γ range (50-150 Hz) during auditory cue presentation and speech production. β power suppression was a ubiquitous phenomenon occurring across time and not temporally specific to processes related solely to speech. A large fraction of STG electrodes displayed either transient or sustained increased γ -activity in response to auditory cues (Hamilton et al., 2018). In line with previous work (Chrabaszcz et al., 2019; Towle et al., 2008), precentral and postcentral gyrus showed γ -elevation preceding speech onset and during speech production, which likely reflects speech-related processing.– **Results, pg. 5***

Figure S21: Cortical ECoG contacts with significant SPC in the PostCG, PreCG and SMG are more task responsive. Comparison of the speech-related power modulation in ECoG contacts without significant t-SPC events (red) and those with significant t-SPC events across any frequency band (blue), as well as within specific bands: θ - α (green), β (orange), γ_L (yellow), and γ_H (magenta). Data are grouped by region of interest (columns) and frequency bands (rows). Shaded area Only conditions with at least 20 ECoG are shown. Shaded areas illustrate the 5th and 95th percentile of the

R1.3: The authors showed that neurons' instantaneous firing rates during the speech task was heterogeneous both within and across recording session (also shown in Figure 1-D). – We are wondering if there is any spatial distribution of each neuron category: Are the increasing/decreasing IFR neurons clustered in sweet spots?

We analyzed the spatial distribution of the four categories of STN neurons based on speech-related firing rate modulation. Our data (please refer to Figure R1.1) showed that these neurons are spatially overlapped in the STN. We have also amended the Results section as follows:

STN neurons with distinct firing rate modulations were not significantly spatially segregated within the STN, as assessed by comparing the average distance between these neuron categories against a null distribution of distances based on our sampling of recording sites (all $p_{perm} > 0.05$).
– Results, pg. 5

Figure R1.1: Neurons with different firing rate modulations are overlapped in the STN. (A) Spatial distribution of STN neurons grouped by firing rate modulation category. Each dot is a single neuron. A little spatial jitter was applied to resolve multiple neurons located in the same run of the task. **(B)** Histogram of the permutation distances between two categories of neurons and the observed distance (vertical red line) are illustrated. Firing rate categories: decreasing type neurons (blue sphere), no modulation type neurons (gray sphere), increasing type neurons (red sphere) and mixed type neurons (green sphere).

R1.4: Line 119 -- Only 11% (2148/19755) of the time-frequency maps displayed significant SPC. What are the neuron types for those which displayed significant SPC: is there any consistent trend in their firing rate in response to speech production: increase or reduce, or are they very heterogeneous? Similar to the LFP channels showing SPC: are they showing most speech related modulation, such as more beta reduction and theta/alpha increase during speech?

Regarding spike-phase coupling and firing rate relationship, the vast majority of neurons (N=206/211, 97.6%) displayed at least one significant SPC pair, and hence these neurons reflect the whole heterogeneity of firing rate modulation that we observed in our dataset. Please refer to new Figure S15A.

We added additional analyses along the lines suggested by the reviewer to provide deeper insight into the relationship between spike-phase coupling and power modulation for LFP channels. Keeping in mind that a thorough analysis of the speech-related power modulation in the ECoG modulation is out of the scope of this work and it would be part of another manuscript under preparation, we focused specifically on the question: "Do cortical ECoG contacts with and without significant SPC exhibit different speech-related power modulation?" by comparing the task-related power modulation across ECoG contacts with different spike-phase coupling profiles.

*We further explored the influence of power magnitude on SPC by examining changes in power during speech production in ECoG contacts, with a specific focus on the θ - α bands in relation to SPC. We categorized ECoG contacts into six groups (No-SPC, SPC in any band, θ - α SPC, β SPC, γ_L SPC, and γ_H SPC) and compared the speech-locked power modulation with respect to the baseline across frequency bands (θ , α , β , γ_L , γ_H) and cortical ROIs. To ensure balanced comparisons, we included only conditions with at least 20 ECoG contacts and estimated the distribution of the mean by resampling 20 ECoG contacts 500 times.— **Methods, pg. 35***

We added a new Supplementary Figure (Figure S21) that we report here again for convenience.

Figure S21: Cortical ECoG contacts with significant SPC in the PostCG, PreCG and SMG are more task responsive. Comparison of the speech-related power modulation in ECoG contacts without significant t-SPC events (red) and those with significant t-SPC events across any frequency band (blue), as well as within specific bands: θ - α (green), β (orange), γ_L (yellow), and γ_H (magenta). Data are grouped by region of interest (columns) and frequency bands (rows). Shaded area Only conditions with at least 20 ECoG are shown. Shaded areas illustrate the 5th and 95th percentile of the

We amended the Results section as follows:

*We further explored the influence of power magnitude on SPC by examining changes in power during speech production in ECoG contacts, with a specific focus on the θ - α bands in relation to SPC (Figure S21). We found that, unlike the β band, where we observed both β power suppression and β -SPC suppression during speech production, the θ - α power and θ - α SPC exhibited distinct patterns. Specifically, θ - α power was suppressed during speech production, while θ - α SPC increased. This indicates that the increase in θ - α SPC during speech cannot be attributed solely to an increase in the overall amplitude of θ - α oscillations. Next, we explored whether ECoG contacts with significant SPC differed from those without SPC in terms of task-related power modulation. We found that ECoG contacts in the SMG, PreCG, and PostCG regions with significant SPC were more responsive to the task, showing greater suppression of low-frequency oscillations and enhanced high γ activity during speech production (Figure S21). These control analyses reinforce the significance of the observed SPC patterns. – **Results, pg. 19***

R1.5: Line 132, ‘single-pair SPC maps showed that STN neurons entrain cortical LFP phases during transient episodes (Figure 2B), which we termed t-SPC’ – The word ‘entrain’ implies a strong causal relationship, I’m wondering if this is what the authors really want to mean, but I don’t think there is any proof for this.

We agree with the reviewer that the term “entrain” might imply a strong causal relationship which cannot be inferred with such spike-LFP functional connectivity metrics. We replaced the term “to entrain” with “to lock to” in all sections of the manuscript.

R1.6a: Results (line 140): The variable-window width procedure to estimate SPC maps, as described in the methods, alters the notion of accurate timing, given that each time bin can have a different window length. Did the author consider this when calculating the median duration of the events?

We thank the reviewer for raising this point and providing us the opportunity to clarify important technical aspects of the spike-phase coupling computation pipeline. We believe that our method is robust in the calculation of the timing of the t-SPC events. The variability of the window size is relatively small (mean (SD), 0.15 ± 0.01 s,) across all bins as a slight modification of the window is sufficient to exclude/ include the required number of spikes when enough trials are available, and the firing rate is not too low.

However, we decided to use a simulation to illustrate the ability of our method in identifying genuine SPC events in a reasonable set of parameters constrained by our data.

We amended the Supplementary Text section as follows:

In-silico evaluation of the SPC pipeline – *As our work mainly relies on the concept of time occurrence of transient spike-phase coupling (t-SPC) events, we evaluated the ability of our pipeline in identifying genuine SPC events in a reasonable set of parameters constrained by our dataset. We simulated a set of trials (N=80) of sinusoidal oscillations at a specific frequency of interest ($f = 8$ Hz) for 3 seconds. To avoid phase resetting, we inserted a noise in the phase of the oscillation. We modeled spiking times employing a Poissonian distribution with baseline spiking rate of 25 Hz. To incorporate firing rate modulation, we modulated the baseline spiking rate with a gaussian distribution centered around the peak of the neuron response (Mean = 1 (no modulation), 0.5 (50% reduction), 1.5 (50% increase); SD = 0.2 s). We also defined a ground-truth of SPC in a 0.3 s long window (comparable to our t-SPC events duration) around a preferred phase ($p/4$) and defined a modulation of the spiking rate based on the cosine distance of the current phase with respect to the preferred phase. This cosine distance was modulated in such a*

way that the ratio between the firing rate at the preferred phase and the firing rate at the opposite phase is equal to the effect size (ES)(Vinck et al., 2010), defined as $(1 + 2\sqrt{PPC})/(1 - 2\sqrt{PPC})$, which means that a PPC of 0.015 (comparable to what we obtained in our dataset, see Figure S20) would correspond to an ES of 1.65. We then ran the same cluster-based permutation test to extract the significance of SPC and defined the t-SPC. We swept a set of parameters and compared the timing identification of the t-SPC event: true PPC strength (or ES equivalently), the duration of the t-SPC event, the time resolution (number of bins or anchor points), the number of trials which regulates the target window size around each bin, the frequency of coupling, the baseline firing rate, the jitter of the behavioral events (standard deviation across trials), the firing rate modulation, and the sampling rate. In Figure S18A-B we reported an example of SPC computation with a simple moving average window (no adaptation of the window width) and our method without true t-SPC events (ES=1, PPC=0) and a genuine t-SPC event (ES=1.65, PPC=0.015) between 1.7 s and 2 s (red patch). The analysis (Figure S18C) also confirmed that the estimation of the duration of the SPC (black crosses: before cluster-based permutation test, red crosses: after cluster-based permutation test) is reliable for a reasonable set of parameters if a conservative cluster-based permutation is applied. Interestingly, sampling rate is extremely important to get accurate estimations and only drastic decreasing (~ 0.25) or increasing (> 2.25) modulations of firing rate might bias the timing identification of the t-SPC event. – **Supplementary Text, pg. 1**

Results are showed in a new Supplementary Figure (Figure S18) which for reviewer's convenience is reported here.

We amended the Results section as follows:

*First, as our work heavily relies on the concept of t-SPC events, we performed simulations to assess the ability of our pipeline to reliably identify genuine SPC events under a range of conditions relevant to our dataset (Figure S18 and Supplementary Text for details). The simulations systematically varied parameters related to task design (e.g., number of trials, variability in intra-participant behavioral events), neural activity (e.g., true SPC strength, SPC duration, baseline firing rate, firing rate modulation), and controllable signal conditioning settings (e.g., sampling rate, number of anchor points, target window width). Our results demonstrate that the variable-width procedure combined with a cluster-based permutation test effectively identifies non-spurious t-SPC events. Additionally, this approach achieves high accuracy in estimating the timing and duration of these events across a wide range of conditions. However, we found that the pipeline performance is particularly sensitive to the sampling rate, requiring sampling rates of at least 1kHz, underscoring the importance of careful parameter selection during data acquisition and preprocessing. – **Results, pg. 18-19***

Figure S18: Our SPC pipeline can accurately estimate transient events of spike-phase coupling. (A) (Top) Example of simulation of a neuron with increasing firing rate and no spike-phase coupling ($ES = 1$, (pairwise phase-consistency) $PPC = 0$). (Center) A moving window (fixed 150 ms width and 25 ms of time resolution) was used to compute the PPC and, hence, the SPC strength. (Bottom) Our SPC pipeline was used to estimate SPC strength. **(B)** Same as **(A)** with a true spike-phase coupling event ($ES = 1.65$, $PPC = 0.015$) in a 0.3 s long window (red patch). **(C)** Our method can accurately detect (before (black cross) and after (red cross) cluster-based permutation test) a short period of true spike-phase coupling (red patch) for a reasonable set of parameters if a conservative cluster-based permutation is applied. Set of parameters: true PPC strength (or ES equivalently), the duration of the t-SPC event, the time resolution (number of bins), the number of trials which regulates the target window size around each bin, the frequency of coupling, the baseline firing rate, the jitter of the behavioral events, the firing rate modulation, and the sampling rate. Sampling rate is extremely important to get accurate estimations and only drastic decreasing (~ 0.25) or increasing (> 2.25) modulations of firing rate might bias the timing identification of the t-SPC event.

R1.6b: In addition, as permutation is used for statistics, are the ‘duration’ and the number of significant SPC events sensitive to the total number of points in the 2D time-frequency map, the resolution in time and frequency, as well as any smoothing along time and along frequency in the 2D mat? How did the authors decide on those parameters?

We applied 2D smoothing only for visualization. No smoothing was applied for the statistical analysis to detect significant t-SPC events. The time resolution of the time-frequency map before smoothing is on average 10 anchor points in a 0.5 s window (50 ms) that implies 21 equidistant anchor points between two behavioral events for a total of 101 anchor points over the whole trial (see below for details). The average duration of a significant t-SPC event is ~ 260 ms which spans over at least 5 anchor points. Moreover, only a negligible portion of t-SPC events (only 3 out of 2987 ($\sim 0.1\%$)) spans only 1 bin (real duration of these events < 50 ms). For these reasons, we

believe that the resolution of the SPC map does not affect the identification of significant SPC events and the computation of SPC events features such as event duration.

We amended the Supplementary Text as follows:

Detailed explanation of the calculation for equidistant anchor points given a target time resolution – To calculate the number of anchor points between behavior events based on a target resolution, defined as $\frac{B_w}{W}$:

$$N_{\text{anchors}} = \left\lceil \frac{\left\lfloor \frac{B_w}{W} * T \right\rfloor + N_{\text{events}}}{N_{\text{events}} + 1} \right\rceil$$

where $\lfloor \cdot \rfloor$ is the floor function, B_w is the target number of anchor points in a window of duration W (B_w/W is the resolution, e.g., 50 ms), T is the duration of the trial, N_{events} is the number of relevant behavior events (4 in our case: auditory cue onset, auditory cue offset, speech production onset and speech production offset), N_{anchors} is the number of equidistant anchor points between two events. This calculation ensures continuity between intervals so that the final anchor of each interval matches the first anchor of the subsequent interval. The points are thus structured as: $[1 - N_{\text{anchors}}, 2 - N_{\text{anchors}}, 2 - N_{\text{anchors}}, 2 - N_{\text{anchors}}]$, creating an uninterrupted series of anchor points across the trial. – **Supplementary Text, pg. 2**

R1.7a: Results (line 150-151): a) The t-SPC modulations seems to have similar patterns to power modulation in theta, alpha and beta but not in gamma. Is the lack of t-SPC modulation in gamma is related to the varying time-window procedure? since gamma is usually time-locked to speech onset...

Regarding point (a), as noted by the reviewer, gamma power modulation is prominent around speech onset, typically extending across the entire speech production window (see Figure S2, Figure S21). Our previous work showed that gamma modulation related to speech began approximately 100 ms before speech onset and persisted until about 100 ms after speech offset in the sensorimotor cortex (Chrabaszcz et al., 2019). In the current analysis, we used a variable-width window approach, calculating SPC strength within windows of approximately 0.15 s centered around a fixed number of anchor points (with an average 50 ms time resolution) spanning contiguous behavioral events. This method (explained further in Supplementary Text) provides an unbiased estimation of SPC across all key task events, rather than focusing solely on single events, such as the auditory cue or speech onset.

In the case that an increase in gamma power is necessary to observe a genuine increase in gamma SPC, the 50 ms resolution ensures that gamma power increases extend across multiple computation bins of the SPC metric, providing consistent temporal precision around speech production onset. However, as illustrated in the new Supplementary Figure (Figure S21), an increase in SPC does not strictly depend on a simultaneous increase in power within the same frequency band (e.g., theta/alpha SPC increases during speech production even though overall theta/alpha power does not). Therefore, we are confident that the lack of gamma SPC modulation is not an artifact of gamma power modulation timing relative to speech onset.

The reviewer can refer to the amendment in R1.4.

R1.7b: Figure 1 shows that gamma power modulation is prominent around speech onset mainly in postCG cortical contacts. Investigating the STN-ROIs SPC maps (ex STN-PostCG, STN-

STG, ...) may reflect region-specific modulations that are shadowed when doing the analysis on all contact pairs.

Following the reviewer's suggestion, we examined STN-ROIs SPC maps (see the new Figure S8) and found no appreciable gamma modulation, even in the STN-PostCG SPC map. As expected, these STN-ROIs SPC maps align with our findings on the timing of spike-phase coupling events presented in Figure 3 (compare Figure 3B with Figure S8). In the flow of the manuscript, we now refer to this visualization to motivate subsequent analysis of transient-SPC (t-SPC) events.

We amended the Results section as follows:

Previous studies have shown that neural activity in the cortex and STN feature spectral topographies during resting state (Averna et al., 2023; Horn et al., 2017) and movement execution (Lofredi et al., 2018; Stolk et al., 2019). Consistent with these findings, when we grouped SPC maps by cortical regions of interest (ROIs), we observed qualitatively distinct SPC patterns (Figure S8). We extracted two t-SPC metrics to better delineate the spectral topography of the cortico-subcortical SPC during speech production. – **Results, pg. 11**

Figure S8: Average of the spike-phase coupling (SPC) maps for all spike-phase coupling pairs ($N = 2148$) in each region of interest (ROI). The pairwise-phase consistency (PPC) index is compared to the permutation distribution and expressed as z-score. Black and purple dashed lines denote auditory cue and speech production windows. The centroid location (black spheres) of the ECoG contacts included in this analysis is shown for each ROI. Refer to Table S3 for the number of pairs included in each ROI.

R1.8: Results (line 216-222): is the density calculated with respect to all contact pairs across all ECoG contacts or with respect to the contacts within the region of interest? this would impact the results if certain ROIs have relatively lower ECoG coverage than others.

We thank the reviewer for the opportunity to clarify this point. The spatial density on the cortical surface was defined as the ratio (multiplied by 100) of the number of spike-phase coupling events to the number of pairs within a region of interest (ROI). Importantly, we included only ROIs from the Destrieux atlas parcellation that met a minimum coverage criterion (>7 participants and >100 pairs), which included the Precentral gyrus (PreCG), Postcentral gyrus (PostCG), Supramarginal gyrus (SMG), Subcentral Gyrus (SCG), Superior temporal gyrus (STG), Middle frontal gyrus (MFG), and the orbital part of the inferior frontal gyrus (pars O.).

We included Table S3 which summarizes the number of participants, pairs, STN neurons, and ECoG contacts for each ROI.

Table S3: Number of pairs, participants, STN neurons and ECoG contacts included in the main analysis for each ROI.

ROI	PreCG	MFG	SMG	STG	PostCG	SCG	pars O.
# pairs	3547	3177	2716	2557	2440	1807	1178
# participants	22	17	18	22	20	19	9
# STN neurons	22	22	22	28	28	28	22
# ECoG contacts	110	91	85	51	75	67	53

Although there are clear differences in coverage across ROIs (an inherent limitation of ECoG recordings, which we address in the limitations section), we ensured that each ROI included at least 20 STN neurons and 50 ECoG contacts. Moreover, in all our permutation analyses, we explicitly accounted for variations in coverage by resampling the total dataset as many times as the sample size for each ROI to generate permuted statistics for that specific ROI. This process was repeated for the number of permutations performed. Consequently, the 5th and 95th percentiles of the null distributions adjusted accordingly, either shrinking (for higher sample sizes) or expanding (for lower sample sizes), which influences the likelihood of obtaining p-values below the significance threshold. For instance, this can be observed in the difference in the width of the shuffle distributions between STG (high coverage) and pars O. (low coverage) in Figure 3B.

R1.9: Results (line 254, reference to Figure S10): Figure S10 do not show the information described. Maybe they should refer to Figure S11.

Fixed it.

R1.10: Results (line 261, reference to Figure S11-C): There is no Figure S11C, should be corrected to Figure 12-C.

Fixed it.

R1.11: Results (line 295-297): Do these neurons cluster in a specific location in the STN? and do they couple with a specific region in the cortex?

We thank the reviewer for suggesting this analysis. We analyzed the spatial location of STN neurons that maintain their beta spike-phase coupling during speech production and whether they couple with a specific cortical region running the same spatial aggregation test that we applied in other analyses (please refer to Methods).

We amended the Results section as follows:

*We found that these neurons ($N = 16/211$, 7.6%) are not specifically clustered in a specific region in the STN ($X = -13.50$ mm, $Y = -14.74$ mm, $Z = -8.25$ mm, $p_{perm} = 0.065$). Interestingly, at the cortical level, these neurons mainly couple to the subcentral gyrus ($X = -62.82$ mm, $Y = -4.19$ mm, $Z = 29.14$ mm, $p_{perm} < 0.01$). – **Results, pg. 17***

R1.12: Results (line 311-313): it seems that neurons with mixed firing rates modulate only beta t-SPC and no other frequency bands. Is there any reference in the literature relating these types of neurons with beta modulation (not necessarily in speech)?

To our knowledge, no previous study has specifically investigated the relationship between mixed firing rate neurons and beta spike-phase coupling during speech production. Research on limb movement has documented varying firing rate modulation patterns, with some studies reporting increases or decreases, while most observe a reduction in beta-range spike-phase coupling during movements (Fischer et al., 2020; Lipski et al., 2017; London et al., 2021; Sharott et al., 2018). However, firing rate modulations involving mixed responses are rarely reported, and the relationship between firing rate dynamics and spike-phase coupling has seldom been explored in detail. In our dataset, we observed that the modulation pattern of firing rates in STN neurons—whether increasing, decreasing, or mixed—significantly influences the frequency specificity of speech-related phase-of-firing coding. Specifically, as shown in Figure S15C, the transition from neurons with decreasing firing rates to those with mixed responses results in a shift in the frequency-specificity of SPC from a low-pass profile (dominated by theta-alpha coupling) to a narrowband profile centered around beta frequencies. (Broicher et al., 2012) provide complementary insights through dynamic-clamp experiments in hippocampal pyramidal neurons under in vivo-like conditions. Their findings demonstrate that SPC frequency profiles are shaped by conductance states and firing rates. In low-conductance states with reduced firing rates, neurons exhibited low-pass coupling. In contrast, high-conductance states with elevated firing rates resulted in bandpass coupling. These differences are likely driven by mechanisms such as spike rate adaptation, which modulates the input-output gain (current-voltage relationship) and functions as a high-pass filter, as well as frequency resonance intrinsic to the spike-generation process.

We have amended the Results section as follows:

Both the group-level SPC maps and t-SPC event analyses revealed that only neurons with decreased firing rates significantly contributed to θ t-SPC events during speech production (Figure S16A-C). In contrast, neurons with either decreased or increased firing rates exhibited similar profiles for a t-SPC events. θ - α t-SPC events occurred less frequently during auditory cue presentation and more frequently during speech production. Notably, neurons with increased or mixed firing rate modulation predominantly contributed to the aggregation of β t-SPC events during the rebound phase. Only neurons with mixed firing rate modulation showed a significant aggregation of β t-SPC events during the baseline period. In summary, neurons with decreasing

firing rates exhibited t-SPC events dominated by θ and α rhythms, while neurons with increasing firing rates showed t-SPC events in the α - β range. This “band-pass” profile was particularly narrowband in β for neurons with mixed firing rate modulation. No distinct pattern of t-SPC coupling was observed in neurons without firing rate modulation at any level of analysis (Figure S16A-C). These results suggest that the pattern of firing rate modulation in STN neurons — whether increasing, decreasing or mixed — affects the frequency specificity of speech-related phase-of-firing coding. – **Results, pg. 17**

We have amended the Discussion section as follows:

STN neurons were preferentially coupled to a single frequency band which was significantly explained by the pattern of firing rate modulation in STN neurons. STN neurons with decreasing firing rates exclusively drove θ SPC. In a previous study (Lipski et al., 2018) we found these neurons to be temporally locked to the onset of the auditory cue. Conversely, neurons with increasing firing rates displayed SPC in β , which was notably narrowband around 17 Hz in neurons with mixed firing rate dynamics. The phenomenon of spike-phase coupling in response to behavioral state transitions and activity shifts remains relatively underexplored. Broicher and colleagues (Broicher et al., 2012) utilized dynamic-clamp experiments to replicate in vivo-like conditions in hippocampal pyramidal neurons, showing that SPC frequency profiles are modulated by conductance states and input firing rates. Specifically, neurons in low-conductance states with reduced firing rates exhibited low-pass coupling, whereas neurons in high-conductance states with elevated firing rates displayed bandpass coupling. These differences can be attributed to mechanisms such as spike rate adaptation, which modulates the input-output gain (current-voltage relationship) and functions as a high-pass filter (Benda & Herz, 2003), as well as frequency resonance intrinsic to the spike-generation process. Our findings suggest that similar mechanisms underlie SPC dynamics in the STN. These results underscore the dynamic nature of phase-of-firing coding within the STN, driven by a complex interplay of neural network states and the intrinsic adaptive properties of individual neurons during speech-related tasks. – **Discussion, pg. 22**

R1.13: Results (line 320-322): Are these results illustrated in Fig S14-D? Fig S14-D suggest that SMG and postCG have high spatial preference for neurons with decreased firing rate. STG seems to be within the 5th-95th percentile of the permutation distribution (grey box)

We agree with the reviewer and acknowledge that this result, particularly in the PostCG, is not limited to decreasing-type neurons but valid for all speech-modulated (decreasing, increasing and mixed) neurons.

We have amended the Results section as follows:

All categories of speech-modulated neurons exhibited a preference for coupling with PostCG, while neurons with decreased and mixed firing rates also showed significant coupling with the SMG. – **Results, pg. 17**

Although not statistically significant, STG SPC ranks as the third highest among regions of interest (ROIs) for decreasing-type neurons. These neurons also show the highest coupling to the STG among all firing rate categories (Figure S16D).

Additionally, neurons with decreased firing rates showed the highest coupling to the STG among all firing rate categories (Figure S16D). – **Results, pg. 18**

R1.14: Methos (line 550-552): How does this linear time wrapping algorithm works considering

three different recording systems with different sampling rates? the authors should explain more on the temporal alignment or provide reference.

We thank the reviewer for giving us the occasion to provide more details as to the temporal alignment among the three different acquired data streams: Ripple (ECoG), Neuro-Omega (MERs) and Zoom-H6 (audio signals). We added a new paragraph in the Methods section “*Electrophysiological data alignment*” that explains in detail the time-alignment method, as follows:

Electrophysiological data alignment – To temporarily align the continuous recordings from the Ripple, Neuro-Omega and Zoom-H6 systems, we employed a linear time-warping algorithm based on the stimulus and produced audio channels. We defined the Ripple files as the “leader” time and independently aligned the Neuro-Omega and Zoom-H6 recordings to it. To this end, we first coarsely align the files from different sources manually (no warping) by marking easily identifiable landmarks on each file (i.e., the beginning of the first trial). We then split the files into chunks of around 100s and performed a staged optimization procedure, independently in each chunk, to find the precise alignment and warping factor. In the first stage, the envelopes of the corresponding audio signals from the two files were calculated at 100 Hz, by calculating the maximal absolute value in 10 ms bins. We next found the delay (j) between the envelopes, which maximized their cross-correlation (r_j) and adjusted the “follower” channel accordingly:

$$r_j(\bar{x}, \bar{y}) = \frac{1}{n} \sum_i \frac{(x_i - \bar{x})(y_{i+j} - \bar{y})}{\sigma_x \sigma_y}$$

Next, we applied the following time-warping algorithm: we calculated a smooth interpolation function, such that for all time points, where is the corresponding follower signal value. We defined the time-warping function, where is the ‘pivot time’ defined as the midpoint of the leader chunk to synchronize, is the time warping factor and a small ‘time translation’ correction. Using this function, we calculated the time-warped follower signal such that. We then optimized the time-warping parameters to maximize the correlation between and the leader signal pattern, that is, . We did the optimization using `fminsearch` in Matlab, by minimizing the cost function, as follows:

$$C(t_0, \gamma) = -\frac{\bar{p} \cdot \bar{s}}{\sqrt{\bar{s} \cdot \bar{s} \times \bar{p} \cdot \bar{p}}} + k_0 t_0^2 + k_1 (\gamma - 1)^4$$

where the regularization parameter was set to 0.0003 and to 0.001. To achieve sub-millisecond precision, a second stage was done using the same synchronization algorithm on the raw audio signal, low pass filtered to 5kHz and resampled to 10kHz for computational efficiency. Note that the fitted warp factor typically differed from unity in one part in 10^5 , meaning that the correction amounted to 1ms every 100s, and was very consistent within the participant and file type. The tolerance of the synchronization was defined as the maximal mismatch in synchronization between adjacent 100s chunks calculated for each participant. Sub-millisecond synchronization precision was achieved. Note that a 1ms mismatch only represents a 3% change in phase in the high b range and a 10% change for high γ . – **Methods, pg. 28**

Reviewer #1 (Remarks on code availability): I've browsed the link to the shared code, but haven't tried them myself.

Reviewer #2:

The manuscript provides an insightful examination of spike-phase coupling (SPC) between subthalamic nucleus (STN) neurons and cortical regions during speech production by collecting and analyzing the dataset from patients with Parkinson's disease (PD) undergoing deep brain stimulation (DBS). The results showed that, during speech production, STN neurons exhibited transient coupling with cortical oscillations, especially in the theta-alpha frequency range, and that delayed or reduced SPC is linked to speech sound errors, suggesting a mechanistic role of the STN in coordinating speech via basal ganglia-cortical pathways. These findings suggest that the STN plays a critical role in coordinating speech by modulating the timing and accuracy of motor commands through its interactions with cortical regions, providing new insights into the understanding and refinement of the current models of speech production. My primary concerns pertain to the research background, motivation, and interpretation of the results. Below are the detailed issues that need to be addressed.

We thank the reviewer for his constructive comments.

R2.1a: The manuscript examined the role of the STN in speech production, focusing on its interactions with cortical regions. However, the Introduction did not provide a comprehensive presentation of the extensive documentation available on the basal ganglia's involvement in speech production. Established theoretical models such as DIVA and a range of empirical studies (e.g., Narayana et al., *NeuroImage Clin.*, 2020; Parkinson et al., *NeuroImage*, 2012; Perez et al., *Laryngoscope*, 2019; Zarate and Zatorre, *NeuroImage*, 2008) that detail this role were insufficiently introduced. Significant contributions of the basal ganglia to speech production, as evidenced in studies on PD patients (e.g., Liotti et al., *Neurology*, 2003; Pinto et al., *Brain*, 2004; Narayana et al., *Hum. Brain. Mapp.*, 2010; Tang et al., *Hum. Brain Mapp.*, 2018), were also not adequately discussed. This omission misses an opportunity to fully contextualize the STN's critical function in speech mechanisms within the broader neurological framework. Moreover, the differential effects of STN-DBS in ON and OFF states on speech production (e.g., Pinto et al., *Brain*, 2004; Behroozmand et al., *Parkinsonism Relat. Disord.*, 2019) provide compelling evidence of the STN's significant impact, which underscored the relevance of examining STN-cortical interactions. These findings should be comprehensively discussed in the Introduction to better contextualize the research purpose.

We have extensively reworked the Introduction to address the relevance of basal ganglia more fully in speech. Due to space constraints, we were not able to integrate all of reviewer's suggested citations into the manuscript. As one example, (Zarate & Zatorre, 2008) offers some evidence that the cingulate cortex, auditory cortex, and putamen are recruited in expert singers learning to monitor their auditory feedback. While this is interesting evidence, we prioritized other papers that we believe more directly address BG involvement in speech.

We amended the Introduction as follows:

The cortico-basal ganglia network is a structural foundation for supporting motor control (Alexander et al., 1986; Lanciego et al., 2012), including human orofacial motor control for speech. Studies of speech and neurological speech impairments strongly support the idea that basal ganglia play a role in speech production. Positron emission tomography (PET) and functional magnetic resonance imaging (fMRI) have shown basal ganglia nuclei activation during

speech production tasks(Ghosh et al., 2008; Price, 2012; Riecker et al., 2005; Wildgruber et al., 2001) and have suggested a role of basal ganglia in timing, rhythm control, and prosody(Frühholz et al., 2015; Klaas et al., 2015; Mitchell et al., 2016; Pichon & Kell, 2013). Clinical observations in patients with basal ganglia lesions or diseases affecting the basal ganglia bolster the findings from basic neuroscience. Lesions to adult basal ganglia can induce stuttering(Ciabarra, 2000; Theys et al., 2024), articulatory impairments(Warren et al., 2000), and dysprosody(Vanlanckersidtis et al., 2006). Individuals with a mutated FOXP2 gene – which is thought to primarily affect neurons in the basal ganglia(Enard et al., 2009) – experience apraxia of speech along with linguistic and grammatical impairments, despite normal intelligence and hearing(Hurst et al., 1990). Approximately 90% of patients with Parkinson’s disease (PD), whose most severe cardinal motor symptoms stem from basal ganglia pathology, suffer from speech disorder known as hypokinetic dysarthria(Duffy, 2020; Ho et al., 1999; Logemann et al., 1978).

Deep Brain Stimulation (DBS) of the subthalamic nucleus (STN), a key basal ganglia node, reliably improves gross motor symptoms in PD, but its effects on speech are poorly understood. There is currently no consensus on why STN-DBS leaves speech unaffected or mildly improved in some patients(Behroozmand et al., 2019; Karlsson et al., 2013; Lundgren et al., 2011; Manes et al., 2024; Moreau et al., 2011; Skodda et al., 2014; Van Lancker Sidtis et al., 2010), but contributes to speech decline in others(Dromey & Bjarnason, 2011; Klostermann et al., 2008; Skodda et al., 2014; Törnqvist et al., 2005; Tripoliti et al., 2008). Recordings from awake DBS surgeries offer a rare window to study the interactions between the STN and cortex during speech. The discovery of single unit(Johari et al., 2023; Lipski et al., 2018, 2024; Tankus et al., 2021; Tankus & Fried, 2019; Watson & Montgomery, 2006) and population level(Hebb et al., 2012; Hell et al., 2023) activity in the subthalamic nucleus that tracks multiple aspects of speech production(Chrabaszcz et al., 2019; Dastolfo-Hromack et al., 2022; Lipski et al., 2018, 2023) and emerging evidence for anatomical(Jorge et al., 2022) and functional connectivity(Weiss et al., 2023) between the STN and sensorimotor and auditory cortical areas, raises the question of how STN and cortex interact to mediate speech-related behavior. – **Introduction, pg. 3**

R2.1b: In addition, the Introduction did not adequately justify the necessity and efficacy of using SPC to investigate STN and cortical interactions in speech production. The manuscript would benefit from an expanded description on how SPC offers advantages over alternative methods for revealing subcortical-cortical interactions to substantiate the selection of this approach. Furthermore, the rationale for employing a syllable triplet repetition task remains unclear. The manuscript did not explain why this particular task was chosen, nor does it address whether the task’s specificity is crucial for probing STN-cortical interactions effectively, which is important to clarify whether the findings are task-dependent.

The choice of which signals to acquire to investigate subcortical-cortical interactions was largely determined by clinical constraints; microelectrodes in STN are clinically indicated for functional mapping and they are one of very rare opportunities to record single units in humans. For research purposes, we additionally recorded cortical LFPs by placing ECoG strips over lateral speech motor cortex through the same burr hole used for DBS leads implantation as well as subcortical LFPs from a macroelectrode 3mm above the microelectrode tip. For this project, we chose to analyze the functional connectivity between cortical LFPs and STN single neurons. Thus, spike-phase coupling was the natural choice of type of inter-areal connectivity to investigate. We want to clarify that the term spike-phase coupling refers to the generic “spike-LFP phase relationship”, while the metric that we adopted, the phase-locking value (PLV) and its extension pairwise-phase consistency (PPC) is a specific algorithm to estimate spike-phase coupling.

The main SPC algorithms are (please refer to this review (Zarei et al., 2018) for details):

1. Cross-Correlation and Coherence Coefficient: These methods evaluate linear relationships between spikes and LFPs but are unsuitable for non-linear or non-stationary dynamics and lack time resolution.
2. Phase-Locking Value (PLV): PLV measures phase consistency between spikes and LFPs but is heavily biased by spike rates, requiring spike equalization across trials, which can lead to significant data loss. Pairwise-phase consistency (PPC) is a correction of PLV which measures phase consistency across spike pairs and is not biased by spike rates. However, it is sensitive to noise, has high variance with low spike counts, and may yield negative values, which are physiologically irrelevant.
3. Spike-Field Coherence (SFC): SFC quantifies synchronization at specific frequencies. However, it is influenced by LFP magnitude, with spikes during high-magnitude cycles disproportionately affecting the results.
4. Spike-Triggered Correlation Matrix Synchronization (SCMS): SCMS constructs correlation matrices of LFP segments around spikes, reporting the largest eigenvalue as the synchronization index. Unlike other methods, SCMS captures phase correlations over broader time windows and accounts for multi-frequency dynamics.

We decided to use PLV (and PPC) because it is one of the most established method and it has been extensively evaluated in literature (Vinck et al., 2010, 2012). We also adapted the event-related variable-width window approach for PLV and PPC previously used in another study of our group (Fischer et al., 2020).

We amended the Methods section as follows:

*Although different methods have been proposed to estimate SPC, we opted to use PLV (and its extension PPC) because it is one of the most established methods and its limitations have been extensively studied in literature (Vinck et al., 2010, 2012). – **Methods, pg. 30***

We originally designed the syllable triplet task to investigate how cortical-subcortical networks sequence and execute speech motor programs. We selected triplets of syllables (that are phonotactically-legal but meaningless) to remove any linguistic confounds without increasing difficulty of articulation for patients. The stimuli were designed to be balanced across articulatory dimensions (dorsal, coronal, labial consonants and high, low, and mid vowels) and as well as acoustic and manner-of-articulation dimensions.

Regarding task-specificity: we were not able to run multiple tasks with the patients, due to clinical constraints. This is an unavoidable limitation of human intracranial research.

We have added a paragraph to the Discussion to address this.

*Lastly, we are unable to draw any conclusions based on our data regarding speech specificity because patients completed only the speech task in the operating room. We therefore can't weigh in on the differences between cortico-basal ganglia interaction in speech versus limb motor control. Given the differential patterning of speech and non-speech motor control in PD and treatments for PD (Kompoliti et al., 2000; Skodda et al., 2010; Tykalova et al., 2022), future research may explore and compare different movements modalities. – **Discussion, pg. 25***

R2.1c: On the other hand, although this study presents compelling findings on the interactions between STN and cortical networks in the context of SPC, several aspects of their interpretations require further clarification. Firstly, in the second paragraph of the Discussion, the authors

highlighted the consistency of SPC between STN neurons and cortical LFPs in motor control and speech production. Given the shared but distinct neural networks underlying these two functions, it would be valuable to further discuss their similarities and differences in STN-cortical SPC patterns. This discussion would be helpful for understanding why DBS is effective for movement disorders yet can be ineffective or even detrimental to speech disorders.

Unfortunately, it remains unclear why DBS effects on speech are generally ineffective. Because we did not have a limb control task and because other studies differed significantly in their cortical coverage, we do not believe we can adequately address this question.

Please refer to the amendment in the Discussion in **R2.1b**.

R2.1d: Secondly, the authors proposed that alpha-band SPC is involved in the internal monitoring of speech output and provides feedback for prediction accuracy. This hypothesis warrants deeper exploration by offering a more comprehensive explanation according to existing literature to support this idea.

We proposed that theta-alpha SPC may be involved in phonological working memory or sensorimotor integration based on its timing and cortical distribution. We have extensively reworked the Discussion, offering a more comprehensive explanation for our interpretation of the theta-alpha SPC differences in error trials.

We amended the Discussion as follows:

We uncovered a neural correlate of speech errors in our syllable repetition task: delayed, lower θ - α t-SPC between STN and SMG-pSTG (posterior STG) (Figure 5). Here we discuss two possible interpretations of how the θ - α SPC differences relate to the speech errors. One possibility is that the errors were related to phonological working memory (PWM)(Baddeley, 1992). We defined error trials as those in which at least one off-target phoneme was produced. Participants frequently made substitution errors in which the off-target phoneme was perceptually dissimilar to the target phoneme (Figure S1); production errors were thus unlikely the result of perceptual errors. Instead, the errors may be rooted in the failure of PWM to maintain the proper sequence in memory until speech production. Additionally, θ - α SPC significant differences appear in the second half of the auditory window (Figure 5B), when we would expect a reliance on PWM to maintain the syllable sequence. θ - α SPC was observed predominantly in cortical regions that have long been implicated in PWM: the inferior parietal cortex and adjacent regions in pSTG (Figure 3)(Jonides et al., 1998; Paulesu et al., 1993). θ SPC has been documented as a mechanism subserving working memory(Lee et al., 2005), lending further credibility to the PWM account of the speech errors.

Another possibility is that lower θ - α SPC in error trials is related to auditory-motor integration, or the interface between auditory input and motor programs in the speech production system(Hickok, 2012). Because our auditory stimuli were phonotactically legal but meaningless, participants could not rely on lexical or semantic anchors to remember the verbal sequences. Participants would instead have to rely heavily on the 'dorsal stream' of auditory processing in speech in the dual-stream model of speech processing(Hickok & Poeppel, 2007). The dorsal stream translates from sensory information to a motor encoding. The neurobiological cornerstone of the dorsal stream is situated just adjacent to the SMG-pSTG complex we identified in this study, in an area called area sometimes called 'Spt' (Sylvian-parieto-temporal). Spt, at parieto-temporal boundary in and around the posterior Sylvian fissure, has been extensively studied for its sensorimotor properties(Awh et al., 1996; Behroozmand et al., 2018; Buchsbaum, Olsen, Koch, & Berman, 2005; Buchsbaum, Olsen, Koch, Kohn, et al., 2005). Spt is critical for auditory repetition as it is hypothesized to compute a 'coordinate transform' from auditory to motor space(Hickok, 2012). Lesions to this

area can cause conduction aphasia—the selective deficit of verbatim repetition, despite fluent spontaneous speech and intact language comprehension(Acharya et al., 2024; Benson et al., 1973). Here, we find evidence that Spt and adjacent regions in the posterior perisylvian cortex might achieve this well-established auditory-motor interfacing by recruiting the BG, and specifically by leveraging θ - α coupling with STN. – Discussion, pg. 23

R2.1e: Thirdly, in lines 429-433, the authors suggest that their findings have significant implications for existing models of speech production (e.g. DIVA and SFC). The manuscript would benefit from a more detailed discussion of how these findings can refine these models by elucidating the role of the basal ganglia in the feedback and feedforward control of speech production, through interactions with cortical regions such as the STG, SMG, and IFG.

The cortical speech motor control regions highlighted in this study—vSMC, SMG, and pSTG (posterior STG)—are key parts of the DIVA, dual-stream, SFC, and HSFC accounts of speech production. However, it is challenging to map these data directly onto the DIVA computational model because 1) the STN is not explicitly represented in the model and 2) these models are largely activation-based and are not concerned with electrophysiological mechanisms like neuron-LFP locking. Future computational models of speech may be able to work across levels of abstraction to maintain tractability but also consider mechanistic descriptions of brain interactions. At minimum, our data and results reinforce the need for accounts of speech that include the BG, and in particular the STN.

We briefly address the gradient order DIVA (GODIVA) model here because it concerns *speech planning* mechanisms(Bohland et al., 2010; Guenther, 2016) , which is informative for the θ - α SPC differences we observed >1 second before speech onset in accurate versus error trials. GODIVA posits a phonological content buffer for upcoming speech sounds. The buffer maintains multiple speech sounds in parallel, releasing them serially at the appropriate time. Our results highlight the role of the pSTG-SMG in the buffering process, while GODIVA posits that the buffer is subserved by areas in and around posterior inferior frontal gyrus. Two possible explanations for this apparent discrepancy are as follows. First, GODIVA is largely grounded in evidence from activation-based studies, while our t-SPC metric is a measure of connectivity. Although they are often closely linked, activation and connectivity are separate mechanisms that can reveal different patterns of neural coding. Electrodes in the inferior frontal gyrus were active in the high-gamma range during both the speech planning and production window (data not shown)—but did not communicate with STN via t-SPC events. Second, our verbatim repetition syllable task may require greater reliance on auditory-to-motor coordinate transform than many of the orthographically cued tasks which informed GODIVA. The nature of the phonological processing required in this auditorily-cued task design may shift the phonological processing load from inferior frontal regions to the posterior perisylvian regions (pSTG-SMG) highlighted in this study.

More data available from basal ganglia during speech may help in the future to refine these models.

We amended the Discussion section as follows:

The cortical task-activated speech regions in this study — SCG, PostCG, PreCG, SMG, and pSTG — are key parts of the DIVA(Guenther, 2016; Tourville & Guenther, 2011), state-feedback control(Houde & Nagarajan, 2011), and hierarchical state-feedback(Hickok, 2012) accounts of speech production. However, it is challenging to map our results directly onto these models because they 1) are largely activation-based and thus agnostic to electrophysiological mechanisms like LFP-spike interareal interaction and 2) focus with single-word production rather than speech sequencing, 3) do not detail different basal ganglia nodes like STN.

We briefly address the gradient order DIVA (GODIVA) model here because it concerns speech planning mechanisms(Bohland et al., 2010; Guenther, 2016) , which is informative for the θ - α SPC differences we observed >1 second before speech onset in accurate versus error trials. GODIVA posits a phonological content buffer for upcoming speech sounds. The buffer maintains multiple speech sounds in parallel, releasing them serially at the appropriate time. Our results highlight the role of the pSTG-SMG in the buffering process, while GODIVA posits that the buffer is subserved by areas in and around posterior inferior frontal gyrus. Two possible explanations for this apparent discrepancy are as follows. First, GODIVA is largely grounded in evidence from activation-based studies, while our t-SPC metric is a measure of connectivity. Although they are often closely linked, activation and connectivity are separate mechanisms that can reveal different patterns of neural coding. Electrodes in the inferior frontal gyrus were active in the high-gamma range during both the speech planning and production window (data not shown) — but did not communicate with STN via t-SPC events. Second, our verbatim repetition syllable task may require greater reliance on auditory-to-motor coordinate transform than many of the orthographically cued tasks which informed GODIVA. The nature of the phonological processing required in this auditorily-cued task design may shift the phonological processing load from inferior frontal regions to the posterior perisylvian regions (pSTG-SMG) highlighted in this study.

*Future computational models of speech may be able to work across levels of abstraction to maintain tractability but also consider mechanistic descriptions of brain interactions. At minimum, our data and results inform future computational models speech production that integrate the basal ganglia. – **Discussion, pg. 24***

R2.1f: Additionally, the implications of these findings for optimizing brain stimulation techniques, both invasive (e.g., DBS) and non-invasive (TMS), should be addressed more thoroughly. Specifically, elucidating the connectivity between the basal ganglia or STN and speech-related cortical areas could facilitate the localization of TMS hotspots tailored for treating speech disorders in PD, drawing parallels with successful applications in mental health treatments.

Thank you for the suggestion. We now address the potential implications of our research for TMS research in the last paragraph of Discussion.

Non-invasive neuromodulation techniques, like transcranial magnetic stimulation (TMS), have been evaluated as therapies to alleviate symptoms in Parkinson’s disease(Chen & Chen, 2019; Deng et al., 2022; Li et al., 2022). Our results could inform TMS studies targeting speech symptoms. Studies have demonstrated an improvement in hypokinetic dysarthric symptoms by stimulating around the pSTG-SMG complex implicated in this study(Brabenec et al., 2019, 2021). Further research is warranted to what degree TMS may alleviate more motoric versus more cognitive aspects of PD speech symptoms(Goodwill et al., 2017). – **Discussion, pg. 25**

R2.1g: Finally, the Discussion section addresses an excessive number of limitations without adequate clarity or conciseness. The first three points, concerning the appropriateness and effectiveness of the analysis methods, should be presented and addressed at the outset of the limitations to affirm the methods’ validity. The latter points pertain to limitations specific to the study, particularly the generalizability of findings from PD patients undergoing DBS to populations without neurological disorders, which require clearer delineation and discussion.

We have restructured the limitations section to improve clarity. We have removed some points that we believe were unnecessarily complicating interpretation of the data. We also added a section on task specificity to address other comments from reviewers.

Our findings should be interpreted in the light of several limitations. First, our intracranial recordings are from patients with PD. Caution must be exercised when interpretations of human neurophysiology are

drawn from observations collected in a pathological state. Specifically, differences in the STN baseline firing rate (Remple et al., 2011), abnormal subcortical beta oscillations (Brown et al., 2001; Hammond et al., 2007; Neumann et al., 2017; Vissani et al., 2021), and loss of movement specificity (Bronfeld & Bar-Gad, 2011) that characterize the Parkinsonian state may confound distinction of whether our observations generalize to speech in individuals without PD. There are no opportunities to record from human basal ganglia nuclei that are not in a pathological state; however, future research can clarify which aspects of our results generalize to non-pathological basal ganglia. Second, because recording locations were clinically determined, we had uneven coverage of the STN and of lateral speech motor cortex. Most microelectrode trajectories traversed the dorsolateral part of the STN, the clinical target for PD DBS (Horn et al., 2017). Hence, sampling of the ventro-medial region of the STN is limited. ECoG coverage also varied across participants and spanned a limited region of the cortical surface. We cannot rule out any other interaction of the STN with other cortical regions. Lastly, we are unable to draw any conclusions based on our data regarding task specificity because patients completed only the speech task in the operating room. We therefore can't weigh in on the differences between cortico-basal ganglia interaction in speech versus limb motor control. Given the differential patterning of speech and non-speech motor control in PD and treatments for PD (Kompolti et al., 2000; Skodda et al., 2010; Tykalova et al., 2022), future research may prioritize running multiple tasks with DBS patients, at the cost of fewer trials per task. – Discussion, pg. 25

There are several minor concerns requiring attention to ensure clarity and consistency:

R2.2: There is inconsistent use of key terms and their abbreviations throughout the text, such as SPC (spike-phase coupling), PD (Parkinson's disease), STG (superior temporal gyrus), and LFP (local field potentials). It is imperative that each term be defined upon its first occurrence, followed by consistent use of the abbreviation thereafter to avoid confusion and enhance readability.

We apologize for the confusion. We have now defined each term upon its first occurrence and verified the terminology consistency through all the manuscript.

R2.3: In the last paragraph of the Introduction, the authors referred to a protocol developed by their group but did not provide supporting references. It is essential to clarify whether this protocol was developed specifically for the current study or if it has been previously established.

Our group has an extensive experience in synchronous cortical (ECoG strips) and subcortical (single neurons) intraoperative recordings during speech production task. We now added references in the revised version of the introduction. Please refer to (Bush et al., 2024; Chrabaszcz et al., 2019; Lipski et al., 2018, 2024) for details.

We amended the Introduction as follows:

We established an intraoperative DBS protocol to simultaneously record local field potentials (LFPs) from high-density electrocorticography (ECoG) strips over speech cortex and single-unit activity from microelectrodes in the STN while PD patients completed a syllable repetition task (Bush et al., 2024; Chrabaszcz et al., 2019; Dastolfo-Hromack et al., 2022; Jorge et al., 2022; Lipski et al., 2018, 2024). – Introduction, pg. 4

R2.4: It appears that only partial data were presented through sub-scores of UPDRS-III related to speech functions. For a more comprehensive understanding, it would be beneficial to include detailed assessments of speech functions, including additional quantitative measures that offer deeper insights into the speech capabilities and limitations of the participants.

We completely agree with the reviewer in the sentiment that UPDRS-III is not a rich characterization of speech function. Although a voice quality assessment from a speech-language pathologist was part of the clinical workflow, this data was not collected consistently enough to be used in this study. However, in the context of our speech production task, we asked a team of trained speech-language pathologist students to evaluate the speech production at the single phoneme level in terms of articulation disorder and voice quality terms. We adopted three common articulation disorder terms (distortion, imprecision, and disfluency) and six voice terms (creaky, horse-harsh, voice-break, strain, tremor, and breathy). For a description of these terms see the Supplementary Text. In Table S2 we reported the percentage of phonemes labelled with these terms. We found that in our cohort all patients exhibited some degree of imprecision of their articulation (6.8 ± 5.3 %) and a creaky voice.

We report here Table S2 for convenience.

Table S2: Articulation and voice quality as evaluated by a trained speech-language pathologist. A trained team of speech-language pathologist students inspected the produced audio signals and evaluated the quality of the voice and articulation at the single phoneme level. Please refer to Supplementary Text for a description of each term. Spirantization is considered as a category of imprecise articulation.

ID	Articulation disorder terms [%]			Voice terms [%]					
	distortion	imprecision	dysfluency	creaky	horse-harsh	voice-break	strain	tremor	breathy
DBS3001	0.30	11.76	0.05	0	0	0	0	0	2.08
DBS3002	0.47	4.30	0	0.09	0	0.09	0	0.28	0.14
DBS3003	2.52	3.97	0.09	0.14	0	0.09	0.19	0.42	0.61
DBS3004	2.41	17.31	0.74	0	0	0	1.25	0.88	1.16
DBS3008	1.57	3.70	0.28	21.39	0	0	0	0.74	0
DBS3010	0.07	8.58	0	0.03	0	0.10	0	0.21	6.53
DBS3011	0.08	3.02	0	4.33	0	0.04	0	0.04	0.08
DBS3012	1.80	10.40	0	2.14	0	0.15	0.11	0.23	0.26
DBS3014	0.07	7.47	0.56	0.31	0	0	0	0	0
DBS3015	0.38	4.97	0	0.42	0	0	0	65.76	0.80
DBS3017	4.54	6.10	0	1.02	0	0.47	1.41	0.47	3.83
DBS3018	0.48	5.52	0	0.33	0	0.04	0	0.66	0.11
DBS3019	1.12	6.46	0.33	0	0	0.26	1.12	0.13	13.18
DBS3020	0.42	5.02	0.05	5.12	0	0.05	0	1.88	0.56
DBS3022	0.69	12.36	0	0	0	0	0.05	0.09	0.19
DBS3023	0.41	3.19	0.33	4.66	0	1.06	0	0.74	0
DBS3024	0.53	3.20	0.10	17.38	0	0.10	0.10	0.38	0.48
DBS3026	5.60	24.53	0.27	0	0	0	0	3.33	0.53
DBS3027	0.97	3.98	0.32	12.36	0	0.05	0	2.73	0.42
DBS3028	0.24	3.21	0.24	0.71	0	0.08	0	5.75	0.75
DBS3029	0.63	3.54	0.07	3.75	0	0	0.49	0	0.28
DBS3030	0.63	5.14	0.35	0.76	0	0	0.56	5.35	7.71
DBS3031	1.75	5.63	0.08	0.48	0	0	0	0.32	2.30
DBS3032	0.48	0.91	0	2.25	0	0.48	0	0	4.77
Mean (SD)	1.17 (1.40)	6.84 (5.27)	0.16 (0.20)	3.24 (5.73)	0 (0)	0.13 (0.14)	0.22 (0.43)	3.77 (13.30)	1.95 (3.20)

We amended the Supplementary Text as follows:

Articulation disorders and voice quality – A team of trained speech-language pathologists used a custom MATLAB GUI (github.com/Brain-Modulation-Lab/SpeechCodingApp) to annotate each phone of syllable triplet productions. We adopted three terms that characterized articulatory dimensions (distortion, imprecision, and disfluency) and six terms that characterized voice dimensions (creaky, horse-harsh, voice-break, strain, tremor, and breathy) of speech. A phone was considered distorted if it was so indistinct that the SLP could not identify it, imprecise if the phoneme was identifiable but lacked clarity and precision. Phones were labeled disfluent if any range of disruptions occurred such as blocks, prolongations, and repetitions. – **Supplementary Text, pg. 1**

We amended the Results section as follows:

*Trained speech-language pathologists annotated articulatory and voice features of each phoneme production (see Supplementary Text). All participants displayed some extent of articulatory imprecision (6.8 ± 5.3 % across phonemes) and creaky voice (3.2 ± 5.7 % across phonemes) (Table S2). – **Results, pg. 4-5***

Reviewer #3:

This is a review of “Spike-phase coupling of subthalamic neurons to posterior opercular cortex predicts speech sound accuracy” (Nature Communications 5188438) by Vissani et al. The manuscript describes an experiment in which patients undergoing DBS electrode placement for Parkinsons underwent simultaneous recording of units in the subthalamic nucleus (STN) of the basal ganglia along with cortical surface (ECoG) electrodes while performing a speech repetition task. The authors performed an exhaustive set of analysis designed to look at spike-phase coupling (SPC) between STN units and cortical oscillations, primarily focusing on low-frequency oscillations (Beta and theta-alpha), examining both group-average SPCs and transient events. Results were compared across different regions, timing of hearing/speaking, and based upon speech production errors. They found the presence of speech-associated low frequency SPCs, with anatomic and temporal differences, as well as some correlation with speech errors.

Overall the work is interesting and well done. These patients provide a unique data set with the potential for insight into the role of basal ganglia in speech, which is something not well understood owing to the difficulty in neuroimaging. These have the potential to be an important advance, as much of what we know about the basal ganglia in vocal communication is from songbirds, and relatively little is known in mammals (although STN doesn't quite match the analogous fields of primary focus in birds). The analyses performed are robust and well done.

We sincerely thank Reviewer 3 for their thorough and constructive feedback on our work. The reviewer raised several important points, which we greatly appreciate and have fully addressed, as outlined below.

My biggest concern, however, is that the manuscript is incredibly difficult to read and follow along. This is a product of the large number of analyses done and, unfortunately, it became very easy to lose sight of the ‘forest for the trees.’ It was difficult to try and fit each new section/analysis into the big picture. By the time I got to a summary in the discussion, I was lost and had forgotten what had been done earlier. I also found myself having to look back at the methods almost every other sentence to understand what was being done. I strongly encourage the authors to revise the manuscript to make this work more accessible by 1) strengthening of the up-front summary/outline of the results, 2) when introducing each new analysis give a little more detail in the body of the manuscript so that the reader has a better sense of the general goals/approach of that analysis,

and 3) end each section with text that better ties the results of that analysis back to the overall conclusions.

In response to the reviewer's general concerns as to the clarity of the methodology, we have created a new Figure S5, which we hope will provide a clearer overview of the methodologies employed throughout the manuscript.

We improved the readability of the manuscript adding multiple sections in the Results sections. We unfold these amendments point-to-point below.

Figure S5: Overview of the SPC metrics. (A) For each participant we calculated the spike-phase coupling in each possible pair (all-to-all schema, green lines) of ECoG contacts (red circle) and STN neurons (cyan circle) and obtained a time-frequency SPC map. A cluster-based permutation test was used to identify significant transient spike-phase coupling (t-SPC) events (black boxes with red contour) and binarize each SPC map. A pair was deemed to be significant if it contained one or more t-SPC events. (B) We grouped and combined these binarized SPC maps in different ways to extract different metrics: t-SPC occurrence was used to evaluate the time occurrence of t-SPC events and their degree of aggregation over time, t-SPC spatial density was used to evaluate the spatial distribution over cortex and STN (grouping by ROI or 3D sphere, refer to Methods for details), and t-SPC index was used for analyses in which the unit of observation was the STN neuron (grouping by neuron). All metrics are expressed as percentage.

I detail some specific comments below:

R3.1: Was there any evidence of auditory responses in the STN neurons? Most of the examples shown focus on the motor/speech side, and not much is shown about the perceptual responses. I ask because striatal auditory responses have been described in animals, but I do not think they recorded from STN.

This is a great point raised by the reviewer. Our group is specifically addressing this in another project, and we are currently working on another manuscript that show that STN neurons do respond to auditory stimuli. Briefly, we did in fact find that ~40% of STN neurons exhibited auditory responses

R3.2: Figure S4, would be useful to see corresponding t-stat maps like shown in Figure

We expanded the Figure S4 including the t-stat maps following the approach in Figure 2A. We also revised the caption of the figure accordingly.

We report here for convenience:

Figure S4: Comparison of average SPC maps across different subsets of pairs. (Top) Results are consistent whether we averaged only significant SPC maps (N = 2148, same panel of Figure 2A), all pairs (N = 19755) or the most significant SPC map for each unit. (Bottom) Same as Top with the group-level statistical test (t-stat) of the significance of the z-score PPC with respect to the baseline. Red and blue lines contour regions of significant SPC increase or decrease, respectively. Auditory cue (AC) (black dashed line) onset and speech production (magenta dashed line) windows are represented.

R3.3: For the figures looking at the individual frequency bands, i.e. Fig 2E, 2F, it might be useful to flip the y-axes so that low frequency bands are at the bottom and high at the top. This is both more intuitive and aligns better with the time frequency maps.

We thank the reviewer for this suggestion. We flip the frequency axes.

R3.4: Line 141-143 discusses that low frequencies have longer t-SPCs than higher. This is, perhaps, not unexpected. Low frequencies inherently last longer per cycle than high. So a spike-entrained event on a given cycle should show longer effects. This is why wavelet-type time-frequency analyses are typically done rather than just short-time fourier transforms.

We agree with the reviewer that low frequency oscillations can facilitate the coordination of longer events of spike-phase coupling. To mitigate this inherent bias, we ran a permutation test that compares the actual pairwise-phase consistency value, i.e., strength of spike-phase coupling, to a proper null distribution (see Methods).

R3.5: Lines 153-159/Fig S5. I am a bit confused about the comparisons across different frequencies. Previous analyses were done at the unit-ECoG pairing level, this seems to be at the unit level. So is this comparison looking at t-SPC similarity across all pairs associated with a given unit, or is this across all repetitions for a given pair (as suggested by line 153). The number of data points in Fig 2G suggests pairs. Also for these comparisons, were analyses done only when both pairs had significant responses, or was an analysis done to see if a unit/pair had one significant t-SPC, what were the chances of its other SPC analyses being significant, or similar patterns without being significant (i.e. how robust are these some of the unit responses, did they only have a few signif SPCs, or if a unit had one, they were likely to have others)?

We thank the reviewer for their careful observation; we acknowledge that the initial description of Figure 2G was not entirely accurate.

In Figure 2G, we plotted the strength of SPC using z-scored pairwise-phase consistency (PPC) for each pair ($N = 19,755$). This analysis includes all pairs, allowing us to examine whether SPC strength in different frequency bands co-varies across pairs without requiring significant SPC in either frequency band individually. We amended the Figure2G description as follows:

*(G) (Top) The extent of the specificity of the t-SPC frequency band is depicted for each neuron ($N = 211$) (please refer to Methods). The pie charts depict the proportion of t-SPC frequency band for two exemplary neurons. Dark gray boxes indicate the 5th and 95th percentile of the permutation distribution. (Bottom) 2D distribution of the SPC strength expressed as PPC (z-score) across all pairs ($N = 19755$) between different frequency (left: θ vs α , center: θ vs β and right: α vs β). Colormap and contours indicate the 2D density of the scatter plot. – **Figure 2 legend, pg. 11***

To compare the SPC at different epochs of the task or frequency bands across neurons and to control for the effect of the firing rate (see **Control analysis**), we computed the ratio between the number of t-SPC events and pairs for each neuron and expressed it as a percentage across different task epochs and frequency bands (refer to new Figure S5). We termed this quantity as **t-SPC index**, and it was used for correlation analyses in which the unit of observation was the single neuron. Figure S6 (old Figure S5) examines the relationship between changes in t-SPC index (speech vs. baseline) across different frequency bands at the single neuron level (see Methods and new Figure S5).

We amended the Methods section as follows:

Spike-phase coupling at the single neuron level (t-SPC index) – *To compare the SPC at different epochs of the task or frequency bands across neurons and to control for the effect of the firing rate (see **Control analysis**), we computed the ratio between the number of t-SPC events and pairs for each neuron and expressed it as a percentage across different task epochs and frequency bands (refer to Figure S5). We termed this quantity as t-SPC index, and it was used*

*for correlation analyses in which the statistical unit of observation was the single neuron. We also assessed the extent to which neuron preferentially couple to the same frequency band. We normalized the t-SPC index across frequency bands (total sum = 1) and defined the frequency-specificity as one minus the entropy of the normalized distribution. With this definition, high (e.g., peaked distribution) and low specificity (e.g., uniform distribution) are mapped onto 1 and 0 values, respectively (see Figure 2G). – **Methods, pg. 33***

The two analyses are related and address similar questions: "How frequency-specific is SPC? Does SPC in the theta/alpha range relate to SPC in the beta range? Are changes in theta/alpha SPC correlated with changes in beta SPC?" Despite using different observation units—pair-level SPC in Figure 2G and single-unit SPC in Figure S6—the results of both approaches are consistent.

We amended the Results section as follows:

*We observed that SPC strength was frequency-band specific at the individual pair level (Figure 2G). We tested whether this specificity was also evident when we aggregated t-SPC events at the single-neuron level. In other words, we examined the relationship between frequency bands for neurons that had multiple t-SPC events: if a neuron coupled in one frequency band, was it more or less likely to couple in another band? To do this, we calculated a t-SPC index for each neuron, defined as the ratio of t-SPC events to pairs, and used an entropy-based metric to quantify the frequency specificity (please see Figure S5). In general, we employed t-SPC index for all analyses thorough the manuscript in which the neuron was the statistical unit of observation. A striking proportion of units (N = 203/211 96%, frequency-specificity ~ 0.56 higher than chance level: 0.08) was significantly specific to a frequency band. Finally, neurons that exhibited increased θ - α t-SPC index did not show any modulation of β SPC index, and vice versa (Figure S6). – **Results, pg. 8***

The t-SPC index has been used for any analysis in which the statistical unit of observation was the single neuron, e.g., when we compared spike-phase coupling with the firing rate. We further repeated this concept by amending the Results section as follows:

*For most of these analyses we used the t-SPC index - the ratio of the number of t-SPC events to all possible pairs at the single neuron level - as a SPC measure to correlate with other single neuron properties such as firing rate (see Figure S5). – **Results, pg. 16***

R3.6: Fig S6, would be useful to sees these analyses in a frequency band-specific manner. Also what is the difference between the top and bottom plots?

We thank the reviewer for their suggestion. We have revised Figure S7 (old Figure S6) to include additional panels showing the analyses for each specific frequency band. These analyses aim to further validate the robustness of our SPC metric.

In the first analysis (Figure S7A), we required that each t-SPC event contain at least two cycles of oscillation at the centroid frequency to be considered reliable, effectively excluding brief, noise-driven clusters. Over 99% of t-SPC events across all frequency bands meet this criterion (8 cycles on average), with only about 2% of clusters in lower frequencies (e.g., theta and alpha) spanning fewer than two cycles, thus reinforcing the robustness of our results. During the revision of the code, we found an error on the calculation of the number of cycles. We apologize for this oversight and now we report the correct value. However, this does not change the validity of our results.

In the second analysis (Figure S7B), we recognized that the surrogate SPC maps generated during the permutation procedure might display surrogate t-SPC events due to the clustering tendencies in small samples of a random distribution. To address this, we z-scored the surrogate

PPC maps and applied the same cluster-based permutation test, defining surrogate t-SPC events using the same criteria as the original t-SPC events. We then compared the number of observed t-SPC events in the actual data to the surrogate t-SPC events. Across all frequency bands — and when examined by frequency band — the observed number of t-SPC events consistently exceeded what would be expected by chance.

Figure S7: Robustness analysis of the t-SPC event identification. (A) Distribution of the number of cycles that span the duration of the t-SPC events in all bands and grouped by frequency band. We also reported the percentage of t-SPC events spanning less than 2 cycles (see Methods). (B) Distribution of the number of t-SPC events observed in the permuted SPC maps. Red dashed line depicts the median of the distribution. All bands (gray), θ (red), α (dark orange), β (light orange), γ_L (green) and γ_H (blue). We applied the t-max correction across frequency bands in each panel to control for multiple comparisons.

We amended the Results section as follows:

*We used two cycles as the lower bound for a well-defined spike-phase coupling event, as is commonly chosen in local field potential oscillatory base analyses for β bursts (Tinkhauser et al., 2018; Vissani et al., 2021). 99% of t-SPC events had more than two cycles (on average 8 cycles). There were approximately ten times more t-SPC events in the actual SPC maps compared to the shuffled SPC maps ($p_{perm} < 0.001$, permutation test across all frequency bands) (Figure S7). These control analyses suggests that t-SPC events reflect genuine, physiological SPC mechanisms. – **Results, pg. 8-9***

R3.7: Fig S7B. I am a bit confused on what is being plotted on the bottom. What is meant by relative occurrence of theta-alpha and Beta events?

In Figure S17B (old Figure S7), we aimed to explore the temporal relationship between the directionality of spike-phase coupling (as indicated by the sign of the time delay) and the frequency specificity of spike-phase coupling events. The time delay (represented by black dots) reflects whether cortical oscillations lead STN spikes (positive value) or vice-versa (negative value). The green line represents the relative proportion of t-SPC events in the theta/alpha and beta range and it is quantified as a contrast $(A-B)/(A+B)$, where:

- **A** represents the time occurrence of spike-phase coupling in the theta/alpha frequency band (red and orange curves in Figure 2F)
- **B** represents the time occurrence of spike-phase coupling in the beta frequency band (yellow curve in Figure 2F)

Please see the “**Time occurrence**” section in Methods for further details.

This contrast ranges from -1 to +1, indicating the dominant frequency band: a value closer to -1 indicates that coupling occurs mainly in the beta frequency band, while a value near +1 indicates that coupling happens mainly in the theta/alpha band.

Our analysis revealed two distinct patterns of information flow between cortical oscillations and STN neurons:

1. Cortical oscillations tend to lead STN spike timing (positive slope) during spike-phase coupling events in the beta frequency band, particularly during the beta rebound phase following speech termination.
2. STN spike timing tends to lead cortical oscillations (negative slope) during spike-phase coupling events in the theta/alpha frequency range.

For further details, please refer to Sharott et al. (Sharott et al., 2018).

We have revised the Methods section as follows:

Time delay analysis – As STN neurons often lock to cortical signals within a narrow frequency range, power-based estimates of time delay between STN and cortex might be suboptimal¹. We calculated time delays using the phase-based analysis, as described in (Sharott et al., 2018). First, we computed the mean preferred phase of units that were significantly locked in each frequency bin (5-30 Hz range). We then averaged these phases to obtain a grand average phase for each frequency bin. By analyzing the gradient of these phases, we determined whether the ECoG channel led (positive sign) or lagged (negative sign) relative to STN neuron, and at what latency this occurred. To test the significance of the time delay, we repeated 500 times the

computation using randomly selecting mean angles from each frequency bin. To obtain a p-value, we compared the correlation coefficient in the original data and the 5th-95th percentiles of the correlation coefficient permutation distribution. In Figure S17B we calculated the relative time occurrence of t-SPC events in the theta-alpha and beta range and it is quantified as a contrast $(A-B)/(A+B)$, where A represents the time occurrence of spike-phase coupling in the theta/alpha frequency band (red and orange curves in Figure 2F) and B represents the time occurrence of spike-phase coupling in the beta frequency band (yellow curve in Figure 2F). – **Methods, pg. 34**

R3.8: Line 216,-217. I think this is a great place to tease some of the analysis methods, as I had discussed in my comments about readability. I assume when you say ‘density’ you actually mean ‘spatial density’. It would be very useful to mention here that you calculated the relative spatial density by doing blah blah (in general terms), otherwise ‘density’ is mentioned from left field, and then reviewer is not scratching their heads and then scanning through the long methods to find out what was being analyzed. Also, on lines 221-222 it says density was ‘less frequent’. I am not sure I know what that means, since I thought this was a measure of spatial position.

We completely agree with the reviewer. We now enforce the term “spatial density” and changed terminology related to time-related quantities such as “less frequent” to “dispersed” through the whole manuscript.

We amended the Results section to introduce the concept of spatial density as follows:

*We extracted two t-SPC metrics to better delineate the spectral topography of the cortico-subcortical SPC during speech production. First, we quantified the spatial density of t-SPC events as the percentage of t-SPC events to total pairs, calculated separately at both the cortical and STN levels (see Figure S5). We then compared this spatial density to a null distribution to identify ROIs with a significantly high or low prevalence of t-SPC events (Figure 3A). Second, we characterized the temporal occurrence of t-SPC events, defined as the likelihood of observing at least one t-SPC event in each STN neuron-ECOG contact pair, across time and frequency band. Similarly, we tested t-SPC temporal occurrence against a null distribution to identify significant windows of aggregation (high overlap) or dispersion (low overlap) of t-SPC events (Figure 3B, see Methods for details).– **Results, pg. 11***

R3.9: Figure S9 (and a few other places). Permutation tests were done to compare the different distributions. Was this done as a series of pairwise testing, or one large permutation, shuffling all the data together? If the former, were multi-comparisons corrections done?

We controlled the family-wise error rate for multiple comparisons only for the 2D maps (time-frequency) SPC maps by applying the cluster-based permutation test, as explained in the method sections. For single permutation tests (pairwise testing) applied over different ROIs (e.g., Figure 3A, Figure S10) or frequency bands (e.g., Figure S7, Figure S9A-B), we initially did not apply any multiple comparison correction. However, in the revised version of the manuscript, we decided to follow the reviewer’s suggestion and decided to apply the Max correction, also referred to as t-max (Blair & Karniski, 1993) or joint correction (Westfall et al., 1993), in the context of multiple pairwise tests. The validity of our results was not affected by the correction. This correction works as follows: on each permutation of the data, the test statistic is computed for each comparison and the most extreme value (either positive or negative) is taken. Repeating this procedure multiple times produces a single, more-conservative permutation distribution, against which the actual test statistic is compared. Thus, the more tests there are to take the maximum across, the more conservative the permutation distribution naturally becomes. We used the PERMUTOOLS

(Crosse et al., 2024) library which applies this correction by default when multiple tests are performed.

We amended the Methods section as follows:

*When multiple pairwise permutation tests were applied over different ROIs (e.g., Figure 3A, Figure S10) or frequency bands (e.g., Figure S7, Figure S9A-B), we control the family-wise error rate by applying the t-max correction (Blair & Karniski, 1993), also referred to as joint correction. This correction works as follows (Crosse et al., 2024): on each permutation of the data, the test statistic is computed for each comparison and the most extreme value (either positive or negative) across comparisons is taken. Repeating this procedure multiple times produces a single, more-conservative permutation distribution, against which the actual test statistic is compared. – **Methods, pg. 36***

We amended the description of the figures in which we applied the t-max correction.

R3.10: Line 257 – The principle component results are mentioned in passing, without having discussed that a PC spatial analysis was done and for what reason. This ties back to my comments about readability, and needing to introduce things a little better before giving their results.

We implemented the reviewer's suggestion by adding a paragraph to explain the rationale for using a PC-based spatial analysis within the STN before presenting the results. This STN-specific reference frame is necessary because the STN is not fully aligned with MNI coordinates. For instance, the posterior-anterior direction in the STN maps onto both the Y and Z axes in MNI space. We amended the Results section as follows:

*Since the STN is not fully aligned with the MNI coordinates, we rotated the MNI reference frame to align with the STN's principal axes or components (PC): posterior-anterior axis (PC1), dorsal-ventral axis (PC2) and medial-lateral axis (PC3) (see Figure S11A-B and Methods for details). – **Results, pg. 14***

R3.11: Figure S13 / lines 285-287. How was the mismatch between the small number of units and the larger number of t-SPC pairs handled? It would seem there would be redundancy of the unit side (i.e. each unit had several SPCs pairings).

In our analysis we enforced a minimum number of pairs for each STN neuron in order to get a reliable estimate of t-SPC prevalence across neurons. For sake of completeness, we report here the distribution of the number of pairs for each neuron (mean \pm SD: 93 ± 33 pairs, see Figure R3.1 below). We agree with the reviewer that the term "spatial-density" could be confusing since it is used in a different context, so we now refer to this measure as t-SPC index (expressed as %), which aggregates SPC maps across pairs for each neuron. Please refer to Figure S5 that we include here again for convenience. In Figure S14 (old Figure S13) we plotted the relationship between the difference of the t-SPC index in two different task epochs (baseline, speech, after speech) in each frequency band and changes in firing rate during speech, expressed as z-score with respect to the baseline. The observation unit is the single neuron.

We amended the Results section as follows:

We then asked if change in firing rate for a given neuron was associated with increased coupling, and in which frequency bands. We plotted t-SPC index changes between behavioral epochs

against z-scored firing rate modulation (with respect to baseline) during speech production (see Figure S14B and Methods). Again, t-SPC index changes were not correlated with firing rate modulation in any frequency band. – **Results, pg. 16-17**

In Methods we included the following explanation of the t-SPC index

Spike-phase coupling at the single neuron level (t-SPC index) – To compare the SPC at different epochs of the task or frequency bands across neurons and to control for the effect of the firing rate (see **Control analysis**), we computed the ratio between the number of t-SPC events and pairs for each neuron and expressed it as a percentage across different task epochs and frequency bands (refer to Figure S5). We termed this quantity as t-SPC index, and it was used for correlation analyses in which the statistical unit of observation was the single neuron. We also assessed the extent to which neuron preferentially couple to the same frequency band. We normalized the t-SPC index across frequency bands (total sum = 1) and defined the frequency-specificity as one minus the entropy of the normalized distribution. With this definition, high (e.g., peaked distribution) and low specificity (e.g., uniform distribution) are mapped onto 1 and 0 values, respectively (see Figure 2G). – **Methods, pg. 33**

Figure S5: Overview of the SPC metrics. (A) For each participant we calculated the spike-phase coupling in each possible pair (all-to-all schema, green lines) of ECoG contacts (red circle) and STN neurons (cyan circle) and obtained a time-frequency SPC map. A cluster-based permutation test was used to identify significant transient spike-phase coupling (t-SPC) events (black boxes with red contour) and binarize each SPC map. A pair was deemed to be significant if it contained one or more t-SPC events. (B) We grouped and combined these binarized SPC maps in different ways to extract different metrics: t-SPC occurrence was used to evaluate the time occurrence of t-SPC events and their degree of aggregation over time, t-SPC spatial density was used to evaluate the spatial distribution over cortex and STN (grouping by ROI or 3D sphere, refer to Methods for details), and t-SPC index was used for analyses in which the unit of observation was the STN neuron (grouping by neuron). All metrics are expressed as percentage.

Figure R3.1: Distribution of the number of pairs for each neuron.

R3.12: Also on Fig S13/S14. I am not sure I understand or can intuit what a change in spatial density means or how it was calculated. I can see a change in the fraction/number of SPCs, but since nothing is moving anatomically, are we really talking about a change in 'density' or just frequency of responsive events. For example, how can a neuron have a higher density, I thought that was a spatial measure (line 302-303)?

We apologize for any confusion. For the meaning of Figure S14 (old Figure S13) please refer to reply to **R3.11**.

In Figure S16D (old Figure S14D) we reported the spatial density of the spike-phase coupling metric across regions of interest according to the Destrieux atlas (Destrieux et al., 2010) (the same as Figure 3A) for each neuron type. In this case the observation of unit is the t-SPC event across neurons.

R3.13: It would be interesting to discuss the different t-SPC responses across areas by unit firing type (Fig S14). Is there any intuition to be gained from this analysis about specific modes of communication for a neuron that has increased vs decrease firing? They do seem to have somewhat different t-SPC bands.

As the reviewer correctly observed, Figures S15C and Figure S16 highlight that the direction of firing rate modulation shapes the frequency profile of spike-phase coupling. Specifically, neurons with a decreasing firing rate engage SPC at low frequencies (theta-alpha) (low-pass coupling profile), while neurons with an increasing firing rate show SPC at higher frequencies (alpha-beta, bandpass profile), with reduced coupling at very low frequencies.

To our knowledge, few studies have explored in depth the spike-phase coupling properties of neurons in response to global activity shifts associated with various behavioral states. A notable example is a 2012 study, which used dynamic-clamp experiments to replicate in vivo-like conditions in hippocampal pyramidal neurons in rats (Broicher et al., 2012). This study investigated how the frequency profile of spike-phase coupling varied under different experimental conditions, such as conductance state and input firing rate. Remarkably, they found that neurons in low-conductance states with lower firing rates displayed a low-pass coupling profile, while neurons with higher firing rates and higher conductance states exhibited a bandpass coupling profile.

The mechanisms behind these differences are thought to involve spike rate adaptation, which adjusts the input-output gain (current-voltage relationship) and functions as a high-pass filter, as predicted by theoretical studies (Benda & Herz, 2003). Additionally, frequency resonance within the spike-generating mechanism contributes to shaping the coupling profile. The extent to which these factors—spike rate adaptation and frequency resonance—respond to changes in firing rate and conductance seems to account for the variation in spike-phase coupling profiles.

Collectively, these findings suggest that the ability of neurons to synchronize with network rhythms of specific frequencies — the difference we observed in SPC profiles between neurons with decreasing versus increasing firing rates — depends both on the overall network state that the neurons experience and on intrinsic properties such as spike rate adaptation. This implies that variations in SPC frequency profiles could be linked to the dynamic conditions within the neural network and the adaptive characteristics of individual neurons.

We included a paragraph in the Results as follows:

*Next, we compared the centroid duration and frequency of t-SPC events across these firing rate categories. Neurons with mixed firing rates (~0.31 s) and decreased firing rates (~0.31 s) had longer t-SPC events, with median centroid frequencies of 18 Hz and 20 Hz, respectively (Figure S15B-C). Both the group-level SPC maps and t-SPC event analyses revealed that only neurons with decreased firing rates significantly contributed to θ t-SPC events during speech production (Figure S16A-C). In contrast, neurons with either decreased or increased firing rates exhibited similar profiles for α t-SPC events. θ - α t-SPC events occurred less frequently during auditory cue presentation and more frequently during speech production. Notably, neurons with increased or mixed firing rate modulation predominantly contributed to the aggregation of β t-SPC events during the rebound phase. Only neurons with mixed firing rate modulation showed a significant aggregation of β t-SPC events during the baseline period. In summary, neurons with decreasing firing rates exhibited t-SPC events dominated by θ and α rhythms, while neurons with increasing firing rates showed t-SPC events in the α - β range. This “band-pass” profile was particularly narrowband in β for neurons with mixed firing rate modulation. No distinct pattern of t-SPC coupling was observed in neurons without firing rate modulation at any level of analysis (Figure S16A-C). These results suggest that the pattern of firing rate modulation in STN neurons — whether increasing, decreasing or mixed — affects the frequency specificity of speech-related phase-of-firing coding. – **Results, pg. 17***

We included a paragraph in the Discussion as follows:

STN neurons were preferentially coupled to a single frequency band which was significantly explained by the pattern of firing rate modulation in STN neurons. STN neurons with decreasing firing rates exclusively drove θ SPC. In a previous study (Lipski et al., 2018) we found these neurons to be temporally locked to the onset of the auditory cue. Conversely, neurons with increasing firing rates displayed SPC in β , which was notably narrowband around 17 Hz in neurons with mixed firing rate dynamics. The phenomenon of spike-phase coupling in response to behavioral state transitions and activity shifts remains relatively underexplored. Broicher and colleagues (Broicher et al., 2012) utilized dynamic-clamp experiments to replicate in vivo-like conditions in hippocampal pyramidal neurons, showing that SPC frequency profiles are modulated by conductance states and input firing rates. Specifically, neurons in low-conductance states with reduced firing rates exhibited low-pass coupling, whereas neurons in high-conductance states with elevated firing rates displayed bandpass coupling. These differences can be attributed to mechanisms such as spike rate adaptation, which modulates the input-output gain (current-voltage relationship) and functions as a high-pass filter (Benda & Herz, 2003), as well as frequency resonance intrinsic to the spike-generation process. Our findings suggest that similar mechanisms underlie SPC dynamics in the STN. These results underscore the dynamic nature of phase-of-firing coding within the STN, driven by a complex interplay of neural network states and the intrinsic adaptive properties of individual neurons during speech-related tasks. – Discussion, pg. 22

R3.14: Figure 5, it seems like the accuracy effects extend very early into the auditory bins, which seems strange to me. Was any analysis done to associate responses with specific words/phonemes being tested? How do we interpret the fact that the primary differences between error and non-error trials are really in the auditory testing (or even earlier) and really go away once speech production actually begins. That seems strange to me, almost suggests more of a sensory/perceptual error than a motor one. It is difficult to tell since there is not a longer delay period between stimulus presentation and repetition.

Based on the timing of the theta-alpha SPC difference between accurate and error trials (in particular Figure 5B), the reviewer is suggesting that the error trials be the result of auditory perceptual errors rather than motor planning errors. We first addressed auditory perceptual errors from a behavioral perspective, and then addressed the reviewer's question about how to interpret the timing of theta-alpha SPC.

Error trials were defined by the production of at least one off-target phoneme in any of the syllables. If the off-target productions were phonetically/acoustically similar to target production, we reasoned that they might be perceptual errors. For example, if the stimulus was "gi ta su" and the patient produced "ki ta su", it's very likely that they misheard the /g/ for a /k/. However, analysis of behavioral data reveals that the off-target productions were not similar to the target production. Figure S1 shows the patterning of speech errors. The most common error across participants was /g/ changing to /v/. /g/ is velar plosive, while /v/ is a labiodental fricative. They are perceptually dissimilar, and therefore unlikely to be confused. The same is true for three of the four other most common errors. Therefore, it is unlikely that production errors were the result of perceptual errors in the auditory epoch.

The theta-alpha SPC differences between accurate and error trials do, as the reviewer remarks, begin in the auditory epoch of the trial. We considered three possible interpretations of this.

- 1) The theta-alpha signal is a readout of uncertainty in speech planning, which begins during the auditory epoch. We can think of this from a dynamical systems perspective, where the motor program sequence that has been selected is the 'best guess' of the speech motor

planning system, but it's not the correct well in the manifold of all possible motor sequences.

- 2) The theta-alpha signal is an error signal that's computed on the efferent copy of the motor program that's being held in working memory.
- 3) Error trials were driven by poor working memory. Here, we suggest that the speech sequence was correctly perceived in all trials, initially resulting in a correct target production. Then, in error trials, the phonological working memory system fails to preserve the speech sequence. Lower theta-alpha SPC in error trials could be related to a slip of phonological working memory. We remain agnostic as to whether theta-alpha SPC differences are a simple index of poor phonological working memory, or instead the cause of poorer working memory.

Explanation 3) seems most likely. If the error trials were driven by poor phonological working memory, the theta-alpha SPC events should be with cortical regions associated with working memory. Indeed, we observe theta-alpha SPC most prominently with the inferior parietal cortex, a region that is well known to subserve phonological working memory (Jonides et al., 1998; Paulesu et al., 1993).

We amended the Discussion as follows:

We uncovered a neural correlate of speech errors in our syllable repetition task: delayed, lower θ - α t-SPC between STN and SMG-pSTG (posterior STG) (Figure 5). Here we discuss two possible interpretations of how the θ - α SPC differences relate to the speech errors. One possibility is that the errors were related to phonological working memory (PWM)(Baddeley, 1992). We defined error trials as those in which at least one off-target phoneme was produced. Participants frequently made substitution errors in which the off-target phoneme was perceptually dissimilar to the target phoneme (Figure S1); production errors were thus unlikely the result of perceptual errors. Instead, the errors may be rooted in the failure of PWM to maintain the proper sequence in memory until speech production. Additionally, θ - α SPC significant differences appear in the second half of the auditory window (Figure 5B), when we would expect a reliance on PWM to maintain the syllable sequence. θ - α SPC was observed predominantly in cortical regions that have long been implicated in PWM: the inferior parietal cortex and adjacent regions in pSTG (Figure 3)(Jonides et al., 1998; Paulesu et al., 1993). θ SPC has been documented as a mechanism subserving working memory(Lee et al., 2005), lending further credibility to the PWM account of the speech errors.

Another possibility is that lower θ - α SPC in error trials is related to auditory-motor integration, or the interface between auditory input and motor programs in the speech production system(Hickok, 2012). Because our auditory stimuli were phonotactically legal but meaningless, participants could not rely on lexical or semantic anchors to remember the verbal sequences. Participants would instead have to rely heavily on the 'dorsal stream' of auditory processing in speech in the dual-stream model of speech processing(Hickok & Poeppel, 2007). The dorsal stream translates from sensory information to a motor encoding. The neurobiological cornerstone of the dorsal stream is situated just adjacent to the SMG-pSTG complex we identified in this study, in an area referred to as 'Spt' (Sylvian-parieto-temporal). Spt, at the parieto-temporal boundary in and around the posterior Sylvian fissure, has been extensively studied for its sensorimotor properties(Awh et al., 1996; Behroozmand et al., 2018; Buchsbaum, Olsen, Koch, & Berman, 2005; Buchsbaum, Olsen, Koch, Kohn, et al., 2005). Spt is critical for auditory repetition as it is hypothesized to compute a 'coordinate transform' from auditory to motor space(Hickok, 2012). Lesions to this area can cause conduction aphasia—the selective deficit of verbatim repetition, despite fluent spontaneous speech and intact language comprehension(Acharya et al., 2024; Benson et al., 1973). Here, we

find evidence that Spt and adjacent regions in the posterior perisylvian cortex might achieve this well-established auditory-motor interfacing by recruiting the BG, and specifically by leveraging θ - α coupling with STN. – Discussion, pg. 23

R3.15: Also in Figure 5, why are there so few neurons for this analysis? Only about a third of the whole paper's units are included in 5A, and even less in 5D/E.

In **Figure 5** (error analysis), we restricted our analysis to include only significant ECoG-neurons pairs (from the main analysis in the low-frequency range of 4-40 Hz) where each condition contained ≥ 20 trials. This resulted in 46 neurons and 827 pairs across 18 participants. The motivation for this decision was two-fold: First, we aimed to focus on pairs that exhibited significant spike-phase coupling (SPC) in frequency bands that showed modulation during speech, specifically below the low gamma range. Second, this approach helped reduce the computational load of the analysis, bringing it from over 19,000 pairs to approximately 800 pairs. Please refer to **Table S5** for details on the coverage of both the main and error analyses in terms of the number of pairs, participants, STN neurons, and ECoG contacts. For sake of completeness, we have also included Table S3 that contains the same information for the main analysis.

Furthermore, in **Figure 5D/E**, the number of units included in the statistical tests is further reduced (Figure 5D: N=20/46 from 13 participants; Figure 5E: N=17/46 for left hemisphere and N=8/46 for right hemisphere). This is because we only included neurons that showed significant SPC in the theta-alpha range during both accurate and error trials.

Table S5: Number of pairs, participants, STN neurons and ECoG contacts included in the error analysis (refer to Methods for details) for each ROI.

ROI	PreCG	MFG	SMG	STG	PostCG	SCG	pars O.
# pairs	126	84	169	117	201	66	0
# participants	13	8	11	9	13	7	0
# STN neurons	15	14	14	12	13	8	0
# ECoG contacts	38	41	47	41	50	31	0

R3.16: I am not sure I completely by the conclusion that these results suggest some sort of error monitoring/generation, since there is little difference in SPCs during the speaking period, when the speech error is actually happening. There is a little difference after speech offset in the theta-alpha band, but this is rather late given what we know about error-related signals for speaking happening during actual production, at least for sensory cortex. I guess there could be some sort of very late linguistic error detection that is delayed, but it still doesn't fit with most linguistic error models which posit a more on-line monitoring function (perhaps delayed for segmental speech parameters, but shouldn't only be occurring well after speech stops).

As discussed in our response to point R3.14, the points raised here have already been addressed.

R3.17: Were any other parameters of speech/voice examined beyond phonemic errors? Parkinson's has known effects on pitch and loudness, which are also known to improve with DBS as discussed.

We thank the reviewer for this input, and we think that this is a great idea for further research. Indeed, our group is currently investigating STN encoding of pitch and loudness in a separate manuscript.

R3.18: Methods – I need more details on the time warping procedure used. All the plots seem to show a fixed onset/offset of sound presentation and speech. The former is obviously easier to design, but variable speech production makes the latter more variable. How did you do a time warp without affecting the frequency contents of the EcOG osc¹²⁵illations? Time warping has been done on spike trains in the past, which should still preserve relative firing rates (mostly – they can cause distortions, but usually works out on average), but continuous time signals are far more susceptible to distortion and frequency shifts with compression or expansion.

We thank the reviewer for raising this point. It is unclear whether the reviewer is referring to the time-warping algorithm used to synchronize different data streams or the variable-window approach used for calculating time-resolved spike-phase coupling.

First, to clarify the synchronization procedure: the linear time-warping algorithm aligns the three different data streams—Ripple (ECoG), Neuro-Omega (MERs), and Zoom-H6 (audio signals)—and introduces a warp factor γ (see the new Methods section “Electrophysiological data alignment” for details on γ). This factor differs from unity by about one part in 10^5 , corresponding to a correction of 1 ms every 100 seconds. This was consistent across participants and file types. Sub-millisecond precision was achieved, where a 1 ms mismatch results in a phase shift of 3% in the high-beta range and 10% in the high-gamma range.

Second, to provide comparable estimates of spike-phase coupling strength around key trial events (e.g., 0.75 s before auditory cue, auditory cue onset/offset, speech production onset/offset and 0.75 s after speech production offset), we used an event-locked, variable-window-width approach. The time intervals between neighboring task events were divided into equidistant bins (please refer to Figure R3.2 for an illustration of the Method and Supplementary Text for further details amended in the revised version of the manuscript). This ensures that coupling strength estimates around all key events are unbiased, instead of being locked to just one event like auditory cue or speech production onset.

Figure R3.2: Illustration of the variable-width window method for SPC computation. In this example we illustrate the implementation of the SPC pipeline in a subset of trials ($N = 20$) of a neuron with decreasing firing rate during the speech production window. For sake of clarity, we only show two anchor bins between two contiguous events (0.75 s before auditory cue onset (blue), auditory cue onset (orange), auditory cue offset (yellow), speech production onset (purple), speech production offset (green), 0.75 seconds after speech production offset (cyan),

Importantly, the alignment of the time axis relative to specific events (e.g., auditory cue or speech onset) in the spike-phase coupling maps (e.g., Figure 2B-C) occurs *after* calculating the coupling metrics and does not affect single-pair coupling estimates. Single-pair metrics, such as frequency centroid or onset/offset times, are independent of the specific event used for plotting. For group-level analyses, such as those shown in Figure 2A, slight differences might emerge when averaging maps, but these discrepancies remain qualitatively small (please refer to Figure R3.3)

Figure R3.3: Effect of event-locking to the average SPC map. Average of SPC maps across all the significant pairs (N=2148) after locking at the auditory cue onset (Left) or speech production onset (Right, same as Figure 2A).

Minor Point:

R3.19: Figure S2 is cut off, I only see 1.5 panels

We apologize for this oversight during the PDF conversion. We fixed the visualization of Figure S2.

R3.20: Figure 1D, I need more details of what is being plotted. What does the shading correspond to? I assume the error bars are SEM (text sort of hints at that, but doesn't explicitly say that)

The shaded area represents the SEM of the normalized firing rate (i.e., percentage change) across trials while the shading highlights time bins with significant firing rate modulation with respect to baseline. As suggested, we revised the description of Figure 1D to make explicitly the meaning of the shading in the plot.

Figure 1: Quantification of STN-cortex spike-phase coupling during an intraoperative syllable triplet repetition task. (A) Illustration of the syllable triplet repetition task. Participants were instructed to repeat unique consonant-vowel ("CV") syllable triplets (magenta). The auditory stimuli were presented through earphones (black). High-density electrocorticography (ECoG)

strips were placed in auditory and sensorimotor areas through the burr hole (cyan). Microelectrode recordings were acquired in the subthalamic nucleus during functional mapping (purple). Spectrograms of the audio signals are shown. **(B)** Timing of behavioral events, relative to speech-onset. Heatmap of the duration of the auditory cue (AC) and speech production (SP) windows expressed as percentage across trials for each participant. Average phonetic accuracy of the produced syllables for each participant are shown on the right. **(C)** ECoG strips localizations. The coverage of the ECoG strips across participants is superimposed on three different target areas from the Destrieux atlas (Destrieux et al., 2010): postcentral gyrus (purple), inferior frontal gyrus (blue) and superior temporal gyrus (green). Exemplary auditory-locked and speech-locked spectrograms of activity in the postcentral gyrus (purple sphere) and superior temporal gyrus (green sphere) after normalization with respect to the baseline are displayed. **(D)** MER localization. Coverage of single units across participants is depicted in grayscale on the STN surface. Spheres denote the location of four exemplary neurons with different categories of instantaneous firing rate (IFR) modulation: Increasing (red), Decreasing (blue), Mixed (green) and No (gray) firing rate modulation. The plots show the percentage change in instantaneous firing rate (IFR) relative to baseline during the speech production window (indicated by the magenta vertical dashed line). The horizontal black line represents the mean IFR across trials, while the gray shaded area denotes the standard error of the mean (SEM). Colored patches highlight time bins with significant firing rate modulation (refer to Methods for details). **(E)** Exemplary transient spike-phase coupling in the a range during the speech production window (magenta dashed line). Spike timestamps, a oscillations, and instantaneous phase are illustrated. Magenta bars delineate the duration of the syllable triplet. List of abbreviations: ITI (inter-trial interval) used as baseline, AC (auditory cue), SP (speech production) and IFR (instantaneous firing rate). – **Figure 1 legend, pg. 6**

R3.21: I don't see Figure 1E mentioned in the text (unless there is typo on line 108 where it cites Figure 1D).

We fixed it.

R3.22: Nit picky, but at times the manuscript uses too many digits of precision that are likely not meaningful (i.e. neurons having average 92.78 LFP pairs, I don't think that additional precision adds much). Not a big deal, though.

We followed the reviewer's suggestion rounding to the closest integer all integer quantities (number of pairs) and percentages of neurons. We kept two digits of precisions for all the other quantities.

R3.23: Fig 2G, could you please title/label the three different plots

We added a title for each plot in Figure 2G as suggested by the reviewer. We will include here Figure 2G for sake of convenience.

Figure 2: STN neurons show transient task-related coupling to the cortex in either θ - α or β bands.

(A) Average of the spike-phase coupling (SPC) maps with significant spike-phase coupling (N = 2148 pairs). The pairwise-phase consistency (PPC) index is compared to the permutation distribution and expressed as z-score. Group-level statistical test (t-stat) of the significance of the z-score PPC with respect to the baseline across all the significant pairs. Red and blue lines contour regions of significant SPC increase or decrease, respectively. (B) Examples of single-pair SPC maps showing that STN neurons preferentially **locked to** cortical phases only during brief and transitory episodes. (C) Definition of the transient SPC event (t-SPC event) in a single-pair SPC map. We calculated the onset and offset times, temporal duration, and the frequency centroid for each t-SPC event. Most of pairs exhibit only one t-SPC event, as shown by the barplot. The inset plot depicts an exemplary SPC map with two t-SPC events in the same frequency band. (D) Distribution of the t-SPC duration and t-SPC frequency centroid. To augment the readability of the t-SPC frequency distribution, we adopted a logarithmic scale. Red dashed line depicts the median of the distribution. (E) List of the t-SPC events (N = 2987) ordered by frequency centroid. (F) t-SPC events occurrence grouped by frequency band. Shaded areas illustrate the 5th and 95th percentiles of the permutation distribution for the aggregation test. (G) (Top) The extent of the specificity of the t-SPC frequency band is depicted for each neuron (N = 211) (please refer to Methods). The pie charts depict the proportion of t-SPC frequency band for two exemplary neurons. Dark gray boxes indicate the 5th and 95th percentile of the permutation distribution. (Bottom) 2D distribution of the SPC strength expressed as PPC (z-score) across all pairs (N = 19755) between different frequency bands (left: θ vs α , center: θ vs β and right: α vs β). Colormap and contours indicate the 2D density of the scatter plot. Red dashed line depicts the median of the distribution. Black and purple dashed lines denote auditory cue and speech production windows. θ (red), α (dark orange), β (yellow), γ L (green) and γ H (blue).

R3.24: I can't find Figure S12 referenced anywhere in them manuscript.

Figure S11A-B (old Figure S12) is now correctly referenced.

*The principal component coordinates (PC1: antero-posterior direction, PC2: dorso-ventral direction and PC3: medio-lateral direction) represents a more suitable reference of frame, as the STN is not fully spatially aligned with the MNI coordinates (Figure S11A-B). Spatial density computation was repeated for each of the three principal axes using a side of 0.8 mm. – **Methods, pg. 33***

*Since the STN is not fully aligned with the MNI coordinates, we rotated the MNI reference frame to align with the STN's principal axes or components (PC): posterior-anterior axis (PC1), dorsal-ventral axis (PC2) and medial-lateral axis (PC3) (see Figure S11A-B and Methods for details). – **Results, pg. 14***

R3.25: Line 552 – do you really mean 'sub-millimetric' precision or sub-millisecond?

As noted by the reviewer the correct word is sub-millisecond. We fixed it.

References

- Acharya, A. B., Lui, F., & Maani, C. V. (2024). Conduction Aphasia. In *StatPearls*. StatPearls Publishing. <http://www.ncbi.nlm.nih.gov/books/NBK537006/>
- Alexander, G. E., DeLong, M. R., & Strick, P. L. (1986). Parallel Organization of Functionally Segregated Circuits Linking Basal Ganglia and Cortex. *Annual Review of Neuroscience*, 9(1), 357–381.
<https://doi.org/10.1146/annurev.ne.09.030186.002041>
- Averna, A., Debove, I., Nowacki, A., Peterman, K., Duchet, B., Sousa, M., Bernasconi, E., Alva, L., Lachenmayer, M. L., Schuepbach, M., Pollo, C., Krack, P., Nguyen, T. K., & Tinkhauser, G. (2023). Spectral Topography of the Subthalamic Nucleus to Inform Next-Generation Deep Brain Stimulation. *Movement Disorders*, mds.29381. <https://doi.org/10.1002/mds.29381>
- Awh, E., Jonides, J., Smith, E. E., Schumacher, E. H., Koeppel, R. A., & Katz, S. (1996). Dissociation of Storage and Rehearsal in Verbal Working Memory: Evidence From Positron Emission Tomography. *Psychological Science*, 7(1), Article 1.
<https://doi.org/10.1111/j.1467-9280.1996.tb00662.x>
- Baddeley, A. (1992). Working Memory. *Science*, 255(5044), Article 5044.
- Behroozmand, R., Johari, K., Kelley, R. M., Kapnoura, E. C., Narayanan, N. S., & Greenlee, J. D. W. (2019). Effect of deep brain stimulation on vocal motor control mechanisms in Parkinson's disease. *Parkinsonism & Related Disorders*, 63, 46–53. <https://doi.org/10.1016/j.parkreldis.2019.03.002>
- Behroozmand, R., Phillip, L., Johari, K., Bonilha, L., Rorden, C., Hickok, G., & Fridriksson, J. (2018). Sensorimotor impairment of speech auditory feedback

processing in aphasia. *NeuroImage*, 165, 102–111.

<https://doi.org/10.1016/j.neuroimage.2017.10.014>

Benda, J., & Herz, A. V. M. (2003). A Universal Model for Spike-Frequency Adaptation. *Neural Computation*, 15(11), 2523–2564.

<https://doi.org/10.1162/089976603322385063>

Benson, D. F., Sheremata, W. A., Bouchard, R., Segarra, J. M., Price, D., & Geschwind, N. (1973). Conduction Aphasia: A Clinicopathological Study. *Archives of Neurology*, 28(5), Article 5.

<https://doi.org/10.1001/archneur.1973.00490230075011>

Blair, R. C., & Karniski, W. (1993). An alternative method for significance testing of waveform difference potentials. *Psychophysiology*, 30(5), 518–524.

<https://doi.org/10.1111/j.1469-8986.1993.tb02075.x>

Bohland, J. W., Bullock, D., & Guenther, F. H. (2010). Neural representations and mechanisms for the performance of simple speech sequences. *Journal of Cognitive Neuroscience*, 22(7), Article 7. <https://doi.org/10.1162/jocn.2009.21306>

Bouchard, K. E., Mesgarani, N., Johnson, K., & Chang, E. F. (2013). Functional organization of human sensorimotor cortex for speech articulation. *Nature*, 495(7441), 327–332. <https://doi.org/10.1038/nature11911>

Brabenec, L., Klobusiakova, P., Barton, M., Mekyska, J., Galaz, Z., Zvoncak, V., Kiska, T., Mucha, J., Smekal, Z., Kostalova, M., & Rektorova, I. (2019). Non-invasive stimulation of the auditory feedback area for improved articulation in Parkinson's disease. *Parkinsonism & Related Disorders*, 61, 187–192.

<https://doi.org/10.1016/j.parkreldis.2018.10.011>

Brabenec, L., Klobusiakova, P., Simko, P., Kostalova, M., Mekyska, J., & Rektorova, I.

(2021). Non-invasive brain stimulation for speech in Parkinson's disease: A randomized controlled trial. *Brain Stimulation*, *14*(3), Article 3.

<https://doi.org/10.1016/j.brs.2021.03.010>

Broicher, T., Malerba, P., Dorval, A. D., Borisjuk, A., Fernandez, F. R., & White, J. A.

(2012). Spike phase locking in CA1 pyramidal neurons depends on background conductance and firing rate. *The Journal of Neuroscience: The Official Journal of the Society for Neuroscience*, *32*(41), 14374–14388.

<https://doi.org/10.1523/JNEUROSCI.0842-12.2012>

Bronfeld, M., & Bar-Gad, I. (2011). Loss of specificity in Basal Ganglia related

movement disorders. *Frontiers in Systems Neuroscience*, *5*, 38.

<https://doi.org/10.3389/fnsys.2011.00038>

Brown, P., Oliviero, A., Mazzone, P., Insola, A., Tonali, P., & Di Lazzaro, V. (2001).

Dopamine dependency of oscillations between subthalamic nucleus and pallidum in Parkinson's disease. *The Journal of Neuroscience: The Official Journal of the Society for Neuroscience*, *21*(3), 1033–1038.

Buchsbaum, B. R., Olsen, R. K., Koch, P., & Berman, K. F. (2005). Human Dorsal and

Ventral Auditory Streams Subserve Rehearsal-Based and Echoic Processes during Verbal Working Memory. *Neuron*, *48*(4), Article 4.

<https://doi.org/10.1016/j.neuron.2005.09.029>

Buchsbaum, B. R., Olsen, R. K., Koch, P. F., Kohn, P., Kippenhan, J. S., & Berman, K.

F. (2005). Reading, hearing, and the planum temporale. *NeuroImage*, *24*(2),

Article 2. <https://doi.org/10.1016/j.neuroimage.2004.08.025>

- Bush, A., Zou, J. F., Lipski, W. J., Kokkinos, V., & Richardson, R. M. (2024). Aperiodic components of local field potentials reflect inherent differences between cortical and subcortical activity. *Cerebral Cortex*, *34*(5), bhae186.
<https://doi.org/10.1093/cercor/bhae186>
- Chen, K.-H. S., & Chen, R. (2019). Invasive and Noninvasive Brain Stimulation in Parkinson's Disease: Clinical Effects and Future Perspectives. *Clinical Pharmacology & Therapeutics*, *106*(4), Article 4. <https://doi.org/10.1002/cpt.1542>
- Chrabaszcz, A., Neumann, W.-J., Stretcu, O., Lipski, W. J., Bush, A., Dastolfo-Hromack, C. A., Wang, D., Crammond, D. J., Shaiman, S., Dickey, M. W., Holt, L. L., Turner, R. S., Fiez, J. A., & Richardson, R. M. (2019). Subthalamic Nucleus and Sensorimotor Cortex Activity During Speech Production. *The Journal of Neuroscience*, *39*(14), 2698–2708. <https://doi.org/10.1523/JNEUROSCI.2842-18.2019>
- Ciabarra, A. M. (2000). Subcortical infarction resulting in acquired stuttering. *Journal of Neurology, Neurosurgery & Psychiatry*, *69*(4), 546–549.
<https://doi.org/10.1136/jnnp.69.4.546>
- Crone, N. E., Sinai, A., & Korzeniewska, A. (2006). High-frequency gamma oscillations and human brain mapping with electrocorticography. *Progress in Brain Research*, *159*, 275–295. [https://doi.org/10.1016/S0079-6123\(06\)59019-3](https://doi.org/10.1016/S0079-6123(06)59019-3)
- Crosse, M. J., Foxe, J. J., & Molholm, S. (2024). *PERMUTOOLS: A MATLAB Package for Multivariate Permutation Testing* (Version 1). arXiv.
<https://doi.org/10.48550/ARXIV.2401.09401>

- Dastolfo-Hromack, C., Bush, A., Chrabaszcz, A., Alhourani, A., Lipski, W., Wang, D., Crammond, D. J., Shaiman, S., Dickey, M. W., Holt, L. L., Turner, R. S., Fiez, J. A., & Richardson, R. M. (2022). Articulatory Gain Predicts Motor Cortex and Subthalamic Nucleus Activity During Speech. *Cerebral Cortex*, 32(7), 1337–1349. <https://doi.org/10.1093/cercor/bhab251>
- Deng, S., Dong, Z., Pan, L., Liu, Y., Ye, Z., Qin, L., Liu, Q., & Qin, C. (2022). Effects of repetitive transcranial magnetic stimulation on gait disorders and cognitive dysfunction in Parkinson's disease: A systematic review with meta-analysis. *Brain and Behavior*, 12(8), Article 8. <https://doi.org/10.1002/brb3.2697>
- Destrieux, C., Fischl, B., Dale, A., & Halgren, E. (2010). Automatic parcellation of human cortical gyri and sulci using standard anatomical nomenclature. *NeuroImage*, 53(1), 1–15. <https://doi.org/10.1016/j.neuroimage.2010.06.010>
- Dromey, C., & Bjarnason, S. (2011). A Preliminary Report on Disordered Speech with Deep Brain Stimulation in Individuals with Parkinson's Disease. *Parkinson's Disease*, 2011, 1–11. <https://doi.org/10.4061/2011/796205>
- Duffy, J. R. (2020). *Motor speech disorders: Substrates, differential diagnosis, and management* (Fourth edition). Elsevier.
- Enard, W., Gehre, S., Hammerschmidt, K., Hölter, S. M., Blass, T., Somel, M., Brückner, M. K., Schreiweis, C., Winter, C., Sohr, R., Becker, L., Wiebe, V., Nickel, B., Giger, T., Müller, U., Groszer, M., Adler, T., Aguilar, A., Bolle, I., ... Pääbo, S. (2009). A Humanized Version of Foxp2 Affects Cortico-Basal Ganglia Circuits in Mice. *Cell*, 137(5), 961–971. <https://doi.org/10.1016/j.cell.2009.03.041>

- Engel, A. K., & Fries, P. (2010). Beta-band oscillations—Signalling the status quo? *Current Opinion in Neurobiology*, *20*(2), 156–165.
<https://doi.org/10.1016/j.conb.2010.02.015>
- Fischer, P., Lipski, W. J., Neumann, W.-J., Turner, R. S., Fries, P., Brown, P., & Richardson, R. M. (2020). Movement-related coupling of human subthalamic nucleus spikes to cortical gamma. *eLife*, *9*, e51956.
<https://doi.org/10.7554/eLife.51956>
- Frühholz, S., Klaas, H. S., Patel, S., & Grandjean, D. (2015). Talking in Fury: The Cortico-Subcortical Network Underlying Angry Vocalizations. *Cerebral Cortex*, *25*(9), 2752–2762. <https://doi.org/10.1093/cercor/bhu074>
- Ghosh, S. S., Tourville, J. A., & Guenther, F. H. (2008). A Neuroimaging Study of Premotor Lateralization and Cerebellar Involvement in the Production of Phonemes and Syllables. *Journal of Speech, Language, and Hearing Research*, *51*(5), 1183–1202. [https://doi.org/10.1044/1092-4388\(2008/07-0119\)](https://doi.org/10.1044/1092-4388(2008/07-0119))
- Goodwill, A. M., Lum, J. A. G., Hendy, A. M., Muthalib, M., Johnson, L., Albein-Urios, N., & Teo, W.-P. (2017). Using non-invasive transcranial stimulation to improve motor and cognitive function in Parkinson's disease: A systematic review and meta-analysis. *Scientific Reports*, *7*(1), Article 1. <https://doi.org/10.1038/s41598-017-13260-z>
- Guenther, F. H. (2016). *Neural Control of Speech*. MIT Press.
- Hamilton, L. S., Edwards, E., & Chang, E. F. (2018). A Spatial Map of Onset and Sustained Responses to Speech in the Human Superior Temporal Gyrus. *Current Biology*, *28*(12), 1860-1871.e4. <https://doi.org/10.1016/j.cub.2018.04.033>

- Hammond, C., Bergman, H., & Brown, P. (2007). Pathological synchronization in Parkinson's disease: Networks, models and treatments. *Trends in Neurosciences*, 30(7), 357–364. <https://doi.org/10.1016/j.tins.2007.05.004>
- Hebb, A. O., Darvas, F., & Miller, K. J. (2012). Transient and state modulation of beta power in human subthalamic nucleus during speech production and finger movement. *Neuroscience*, 202, 218–233. <https://doi.org/10.1016/j.neuroscience.2011.11.072>
- Hell, F., Plate, A., Mehrkens, J. H., & Bötzel, K. (2023). Subthalamic oscillatory activity during normal and impaired speech. *Clinical Neurophysiology*. <https://doi.org/10.1016/j.clinph.2023.02.166>
- Hickok, G. (2012). Computational neuroanatomy of speech production. *Nature Reviews Neuroscience*, 13(2), 135–145. <https://doi.org/10.1038/nrn3158>
- Hickok, G., & Poeppel, D. (2007). The cortical organization of speech processing. *Nature Reviews Neuroscience*, 8(5), 393–402. <https://doi.org/10.1038/nrn2113>
- Ho, A. K., Iansek, R., Marigliani, C., Bradshaw, J. L., & Gates, S. (1999). Speech impairment in a large sample of patients with Parkinson's disease. *Behavioural Neurology*, 11(3), 131–137.
- Horn, A., Neumann, W.-J., Degen, K., Schneider, G.-H., & Kühn, A. A. (2017). Toward an electrophysiological “sweet spot” for deep brain stimulation in the subthalamic nucleus. *Human Brain Mapping*, 38(7), 3377–3390. <https://doi.org/10.1002/hbm.23594>
- Houde, J. F., & Nagarajan, S. S. (2011). Speech Production as State Feedback Control. *Frontiers in Human Neuroscience*, 5. <https://doi.org/10.3389/fnhum.2011.00082>

- Hurst, J. A., Baraitser, M., Auger, E., Graham, F., & Norell, S. (1990). An extended Family with a Dominantly Inherited Speech Disorder. *Developmental Medicine & Child Neurology*, 32(4), 352–355. <https://doi.org/10.1111/j.1469-8749.1990.tb16948.x>
- Johari, K., Kelley, R. M., Tjaden, K., Patterson, C. G., Rohl, A. H., Berger, J. I., Corcos, D. M., & Greenlee, J. D. W. (2023). Human subthalamic nucleus neurons differentially encode speech and limb movement. *Frontiers in Human Neuroscience*, 17, 962909. <https://doi.org/10.3389/fnhum.2023.962909>
- Jonides, J., Schumacher, E. H., Smith, E. E., Koeppe, R. A., Awh, E., Reuter-Lorenz, P. A., Marshuetz, C., & Willis, C. R. (1998). The Role of Parietal Cortex in Verbal Working Memory. *The Journal of Neuroscience*, 18(13), 5026–5034. <https://doi.org/10.1523/JNEUROSCI.18-13-05026.1998>
- Jorge, A., Lipski, W. J., Wang, D., Crammond, D. J., Turner, R. S., & Richardson, R. M. (2022). Hyperdirect connectivity of opercular speech network to the subthalamic nucleus. *Cell Reports*, 38(10), 110477. <https://doi.org/10.1016/j.celrep.2022.110477>
- Karlsson, F., Olofsson, K., Blomstedt, P., Linder, J., & Van Doorn, J. (2013). Pitch Variability in Patients With Parkinson's Disease: Effects of Deep Brain Stimulation of Caudal Zona Incerta and Subthalamic Nucleus. *Journal of Speech, Language, and Hearing Research*, 56(1), 150–158. [https://doi.org/10.1044/1092-4388\(2012/11-0333\)](https://doi.org/10.1044/1092-4388(2012/11-0333))

- Klaas, H. S., Fröhlich, S., & Grandjean, D. (2015). Aggressive vocal expressions—an investigation of their underlying neural network. *Frontiers in Behavioral Neuroscience*, 9. <https://doi.org/10.3389/fnbeh.2015.00121>
- Klostermann, F., Ehlen, F., Vesper, J., Nubel, K., Gross, M., Marzinzik, F., Curio, G., & Sappok, T. (2008). Effects of subthalamic deep brain stimulation on dysarthrophonia in Parkinson's disease. *Journal of Neurology, Neurosurgery & Psychiatry*, 79(5), 522–529. <https://doi.org/10.1136/jnnp.2007.123323>
- Kompoliti, K., Wang, Q. E., Goetz, C. G., Leurgans, S., & Raman, R. (2000). Effects of central dopaminergic stimulation by apomorphine on speech in Parkinson's disease. *Neurology*, 54(2), Article 2. <https://doi.org/10.1212/wnl.54.2.458>
- Lanciego, J. L., Luquin, N., & Obeso, J. A. (2012). Functional Neuroanatomy of the Basal Ganglia. *Cold Spring Harbor Perspectives in Medicine*, 2(12). <https://doi.org/10.1101/cshperspect.a009621>
- Lee, H., Simpson, G. V., Logothetis, N. K., & Rainer, G. (2005). Phase locking of single neuron activity to theta oscillations during working memory in monkey extrastriate visual cortex. *Neuron*, 45(1), Article 1. <https://doi.org/10.1016/j.neuron.2004.12.025>
- Li, R., He, Y., Qin, W., Zhang, Z., Su, J., Guan, Q., Chen, Y., & Jin, L. (2022). Effects of Repetitive Transcranial Magnetic Stimulation on Motor Symptoms in Parkinson's Disease: A Meta-Analysis. *Neurorehabilitation and Neural Repair*, 36(7), Article 7. <https://doi.org/10.1177/15459683221095034>
- Lipski, W. J., Alhourani, A., Pirnia, T., Jones, P. W., Dastolfo-Hromack, C., Helou, L. B., Crammond, D. J., Shaiman, S., Dickey, M. W., Holt, L. L., Turner, R. S., Fiez, J.

- A., & Richardson, R. M. (2018). Subthalamic Nucleus Neurons Differentially Encode Early and Late Aspects of Speech Production. *The Journal of Neuroscience*, 38(24), 5620–5631. <https://doi.org/10.1523/JNEUROSCI.3480-17.2018>
- Lipski, W. J., Bush, A., Chrabaszcz, A., Crammond, D. J., Fiez, J. A., Turner, R. S., & Richardson, R. M. (2023). *Subthalamic nucleus neurons encode syllable sequence and phonetic characteristics during speech* [Preprint]. *Neuroscience*. <https://doi.org/10.1101/2023.12.11.569290>
- Lipski, W. J., Bush, A., Chrabaszcz, A., Crammond, D. J., Fiez, J. A., Turner, R. S., & Richardson, R. M. (2024). Subthalamic nucleus neurons encode syllable sequence and phonetic characteristics during speech. *Journal of Neurophysiology*, jn.00471.2023. <https://doi.org/10.1152/jn.00471.2023>
- Lipski, W. J., Wozny, T. A., Alhourani, A., Kondylis, E. D., Turner, R. S., Crammond, D. J., & Richardson, R. M. (2017). Dynamics of human subthalamic neuron phase-locking to motor and sensory cortical oscillations during movement. *Journal of Neurophysiology*, 118(3), 1472–1487. <https://doi.org/10.1152/jn.00964.2016>
- Lofredi, R., Neumann, W.-J., Bock, A., Horn, A., Huebl, J., Siegert, S., Schneider, G.-H., Krauss, J. K., & Kühn, A. A. (2018). Dopamine-dependent scaling of subthalamic gamma bursts with movement velocity in patients with Parkinson's disease. *eLife*, 7. <https://doi.org/10.7554/eLife.31895>
- Logemann, J. A., Fisher, H. B., Boshes, B., & Blonsky, E. R. (1978). Frequency and Cooccurrence of Vocal Tract Dysfunctions in the Speech of a Large Sample of

Parkinson Patients. *Journal of Speech and Hearing Disorders*, 43(1), 47–57.

<https://doi.org/10.1044/jshd.4301.47>

London, D., Fazl, A., Katlowitz, K., Soula, M., Pourfar, M. H., Mogilner, A. Y., & Kiani, R.

(2021). Distinct population code for movement kinematics and changes of

ongoing movements in human subthalamic nucleus. *eLife*, 10, e64893.

<https://doi.org/10.7554/eLife.64893>

Lundgren, S., Saeys, T., Karlsson, F., Olofsson, K., Blomstedt, P., Linder, J., Nordh, E.,

Zafar, H., & Van Doorn, J. (2011). Deep Brain Stimulation of Caudal Zona Incerta

and Subthalamic Nucleus in Patients with Parkinson's Disease: Effects on Voice

Intensity. *Parkinson's Disease*, 2011, 1–8. <https://doi.org/10.4061/2011/658956>

Manes, J. L., Bullock, L., Meier, A. M., Turner, R. S., Richardson, R. M., & Guenther, F.

H. (2024). A neurocomputational view of the effects of Parkinson's disease on

speech production. *Frontiers in Human Neuroscience*, 18, 1383714.

<https://doi.org/10.3389/fnhum.2024.1383714>

Mitchell, R. L. C., Jazdyk, A., Stets, M., & Kotz, S. A. (2016). Recruitment of

Language-, Emotion- and Speech-Timing Associated Brain Regions for

Expressing Emotional Prosody: Investigation of Functional Neuroanatomy with

fMRI. *Frontiers in Human Neuroscience*, 10.

<https://doi.org/10.3389/fnhum.2016.00518>

Moreau, C., Pennel-Ployart, O., Pinto, S., Plachez, A., Annic, A., Viallet, F., Destée, A.,

& Defebvre, L. (2011). Modulation of dysarthropneumophonia by low-frequency

STN DBS in advanced Parkinson's disease. *Movement Disorders*, 26(4), 659–

663. <https://doi.org/10.1002/mds.23538>

Neumann, W.-J., Staub-Bartelt, F., Horn, A., Schanda, J., Schneider, G.-H., Brown, P., & Kühn, A. A. (2017). Long term correlation of subthalamic beta band activity with motor impairment in patients with Parkinson's disease. *Clinical Neurophysiology: Official Journal of the International Federation of Clinical Neurophysiology*, 128(11), 2286–2291.

<https://doi.org/10.1016/j.clinph.2017.08.028>

Paulesu, E., Frith, C. D., & Frackowiak, R. S. J. (1993). The neural correlates of the verbal component of working memory. *Nature*, 362(6418), 342–345.

<https://doi.org/10.1038/362342a0>

Peter, J., Ferraioli, F., Mathew, D., George, S., Chan, C., Alalade, T., Salcedo, S. A., Saed, S., Tatti, E., Quartarone, A., & Ghilardi, M. F. (2022). Movement-related beta ERD and ERS abnormalities in neuropsychiatric disorders. *Frontiers in Neuroscience*, 16, 1045715. <https://doi.org/10.3389/fnins.2022.1045715>

Pichon, S., & Kell, C. A. (2013). Affective and Sensorimotor Components of Emotional Prosody Generation. *The Journal of Neuroscience*, 33(4), 1640–1650.

<https://doi.org/10.1523/JNEUROSCI.3530-12.2013>

Price, C. J. (2012). A review and synthesis of the first 20 years of PET and fMRI studies of heard speech, spoken language and reading. *NeuroImage*, 62(2), 816–847.

<https://doi.org/10.1016/j.neuroimage.2012.04.062>

Remple, M. S., Bradenham, C. H., Kao, C. C., Charles, P. D., Neimat, J. S., & Konrad, P. E. (2011). Subthalamic nucleus neuronal firing rate increases with Parkinson's disease progression: STN Neurophysiology in Early vs Late-Stage PD.

Movement Disorders, 26(9), 1657–1662. <https://doi.org/10.1002/mds.23708>

Riecker, A., Mathiak, K., Wildgruber, D., Erb, M., Hertrich, I., Grodd, W., & Ackermann, H. (2005). fMRI reveals two distinct cerebral networks subserving speech motor control. *Neurology*, *64*(4), 700–706.

<https://doi.org/10.1212/01.WNL.0000152156.90779.89>

Sharott, A., Gulberti, A., Hamel, W., Köppen, J. A., Münchau, A., Buhmann, C., Pötter-Nerger, M., Westphal, M., Gerloff, C., Moll, C. K. E., & Engel, A. K. (2018). Spatio-temporal dynamics of cortical drive to human subthalamic nucleus neurons in Parkinson's disease. *Neurobiology of Disease*, *112*, 49–62.

<https://doi.org/10.1016/j.nbd.2018.01.001>

Shimamoto, S. A., Ryapolova-Webb, E. S., Ostrem, J. L., Galifianakis, N. B., Miller, K. J., & Starr, P. A. (2013). Subthalamic Nucleus Neurons Are Synchronized to Primary Motor Cortex Local Field Potentials in Parkinson's Disease. *The Journal of Neuroscience*, *33*(17), 7220–7233. <https://doi.org/10.1523/JNEUROSCI.4676-12.2013>

Skodda, S., Grönheit, W., Schlegel, U., Südmeyer, M., Schnitzler, A., & Wojtecki, L. (2014). Effect of Subthalamic Stimulation on Voice and Speech in Parkinson's Disease: For the Better or Worse? *Frontiers in Neurology*, *4*.

<https://doi.org/10.3389/fneur.2013.00218>

Skodda, S., Visser, W., & Schlegel, U. (2010). Short- and long-term dopaminergic effects on dysarthria in early Parkinson's disease. *Journal of Neural Transmission (Vienna, Austria: 1996)*, *117*(2), Article 2.

<https://doi.org/10.1007/s00702-009-0351-5>

- Stolk, A., Brinkman, L., Vansteensel, M. J., Aarnoutse, E., Leijten, F. S., Dijkerman, C. H., Knight, R. T., de Lange, F. P., & Toni, I. (2019). Electrocorticographic dissociation of alpha and beta rhythmic activity in the human sensorimotor system. *eLife*, *8*, e48065. <https://doi.org/10.7554/eLife.48065>
- Tankus, A., & Fried, I. (2019). Degradation of Neuronal Encoding of Speech in the Subthalamic Nucleus in Parkinson's Disease. *Neurosurgery*, *84*(2), 378–387. <https://doi.org/10.1093/neuros/nyy027>
- Tankus, A., Lustig, Y., Fried, I., & Strauss, I. (2021). Impaired Timing of Speech-Related Neurons in the Subthalamic Nucleus of Parkinson Disease Patients Suffering Speech Disorders. *Neurosurgery*, *89*(5), 800–809. <https://doi.org/10.1093/neuros/nyab293>
- Theys, C., Jaakkola, E., Melzer, T. R., De Nil, L. F., Guenther, F. H., Cohen, A. L., Fox, M. D., & Joutsa, J. (2024). Localization of stuttering based on causal brain lesions. *Brain*, *147*(6), 2203–2213. <https://doi.org/10.1093/brain/awae059>
- Tinkhauser, G., Torrecillos, F., Duclos, Y., Tan, H., Pogosyan, A., Fischer, P., Carron, R., Welter, M.-L., Karachi, C., Vandenberghe, W., Nuttin, B., Witjas, T., Régis, J., Azulay, J.-P., Eusebio, A., & Brown, P. (2018). Beta burst coupling across the motor circuit in Parkinson's disease. *Neurobiology of Disease*, *117*, 217–225. <https://doi.org/10.1016/j.nbd.2018.06.007>
- Törnqvist, A. L., Schalén, L., & Rehnström, S. (2005). Effects of different electrical parameter settings on the intelligibility of speech in patients with Parkinson's disease treated with subthalamic deep brain stimulation. *Movement Disorders*, *20*(4), 416–423. <https://doi.org/10.1002/mds.20348>

- Tourville, J. A., & Guenther, F. H. (2011). The DIVA model: A neural theory of speech acquisition and production. *Language and Cognitive Processes*, 26(7), Article 7. <https://doi.org/10.1080/01690960903498424>
- Towle, V. L., Yoon, H.-A., Castelle, M., Edgar, J. C., Biassou, N. M., Frim, D. M., Spire, J.-P., & Kohrman, M. H. (2008). ECoG gamma activity during a language task: Differentiating expressive and receptive speech areas. *Brain*, 131(8), 2013–2027. <https://doi.org/10.1093/brain/awn147>
- Tripoliti, E., Zrinzo, L., Martinez-Torres, I., Tisch, S., Frost, E., Borrell, E., Hariz, M. I., & Limousin, P. (2008). Effects of contact location and voltage amplitude on speech and movement in bilateral subthalamic nucleus deep brain stimulation. *Movement Disorders*, 23(16), 2377–2383. <https://doi.org/10.1002/mds.22296>
- Tykalova, T., Novotny, M., Ruzicka, E., Dusek, P., & Rusz, J. (2022). Short-term effect of dopaminergic medication on speech in early-stage Parkinson's disease. *NPJ Parkinson's Disease*, 8(1), Article 1. <https://doi.org/10.1038/s41531-022-00286-y>
- Van Lancker Sidtis, D., Rogers, T., Godier, V., Tagliati, M., & Sidtis, J. J. (2010). Voice and Fluency Changes as a Function of Speech Task and Deep Brain Stimulation. *Journal of Speech, Language, and Hearing Research*, 53(5), 1167–1177. [https://doi.org/10.1044/1092-4388\(2010/09-0154\)](https://doi.org/10.1044/1092-4388(2010/09-0154))
- Vanlanckersidtis, D., Pachana, N., Cummings, J., & Sidtis, J. (2006). Dysprosodic speech following basal ganglia insult: Toward a conceptual framework for the study of the cerebral representation of prosody. *Brain and Language*, 97(2), 135–153. <https://doi.org/10.1016/j.bandl.2005.09.001>

- Vinck, M., Battaglia, F. P., Womelsdorf, T., & Pennartz, C. (2012). Improved measures of phase-coupling between spikes and the Local Field Potential. *Journal of Computational Neuroscience*, 33(1), 53–75. <https://doi.org/10.1007/s10827-011-0374-4>
- Vinck, M., van Wingerden, M., Womelsdorf, T., Fries, P., & Pennartz, C. M. A. (2010). The pairwise phase consistency: A bias-free measure of rhythmic neuronal synchronization. *NeuroImage*, 51(1), 112–122. <https://doi.org/10.1016/j.neuroimage.2010.01.073>
- Vissani, M. (2024). *Sample code and preprocessed dataset for: Spike-phase coupling of subthalamic neurons to posterior opercular cortex predicts speech sound accuracy* (Version v1.0.1) [Computer software]. Zenodo. <https://doi.org/10.5281/ZENODO.12610957>
- Vissani, M., Palmisano, C., Volkmann, J., Pezzoli, G., Micera, S., Isaias, I. U., & Mazzoni, A. (2021). Impaired reach-to-grasp kinematics in parkinsonian patients relates to dopamine-dependent, subthalamic beta bursts. *Npj Parkinson's Disease*, 7(1), 53. <https://doi.org/10.1038/s41531-021-00187-6>
- Warren, J. D., Smith, H. B., Denson, L. A., & Waddy, H. M. (2000). Expressive language disorder after infarction of left lentiform nucleus. *Journal of Clinical Neuroscience*, 7(5), 456–458. <https://doi.org/10.1054/jocn.1999.0238>
- Watson, P., & Montgomery, E. (2006). The relationship of neuronal activity within the sensori-motor region of the subthalamic nucleus to speech. *Brain and Language*, 97(2), Article 2. <https://doi.org/10.1016/j.bandl.2005.11.004>

- Weiss, A. R., Korzeniewska, A., Chrabaszczyk, A., Bush, A., Fiez, J. A., Crone, N. E., & Richardson, R. M. (2023). Lexicality-Modulated Influence of Auditory Cortex on Subthalamic Nucleus During Motor Planning for Speech. *Neurobiology of Language*, 4(1), 53–80. https://doi.org/10.1162/nol_a_00086
- Westfall, P. H., Young, S. S., & Wright, S. P. (1993). On Adjusting P-Values for Multiplicity. *Biometrics*, 49(3), 941. <https://doi.org/10.2307/2532216>
- Wildgruber, D., Ackermann, H., & Grodd, W. (2001). Differential Contributions of Motor Cortex, Basal Ganglia, and Cerebellum to Speech Motor Control: Effects of Syllable Repetition Rate Evaluated by fMRI. *NeuroImage*, 13(1), 101–109. <https://doi.org/10.1006/nimg.2000.0672>
- Zarate, J. M., & Zatorre, R. J. (2008). Experience-dependent neural substrates involved in vocal pitch regulation during singing. *NeuroImage*, 40(4), 1871–1887. <https://doi.org/10.1016/j.neuroimage.2008.01.026>
- Zarei, M., Jahed, M., & Daliri, M. R. (2018). Introducing a Comprehensive Framework to Measure Spike-LFP Coupling. *Frontiers in Computational Neuroscience*, 12. <https://doi.org/10.3389/fncom.2018.00078>